*Report*

# BBSome-deficient cells activate intraciliary CDC42 to trigger actin-dependent ciliary ectocytosis

Avishek Prasai [ID] [1,2,3,5], Olha Ivashchenko [ID] [1,2,5], Kristyna Maskova [ID] [1], Sofiia Bykova [ID] [1], Marketa Schmidt Cernohorska [ID] [1,4], Ondrej Stepanek [ID] [1] & Martina Huranova [ID] [1✉]

## Abstract

**Bardet-Biedl syndrome (BBS) is a pleiotropic ciliopathy caused by dysfunction of the BBSome, a cargo adaptor essential for export of transmembrane receptors from cilia. Although actin-dependent ectocytosis has been proposed to compensate defective cargo retrieval, its molecular basis remains unclear, especially in relation to BBS pathology. In this study, we investigated how actin polymerization and ectocytosis are regulated within the cilium. Our findings reveal that ciliary CDC42, a RHO-family GTPase triggers in situ actin polymerization, ciliary ectocytosis, and cilia shortening in BBSome-deficient cells. Activation of the Sonic Hedgehog pathway further enhances CDC42 activity specifically in BBSome-deficient cilia. Inhibition of CDC42 in BBSome-deficient cells decreases the frequency and duration of ciliary actin polymerization events, causing buildup of G protein coupled receptor 161 (GPR161) in bulges along the axoneme during Sonic Hedgehog signaling. Overall, our study identifies CDC42 as a key trigger of ciliary ectocytosis. Hyperactive ciliary CDC42 and ectocytosis and the resulting loss of ciliary material might contribute to BBS disease severity.**

**Keywords** Bardet-Biedl Syndrome; Cilium; Ectocytosis; Actin; CDC42
**Subject Categories** Cell Adhesion, Polarity & Cytoskeleton; Membranes & Trafficking; Signal Transduction

## Introduction

Primary cilia are microtubule-based organelles protruding from the cell surface. Primary cilia, referred to as 'cilia' onwards, are rich in signaling receptors, which sense various extracellular stimuli. Particular mutations in genes critical for cilia function cause pleiotropic human diseases, collectively called ciliopathies (reviewed in (Reiter and Leroux, 2017)). Bardet-Biedl syndrome (BBS) is a multi-organ ciliopathy caused by dysfunction of the BBSome, an octameric cargo adaptor involved in export of specific G-protein coupled receptors from primary cilia (Jin et al, 2010; Mukhopadhyay et al, 2013; Nager et al, 2017; Nozaki et al, 2018; Ye et al, 2018). BBS presents with developmental and functional anomalies in the retina, brain, kidney, liver, heart, and other organs (Forsythe and Beales, 2013; Niederlova et al, 2019). However, the molecular mechanisms of how the BBSome deficiency leads to the particular pathological outcomes in BBS are still incompletely understood.

The dysfunction of the BBSome results in the accumulation of signaling receptors in the cilia. This triggers the alternative pathway to shut down the signaling via the release of the ciliary receptors in ciliary vesicles (Lechtreck et al, 2013; Nager et al, 2017; Nozaki et al, 2018). BBSome-deficient cells have usually shorter cilia than wild type (WT) cells (Chiuso et al, 2023; Hernandez-Hernandez et al, 2013; Prasai et al, 2020; Uytingco et al, 2019), which could be a consequence of the continuous ectocytosis.

The removal of the ciliary membrane via ectocytosis regulates ciliary signaling also under physiological conditions and drives cilia disassembly prior mitosis (Kanamaru et al, 2022; Loukil et al, 2021; Nager et al, 2017; Phua et al, 2017; Stilling et al, 2022; Wang et al, 2019). The ectosomes are formed at the ciliary tip and are released upon an actin polymerization event inside cilia (Loukil et al, 2021; Phua et al, 2017; Wang et al, 2019). In the cytoplasm, actin polymerization is controlled by the GTPases of the RHO family, which cycle between the GTP and GDP loaded state (Hodge and Ridley, 2016). RHO GTPases are vital for diverse cellular processes, including cell polarization, migration, and division, regulating cytoskeletal dynamics, membrane remodeling, and signaling pathways. Several RHO family members (CDC42, RHOA, RHOC, RAC1), their regulators and effectors were detected in the proteomic analyses of the ciliary content (Kohli et al, 2017; Mick et al, 2015). Although studies on actin polymerization in the cilia are only emerging (Kiesel et al, 2020; Lee et al, 2018; Loukil et al, 2021; Phua et al, 2017), roles of cortical F-actin and its regulators from the RHO family in ciliogenesis are well established (Drummond et al, 2018; Hernandez-Hernandez et al, 2013; Pan et al, 2007; Pitaval et al, 2010; Rangel et al, 2019; Saito et al, 2017; Stilling et al, 2022; Zuo et al, 2011). Whether and how the individual RHO GTPases regulate actin polymerization in the cilia and the ectocytosis has not been addressed.

CDC42, a member of the RHO GTPase family, is a crucial regulator of cell polarity and actin-based morphogenesis (Pichaud

[1]Laboratory of Adaptive Immunity, Institute of Molecular Genetics of the Czech Academy of Sciences, Prague, Czech Republic. [2]Faculty of Science, Department of Developmental and Cell Biology, Charles University, Prague, Czech Republic. [3]Center for Molecular Signaling (PZMS), Department of Medical Biochemistry and Molecular Biology, Saarland University School of Medicine, Homburg, Germany. [4]Max Perutz Labs, University of Vienna, Vienna Biocenter (VBC), Vienna, Austria. [5]These authors contributed equally: Avishek Prasai, Olha Ivashchenko. ✉E-mail: martina.huranova@img.cas.cz

et al, 2019). In this study, we demonstrate that actin polymerization within primary cilia is regulated by a ciliary pool of CDC42. In addition, we discovered that CDC42 is hyperactivated during Sonic Hedgehog (SHH) signaling in BBSome-deficient cells. Finally, we show that intraciliary CDC42 activity facilitates the ectocytosis of excess GPCRs when their retrograde transport is defective, a condition observed in ciliopathies such as BBS.

## Results and discussion

### Differential roles of RHO family members in the regulation of the cilia length

The BBSome functions mostly as a retrograde cargo adaptor for the G-protein coupled receptors (GPCRs) in cilia (Ye et al, 2018). Loss of BBSome alters the ciliary export leading to ectocytosis of accumulated cargoes as cilia-derived vesicles (Nager et al, 2017). This pivotal work in the field of the ciliary ectocytosis also demonstrated that the BBS mutants undergo constitutive ectocytosis, thus independent of experimentally-induced GPCR stimulation (Nager et al, 2017). We hypothesized that the cilia shortening in the BBSome-deficient cell lines, previously observed by others (Chiuso et al, 2023; Hernandez-Hernandez et al, 2013; Uytingco et al, 2019) and us (Prasai et al, 2020) (Fig. 1A,B) is caused by constitutive ectocytosis. We observed that a portion of the primary cilia in the $BBS4^{KO/KO}$, $BBS1^{KO/KO}$, and $BBS7^{KO/KO}$, and to lesser extent also WT RPE1 cell lines exhibit enlarged cilia tips, possibly representing nascent ectosomes (Figs. 1C and EV1A,B).

Since actin polymerization is typically driven by different RHO family GTPases depending on the context, we aimed to identify the RHO family member(s) triggering the actin polymerization in cilia during ectocytosis. We treated the cells with established inhibitors of the three major RHO GTPases: Y27632 (inhibits ROCK1, a downstream effector of RHOA (Hernandez-Hernandez et al, 2013)), ML141 (inhibits CDC42 (Hong et al, 2013)) and CAS 1177865-17-6 (shortly CAS; inhibits RAC1 (Gao et al, 2004)) (Figs. 1D and EV1C). As expected, the inhibition of ROCK1 prolonged cilia both in parental WT (WT), $BBS1^{KO/KO}$, and $BBS4^{KO/KO}$ RPE1 cells (Fig. 1D,E) (Hernandez-Hernandez et al, 2013), consistent with the role of RHOA-dependent cortical F-actin network in the regulation of the cilia length (Pan et al, 2007; Rangel et al, 2019). Accordingly, the treatment with Cytochalasin D and ARP2/3 inhibitor resulted in cilia prolongation in both WT and $BBS4^{KO/KO}$ RPE1 cell lines (Fig. EV1C,D). In contrast, RAC1 inhibition induced only very subtle changes in cilia length (Fig. EV1E). Only the inhibition of CDC42 prolonged cilia specifically in the BBSome-deficient cells, whereas the treatment had no effect in WT cells (Fig. 1D,F). Moreover, we observed foci with accumulated ciliary membrane marker ARL13B extending ~0.5 μm beyond the cilia axoneme (Ac-tub), particularly at the ciliary tips of $BBS4^{KO/KO}$ cells (Figs. 1G,H and EV1F). Inhibition of CDC42 increased the amount of the ARL13B foci specifically in the $BBS4^{KO/KO}$ cells (Fig. 1H). $BBS4^{KO/KO}$ RPE cells expressing YFP-BBS4 (Prasai et al, 2020) show rescued cilia length and foci formation and are not sensitive to CDC42 inhibition in this respect (Fig. EV1G,H).

In the next step, we utilized WT and $Bbs4^{KO/KO}$ mouse embryonic fibroblasts (MEFs) (Tsyklauri et al, 2021) to verify our observations in a primary cell line model that responds to SHH signaling. Unlike in WT RPE1 cells, inhibition of CDC42 in WT MEFs resulted in

cilia shortening (Fig. 1I,J). However, inhibition of CDC42 increased the cilia length in the in $Bbs4^{KO/KO}$ MEFs corroborating our initial findings in RPE1 cells regarding the role of CDC42 in ciliary ectocytosis (Fig. 1F–J). In some instances, we observed that the ciliary membrane extended beyond the axoneme by more than 0.5 μm upon the inhibition of CDC42 indicating sustained ectocytosis in these cells (Figs. 1F and EV1I).

CDC42 has been shown to control multiple aspects of ciliogenesis including its involvement in vesicular transport (Zuo et al, 2011), endocytosis (Saito et al, 2017) and signal transduction (Drummond et al, 2018). The proteomic analysis identified CDC42 within cilia (Kohli et al, 2017; Mick et al, 2015) and it is so far the only GTPase with documented localization at the basal body (Drummond et al, 2018). CDC42 apparently cycles between the basal body and cilia and exerts its functions via engagement with specific effectors in a spatiotemporal manner.

To directly address that the reduced cilia length in BBS-deficient cells is caused by CDC42-dependent ectocytosis, we further quantified the amount of cilia-derived extracellular vesicles in the supernatants of WT and $Bbs4^{KO/KO}$ MEFs expressing GPR161-mCherry after a short pulse of SHH signaling (Fig. 1K). To ensure that cell debris did not contaminate our EV fractions, we tested them for the presence of Calnexin, an ER marker, and found it to be mostly undetectable in our EV preparations (Fig. 1K). We observed that $Bbs4^{KO/KO}$ MEFs release more GPR161 and IFT88 positive vesicles compared to the WT MEFs (Figs. 1L,M and EV1J,K), whereas the total amount of exosomes was comparable as shown by CD9 staining, respectively (Loukil et al, 2021; Volz et al, 2021). TSG101, another marker of exosomes (Volz et al, 2021), was hardly detectable in the supernatant in these conditions. This low detection of exosomes could be due to the short window of EV production, aligning with previous findings that showed negligible amounts of TSG101 after 24 h of EV enrichment in WT conditions (Nager et al, 2017; Volz et al, 2021). Concomitant inhibition of CDC42 with SHH triggering lead to decrease in the ciliary vesicle release in $Bbs4^{KO/KO}$ MEFs (Fig. 1K–M).

Overall, these findings imply that the loss of BBSome triggers CDC42-mediated ectocytosis and concurrent shortening of cilia in various cell lines.

### CDC42 controls GPR161 and cilia dynamics in BBSome-deficient cells

SHH signaling initiates the BBSome-dependent removal of GPR161 from the cilia, which triggers the downstream signaling (Mukhopadhyay et al, 2013). In line with the previous results (Nager et al, 2017; Nozaki et al, 2018), we observed increased frequency of GPR161 positive cilia in $Bbs4^{KO/KO}$ MEF cells in the steady state, which was not altered after activation of the SHH pathway via the SMO agonist (SAG) (Fig. 2A,B). We observed GPR161 localized to specific foci at the ciliary tip of SAG-stimulated $Bbs4^{KO/KO}$ MEFs, but not WT cells (Fig. 2C). Moreover, SHH signaling decreased the cilia length in both WT and $Bbs4^{KO/KO}$ MEFs. Since the SHH-induced cilia shortening in the $Bbs4^{KO/KO}$, but not WT, cells was inhibited by ML141, the mechanisms of cilia shortening are probably different in these two lines (Fig. 2A,D,E). Whereas in the WT cells, the cilia shortening might reflect the removal of GPR161 from cilia via endocytosis (Pal et al, 2016), ectocytosis is the candidate mechanism for SHH-induced shortening of cilia in the

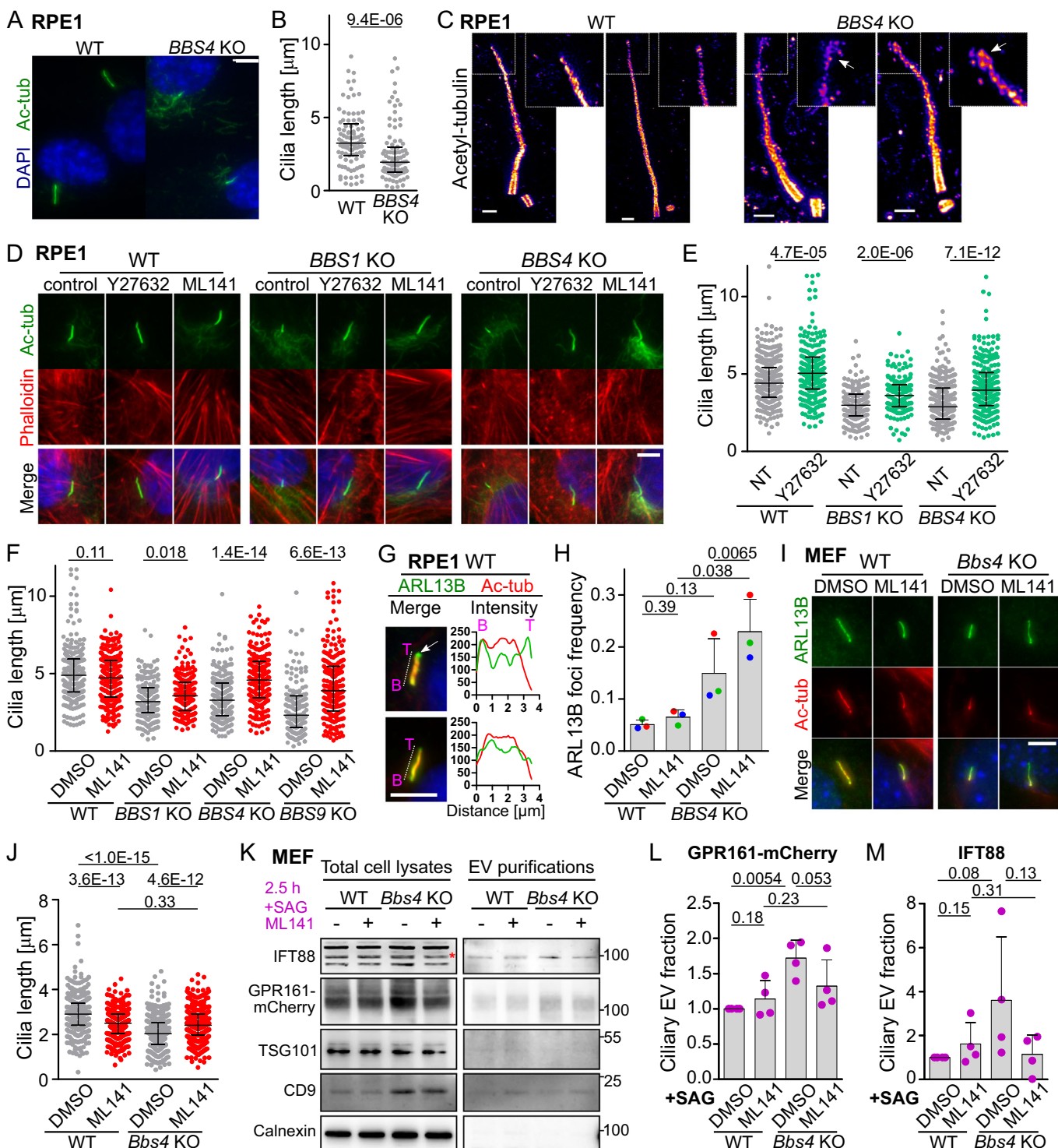

BBS-deficient cells (Fig. 1D,E). It has been shown that ectocytosis of GPCRs is much less effective than their retrograde transport (Nager et al, 2017). The replenishment of the ectocytosed pool by newly imported GPR161 molecules therefore explains why we did not observe a clear depletion of GPR161 from cilia in SAG-treated *Bbs4*^KO/KO MEFs (Fig. 2A,B). On the other hand, the cilia shorten in these conditions, which leads to reduction of the total amount of

ciliary GPR161. This is conceptually intriguing, as it implies that the rapid decrease in cAMP levels caused by the sudden removal of GPR161 with PKA via ectosomes is sufficient to sustain substantial levels of SHH signaling in the BBSome-deficient conditions (Bachmann et al, 2016; Nager et al, 2017; Zhang et al, 2012). Given that the signal of the endogenous GPR161 is generally low, we also examined the dynamics of the ciliary membrane marker

**Figure 1. RHO family members regulate differentially the cilia length.**

(A, B) Representative micrographs of cilia (A) and quantification of the cilia length (B), stained with antibody to acetylated tubulin (Ac-tub), in the WT and $BBS4^{KO/KO}$ RPE1 cells. Scale bar, 5 μm. Medians with interquartile range from three independent experiment ($n = 96$ cilia). (C) Expansion microscopy of the cilia axoneme of the WT and $BBS4^{KO/KO}$ RPE1 cells visualized via staining with antibody to acetylated tubulin. Enlarged insets and white arrows point to bulges at cilia tips observed in $BBS4^{KO/KO}$ cells. Scale bar, 2 μm. (D) Representative micrographs of cilia visualized by staining with antibody to acetylated tubulin (Ac-tub) and staining of F-actin (Phalloidin) in WT, $BBS1^{KO/KO}$, and $BBS4^{KO/KO}$ RPE1 cells treated with the ROCK1 inhibitor Y27632 or CDC42 inhibitor ML141 for 2 h. Cilia base at bottom. Scale bar, 5 μm. (E) Quantification of the cilia length based on Ac-tub signal in the WT, $BBS1^{KO/KO}$, and $BBS4^{KO/KO}$ RPE1 cells non-treated (NT) or treated with Y27632 or treated with Y27632 for 2 h. Medians with interquartile range from three independent experiments ($n = 160$–290 cilia). (F) Quantification of the cilia length based on Ac-tub signal in the WT, $BBS1^{KO/KO}$, $BBS4^{KO/KO}$, and $BBS9^{KO/KO}$ RPE1 cells treated with vehicle or ML141 for 2 h. Medians with interquartile range from three independent experiments ($n = 145$–270 cilia). (G) Merged micrographs (left) show representative cilia stained with antibodies against ARL13B (green) and Ac-tubulin (red), along with the corresponding intensity line scans (dashed line next to the cilium) from the cilia base (B) to the cilia tip (T) in WT RPE1 cells. The arrow indicates the foci at the tip of the top cilium, defined as the ARL13B signal extending (-0.5 μm) beyond the Ac-tubulin signal, as illustrated in the intensity plot on the right. The bottom cilium does not exhibit foci based on this criterion. Scale bar, 5 μm. (H) Quantification of the frequency of ciliary tip foci observed in (G). Mean and SD of three independent experiments ($n = 100$–200 cilia). (I, J) Representative micrographs (I) and quantification of the cilia length (J), in WT and $Bbs4^{KO/KO}$ MEFs treated with vehicle or ML141 and stained with antibodies to acetylated tubulin (Ac-tub) and ARL13B. Cilia base at bottom. Scale bar, 5 μm. Medians with interquartile range from three independent experiments ($n = 300$–400 cilia). (K) Representative Western blots from total cell lysates (TCL, left blots) and EVs purifications (right blots) prepared from GPR161-mCherry WT and $Bbs4^{KO/KO}$ MEFs treated with SAG and vehicle or ML141 for 2.5 h probed with antibodies to IFT88 (red star—middle band), mCherry, TSG101, CD9 and Calnexin. (L, M) Quantification of the GPR161-mCherry (L) and IFT88 (M) present in the ciliary EV fraction purified by ultracentrifugation of cell supernatants of GPR161-mCherry WT and $Bbs4^{KO/KO}$ MEFs treated with SAG and vehicle or ML141 for 2.5 h. Mean and SD of four independent experiments. Data information: Statistical significance was calculated using the two-tailed Mann–Whitney test (B, E, F, J), two-tailed paired t-test (H), and one-tailed paired t-test (L, M) and the obtained p-values are indicated. Merged micrographs show nuclei staining by DAPI—blue (D, G, I). Source data are available online for this figure.

ARL13B along with cilia length in both WT and $Bbs4^{KO/KO}$ MEFs in the steady state and upon SHH pathway triggering (Fig. EV2A,B). We observed that frequency of ARL13B foci is higher in $Bbs4^{KO/KO}$ MEFs and increases upon SAG treatment (Fig. EV2B). Inhibition of CDC42 prevents cilia shortening and leads to the accumulation of ARL13B foci in $Bbs4^{KO/KO}$ MEFs, whereas in WT cells, the frequency of ARL13B foci remains relatively stable across all tested conditions (Fig. EV2A,B).

To address whether cytoplasmic or ciliary CDC42 promotes the cilia shortening in BBSome-deficient cells, we expressed WT or dominant negative (DN) CDC42 with a ciliary targeting motif (Mick et al, 2015) in the WT and Bbs4-deficient MEFs (Fig. 2F). We observed that CDC42-DN abrogated cilia shortening and promoted the formation of the GPR161 foci in the cells lacking the BBSome in the steady state (Fig. 2F–H). Activation of the SHH pathway lead to comparable cilia shortening and GPR161 removal in CDC42-WT and DN expressing WT MEFs (Fig. 2G,I). We observed that expressing cilia-targeted CDC42-WT in $Bbs4^{KO/KO}$ MEFs enhanced ectocytosis of GPR161, as evidenced by a decreased frequency of GPR161 positive cilia and the depletion of GPR161 foci (Fig. 2H,I). This indicates that CDC42 might be the limiting factor for this process. In contrast, CDC42-DN blocked SAG-induced cilia shortening and removal of ciliary GPR161 in the $Bbs4^{KO/KO}$ MEFs (Fig. 2G–I). These findings demonstrate that both the chemical and genetic inhibition of CDC42 yield the same effect, ruling out possible off-target effects.

The outcomes of perturbing CDC42 function have been rather contradictory (Drummond et al, 2018; Zuo et al, 2011). These discrepancies could be potentially explained by the intricate interplay between the extraciliary and intraciliary functions of CDC42 in ciliogenesis with variable outcomes, depending on the specific context and cell type. CDC42 knockdown or expression of a non-targeted DN CDC42 mutant impairs the CDC42 activity both in the cilia and in the cell body and thus, cannot discriminate between the specific and perhaps counteracting roles of these two pools (Drummond et al, 2018; Zuo et al, 2011). In contrast, our approach using ciliary-targeted CDC42-DN specifically addresses the function of the ciliary pool of CDC42. Collectively, these data demonstrate that ciliary CDC42 promotes ectocytosis, leading to

the cilia shortening in cells lacking the BBSome, both in the steady state and during SHH signaling.

## CDC42 is hyperactivated in cilia in BBSome-deficient cells during SHH signaling

To directly assess the activity of CDC42 inside the cilia, we expressed the Raichu-CDC42 FRET-FLIM genetic probe (Yoshizaki et al, 2003) appended with an N-terminal ciliary anchor (Mick et al, 2015) (N-Raichu-CDC42) in WT and $Bbs4^{KO/KO}$ MEFs (Fig. 3A,B). Activation of CDC42 in the probe leads to an intramolecular interaction with PAK resulting in the Förster resonance energy transfer (FRET) from a donor (CFP) to acceptor (YFP) and thus decrease in donor lifetime. We measured the activity of CDC42 in cilia in non-stimulated cells and after SHH activation (Fig. 3C,D). The estimated FRET efficiency corresponded to the fraction of the reporter CDC42 molecules in the active conformation (Fig. 3E). We detected basal activity of CDC42 in cilia in both the WT and $Bbs4^{KO/KO}$ cells as depicted by the general decrease in CFP lifetime when compared to the no-FRET reference control N-CFP-PAK-CDC42 or to the Raichu probe treated with the CDC42 inhibitor (Figs. 3C and EV2C). In the steady state, the CDC42-GTP fraction was slightly higher in the $Bbs4^{KO/KO}$ cells (Fig. 3E), which could explain the low-grade constitutive CDC42-dependent ciliary ectocytosis in these cells. The activity of CDC42 was unaffected by the SHH signaling in the WT cells (Fig. 3C–E), indicating that the concomitant cilia shortening is CDC42-independent (Fig. 2E). On the other hand, the activation of the SHH pathway in $Bbs4^{KO/KO}$ cells lead to substantial increase in the CDC42 activity (Fig. 3C–E). We examined whether increased CDC42 activity led to cilia shortening via the ectocytosis by correlating the FRET-FLIM data with cilia length (Fig. 3F). We observed a moderate correlation between cilia length and donor lifetime in $Bbs4^{KO/KO}$ MEFs stimulated with SAG (Fig. 3F), which suggests higher CDC42 activity in shortened cilia.

Despite the detection of RHO GTPases and their regulators within cilia (Kohli et al, 2017; Mick et al, 2015), their specific functions within cilia remain largely unknown. To date, only the

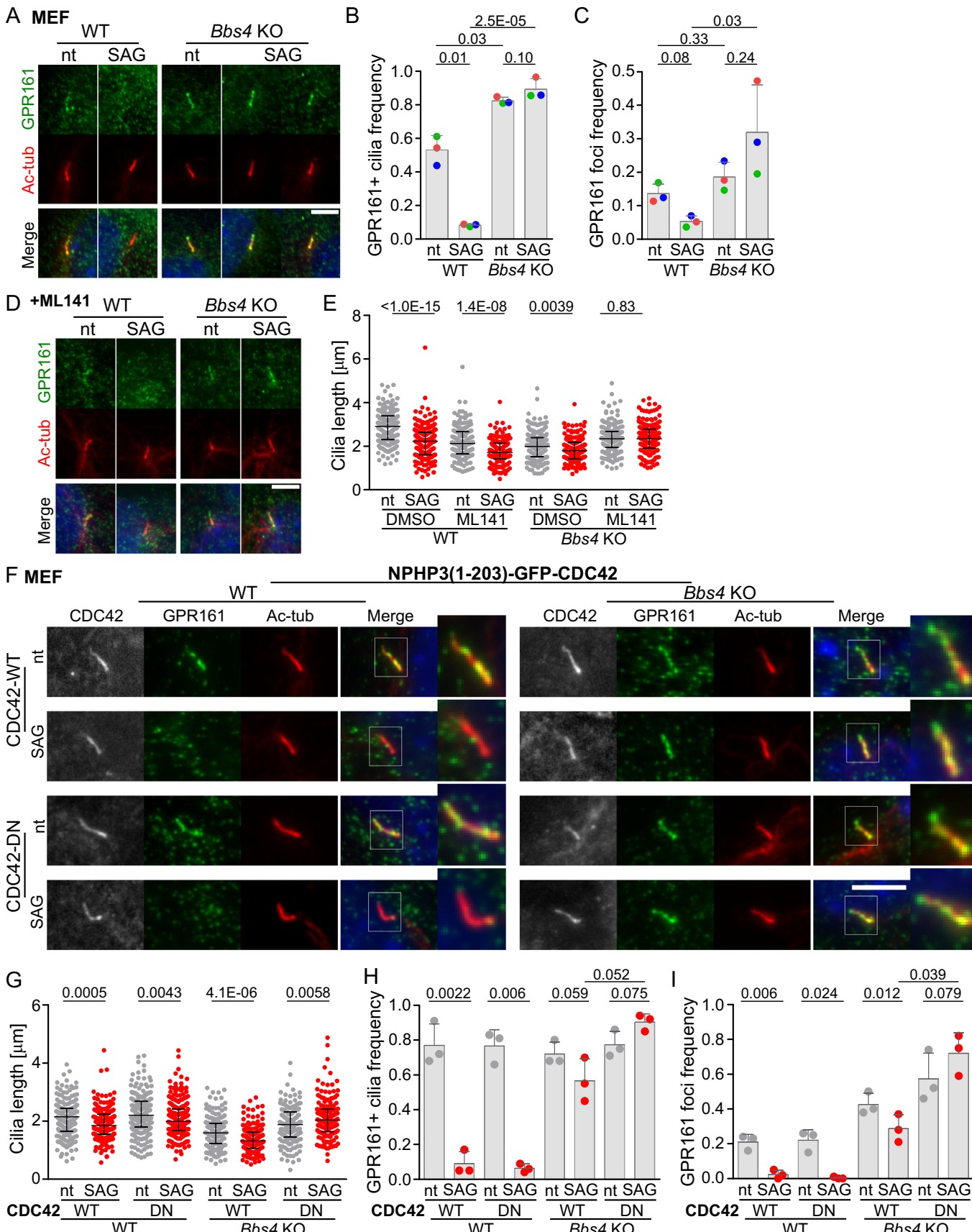

**Figure 2.   CDC42 controls GPR161 and cilia dynamics in BBSome-deficient cells.**

(A) Representative micrographs depict cilia visualized via staining with antibodies to acetylated tubulin (Ac-tub) and GPR161 in non-treated (nt) and SAG induced—2 h, WT and $Bbs4^{KO/KO}$ MEFs. Cilia base at bottom. Scale bar, 5 μm. (B, C) Quantification of the frequency of GPR161 positive cilia (B) and of the frequency of GPR161 foci at the cilia tip (C) in non-treated (nt) and SAG induced—2 h, WT and $Bbs4^{KO/KO}$ MEFs in (A). Mean and SD of three independent experiments ($n = 160$–$210$ cilia). (D) Representative micrographs depict cilia visualized via staining with antibodies to acetylated tubulin (Ac-tub) and GPR161 in non-treated (nt) and SAG induced—2 h, WT and $Bbs4^{KO/KO}$ MEFs concomitantly treated with ML141. Cilia base at bottom. Scale bar, 5 μm. (E) Quantification of the cilia length, in non-treated (nt) and SAG induced WT and $Bbs4^{KO/KO}$ MEFs concomitantly treated with DMSO or ML141 and stained with antibodies to acetylated tubulin (Ac-tub) and GPR161. Medians with interquartile range from three independent experiments ($n = 170$–$190$ cilia). (F) Representative micrographs depict cilia visualized via staining with antibodies to acetylated tubulin (Ac-tub) and GPR161 (Merge and insets), in non-treated (nt) and SAG induced—2 h, WT and $Bbs4^{KO/KO}$ MEFs expressing the cilia targeted GFP-CDC42 WT or DN variant. Cilia base at bottom. Scale bar, 5 μm. (G) Quantification of the cilia length, in non-treated (nt) and SAG induced—2 h, WT, and $Bbs4^{KO/KO}$ MEFs expressing cilia targeted GFP-CDC42 WT or DN variant stained with antibodies to acetylated tubulin (Ac-tub) and GPR161. Medians with interquartile range from three independent experiments ($n = 160$–$220$ cilia). (H) Quantification of the frequency of GPR161 positive cilia in non-treated (nt) and SAG induced—2 h, WT and $Bbs4^{KO/KO}$ MEFs expressing cilia targeted GFP-CDC42 WT or DN variant. Mean and SD of three independent experiments ($n = 160$–$190$ cilia). (I) Quantification of the frequency of GPR161 tip foci in non-treated (nt) and SAG induced—2 h, WT and $Bbs4^{KO/KO}$ MEFs expressing cilia targeted GFP-CDC42 WT or DN variant in (H). Mean and SD of three independent experiments ($n = 160$–$190$ cilia). Data information: Statistical significance was calculated using two-tailed paired t-test (B, C, H, I), and two-tailed Mann–Whitney test (E, G) and the obtained p-values are indicated. Merged micrographs show nuclei staining with DAPI—blue (A, D, F). Source data are available online for this figure.

function of the TrioGEF-RHOA module within neuronal cilia was investigated, using a cilia-targeted FRET probe to measure RHOA activity in situ (Sheu et al, 2022). Our approach expands this limited understanding by investigating the function of the ciliary pool of CDC42 during SHH signaling, using the ciliary-targeted FRET probe Raichu-CDC42. Our data demonstrate that the BBSome deficiency induces the CDC42 activity in cilia in the steady state and particularly during ciliary signaling to trigger ectocytosis-mediated cilia shortening.

## SHH signaling induces CDC42-dependent cilia shortening in BBSome-deficient cells

To observe whether the excess of the BBSome-dependent cargoes in cilia triggers the CDC42-mediated ectocytosis followed by cilia shortening in vivo, we employed the WT and $Bbs4^{KO/KO}$ MEFs overexpressing GPR161-mCherry (Fig. 4A). GPR161 is a BBSome-dependent cargo that has already been exogenously expressed to study the mechanism of ectocytosis (Nager et al, 2017). The GPR161-mCherry accumulated at the ciliary tips and with a frequency which was approximately twice higher than the endogenous GPR161 foci in both cell lines (Figs. 2C and 4A,B). SHH induction resulted in SMO import into cilia in both WT and $Bbs4^{KO/KO}$ MEFs, and increased the frequency of GPR161-mCherry foci specifically in $Bbs4^{KO/KO}$ MEFs, suggesting ongoing ectocytosis, unlike in WT MEFs (Fig. 4A,B). Inhibition of CDC42 increased the frequency of the GPR161 positive foci both in the steady state and SAG induced $Bbs4^{KO/KO}$ MEFs (Fig. 4A,B). Notably, we observed that the overexpressed GPR161 accumulated in foci also along the axoneme when CDC42 was inhibited over the course of SHH signaling (Fig. 4A).

We next monitored the ectocytosis of the GPR161-mCherry foci and its effect on the cilia length during the course of SHH activation in MEFs pre-treated for 30 min with the CDC42 inhibitor via time-lapse live cell imaging (Figs. 4C,E and EV2D,E). We observed that the ectocytosis of the GPR161-mCherry is initiated by local accumulation of GPR161-mCherry within foci at the ciliary tips in $Bbs4^{KO/KO}$ MEFs (Fig. 4C), and these foci are similar in size (~0.5 μm) to those observed through immunofluorescence staining (Fig. 1G). We detected slightly more ectocytosis events in GPR161-mCherry $Bbs4^{KO/KO}$ MEFs (Fig. 4D); but the statistical power is limited by the data scarcity.

In the $Bbs4^{KO/KO}$ MEFs, we observed that SHH signaling triggered cilia shortening (Figs. 4E and EV2D,E), which was blocked by the inhibition of CDC42 (Fig. 4C–E). In contrast, the activation of the SHH pathway did not lead to cilia shortening in GPR161-mCherry WT MEFs (Figs. 4E and EV2D,E), which differed from the results of the previous experiments (Figs. 2E and EV2A). However, the SHH-induced cilia shortening in WT cells has been inconsistently reported across various studies and is apparently sensitive to specific experimental conditions in particular assays (Ansari et al, 2024; Gomez et al, 2022). As we did not observe SHH-induced cilia shortening in the experiments with cells expressing exogenous GPR161 (Fig. 4E), the overloading of the cilia compartment with GPR161 might interfere with the ciliary cAMP homeostasis and the SHH-induced cilia shortening (Ansari et al, 2024). As this controversy concerning the cilia dynamics during the SHH-signaling in WT cells was not the aim of this study, future investigations are needed to come to its solution. Nevertheless, the stability of the cilia length in WT cells in this assay is advantageous, as it highlights the unique mechanism of SHH-induced cilia shortening in BBS conditions via CDC42-mediated ectocytosis. Overall, these data demonstrate that the signal-dependent shortening of primary cilia is specific to the BBS condition in the $Bbs4^{KO/KO}$ cells, occurs via ectocytosis, and is controlled by CDC42.

## CDC42 is required for actin polymerization inside the cilia

Our data indicated that the accumulation of signaling receptors activates CDC42 to trigger the ectocytosis. In the next step, we examined whether CDC42 induces actin polymerization inside the cilia. We expressed the membrane marker ARL13B-mNeonGreen (ARL13B-NG) and actin binding protein LifeAct-TagRFP in the WT and $Bbs4^{KO/KO}$ MEFs. We monitored the dynamics of the ciliary membrane and actin polymerization upon SHH activation in the absence or presence of ML141 using time-lapse imaging (Figs. 5A,B, EV3 and EV4, Movies EV1, EV2, EV3 and EV4). In several cases, we observed actin polymerization with following ectocytosis, which we defined as a visible separation of the ARL13B positive membrane segments (Fig. 5B, Movies EV1 and EV2).

We detected more actin polymerization events in the cilia of the $Bbs4^{KO/KO}$ cells in comparison to WT cells (Figs. 5C, EV3A and EV4A). The overall dynamics and duration of the F-actin patches was very variable (Fig. 5D). We observed that over the course of

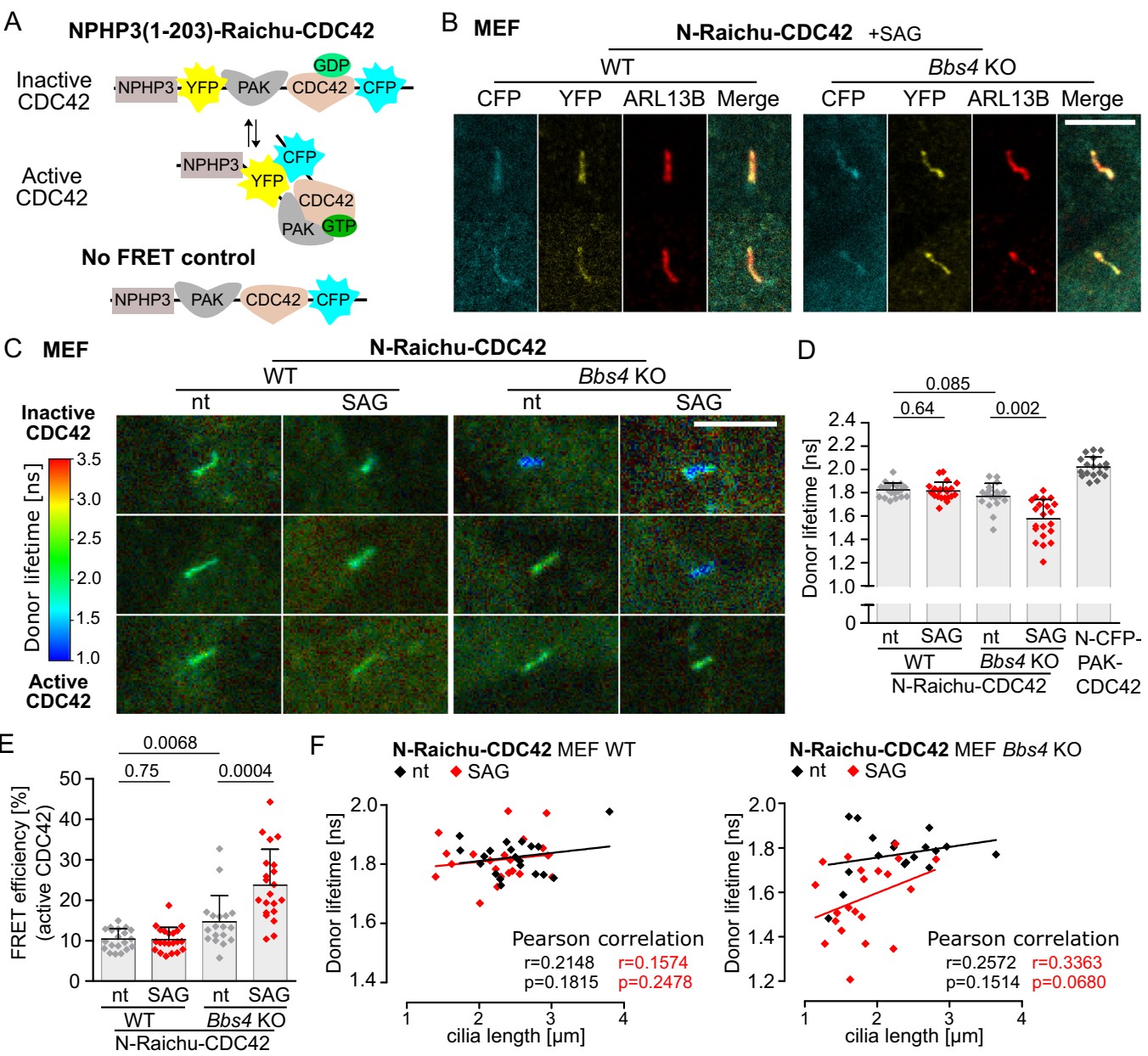

**Figure 3. CDC42 is hyperactivated in cilia in BBSome-deficient cells.**

(A) Schematic representation of the Raichu-CDC42 FRET probe with the ciliary targeting motif NPHP3 (1–203) and the no-FRET control lacking YFP. Activation of CDC42 is measured as decrease in FRET donor (CFP) lifetime. (B) Representative micrographs show cilia visualized by staining with antibody to ARL13B and the cellular localization of N-Raichu-CDC42 probe (YFP and CFP) expressed in WT and Bbs4$^{KO/KO}$ MEFs induced with SAG for 2 h. Cilia base at bottom. Scale bar, 5 µm. (C) Representative micrographs show donor lifetime values in non-treated (nt) and SAG induced—2 h, WT and Bbs4$^{KO/KO}$ MEFs expressing N-Raichu-CDC42 probe determined by the FLIM-FRET analysis in fixed cells. Lifetimes are shown in pseudocolours ranging from blue to red. Three cilia are showed per condition for better illustration. Scale bar, 5 µm. (D) The donor lifetime values extracted from the FRET-FLIM analysis of cilia as a region of interest in non-treated (nt) and SAG induced—2 h, WT and Bbs4$^{KO/KO}$ MEFs expressing N-Raichu-CDC42 and values measured for the no-FRET control expressed in WT MEFs. Mean and SD of three (WT) and four (KO, no-FRET) independent experiments (n = 18–21 cilia). (E) The FRET efficiency extracted from the FRET-FLIM analysis of cilia as a region of interest in non-treated (nt) and SAG induced—2 h, WT and Bbs4$^{KO/KO}$ MEFs expressing N-Raichu-CDC42. Mean and SD of three (WT) and four (KO) independent experiments (n = 18–21 cilia). (F) Correlation plots of the simultaneously analyzed cilia length and the donor lifetimes in non-treated (nt) and SAG induced—2 h, WT and Bbs4$^{KO/KO}$ MEFs expressing N-Raichu-CDC42 in (D). Linear regression is indicated by the solid line, with the Pearson correlation coefficient r (moderate positive correlation 0.3–0.5, weak positive correlation 0.1–0.3) and p value for each condition; nt (black), SAG (red), WT—left plot, KO—right plot (n = 18–21 cilia). Data information: Statistical significance was calculated using the two-tailed Mann–Whitney test (D, E) and one-tailed Pearson correlation (F) and the obtained p-values are indicated. Source data are available online for this figure.

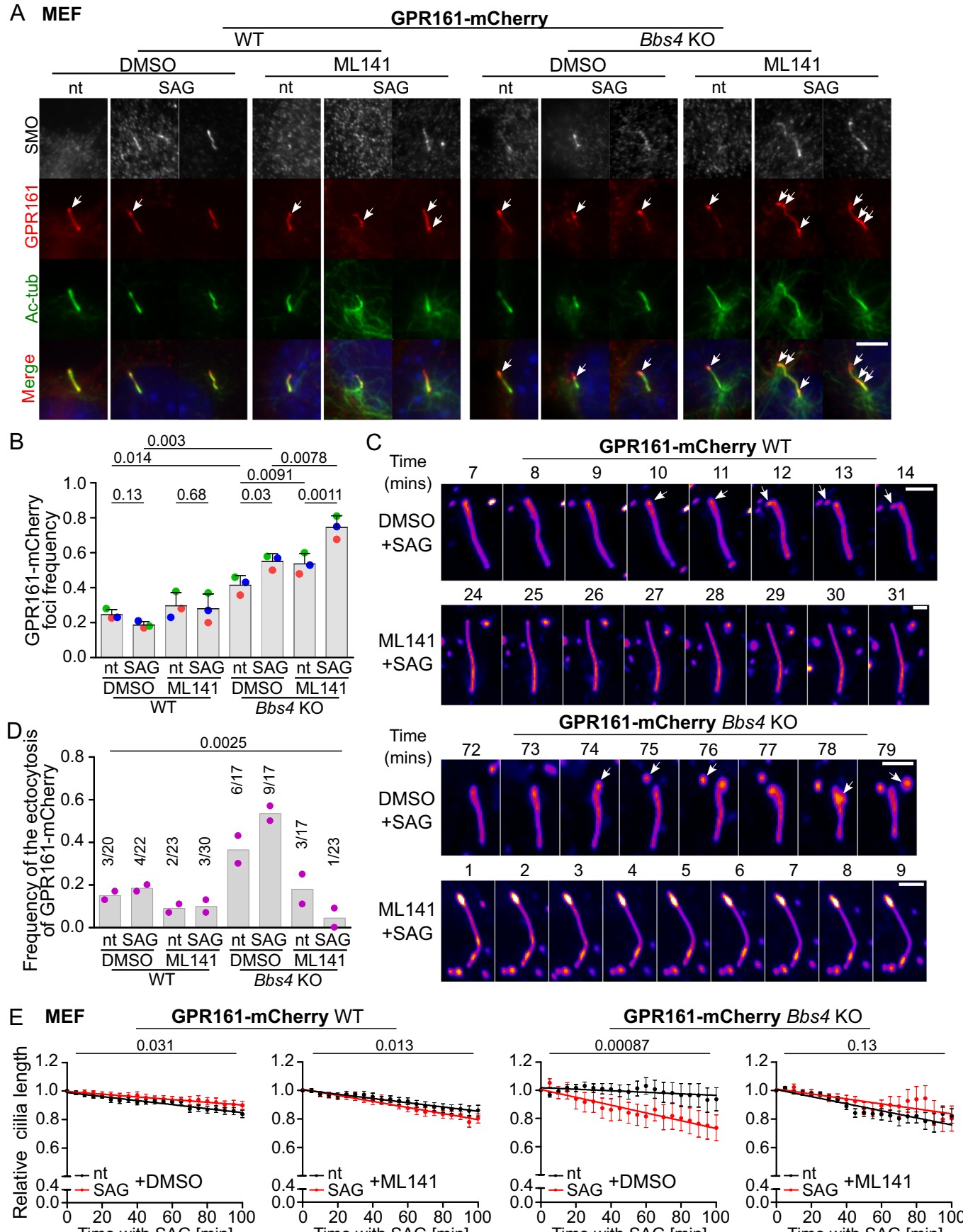

Figure 4. SHH signaling induces CDC42-dependent cilia shortening in BBSome-deficient cells.

(A, B) Representative micrographs (A) and quantification of the frequency (B) of GPR161-mCherry foci (arrows) in cilia visualized by staining with antibody to acetylated tubulin (Ac-tub) in non-treated (nt) and SAG induced—2 h, WT and *Bbs4*[KO/KO] MEFs concomitantly treated with vehicle or ML141. Staining with antibody to SMO was employed to visualize the activation of the SHH pathway. Cilia base at bottom. Scale bar, 5 μm. Mean and SD of three independent experiments (*n* = 100–160 cilia). Statistical significance was calculated using the two-tailed paired t-test. (C) Representative micrographs depicting ciliary ectocytosis detected by live cell imaging of WT and *Bbs4*[KO/KO] MEFs expressing mCherry-GPR161 and induced with SAG for 2 h. The cells were pre-treated with DMSO (top) or ML141 (bottom) for 30 min and then imaged every 1 min for 100 min. White arrows point to the foci formation, ectocytosis and ectosomes. Cilia base at bottom. Scale bar, 2 μm. (D) The graph shows the frequency of cilia with ectocytosis events detected by live cell imaging of WT and *Bbs4*[KO/KO] MEFs expressing mCherry-GPR161 non-treated or upon 2 h treatment with SAG and pre-treated with DMSO or ML141 before acquisition. The number of observed ectocytosis out of the total count of imaged cilia from two independent live cell imaging experiments is shown. Statistical analysis was done using the contingency table and Chi-square test. (E) Plots depict the average dynamics of the cilia length in non-treated (nt) and SAG induced—2 h, WT and *Bbs4*[KO/KO] MEFs expressing GPR161-mCherry pre-treated with DMSO or ML141 for 30 min and then imaged for 100 min. The length of the cilium was normalized to the length measured at time 0 min. In total, 17–30 cilia were monitored per condition in two independent live cell imaging experiments. The data points (Mean ± SEM) were fitted using the linear regression and the statistical significance (two-tailed) of the slope of the regression lines was calculated for the indicated conditions. Data information: Statistical significance was calculated using the two-tailed paired t-test (B), Chi-square test (D), and two-tailed F-test for the difference between two slopes (E) and the obtained *p*-values are indicated. Source data are available online for this figure.

SHH activation, the inhibition of CDC42 reduces the frequency of the ciliary actin polymerization events (Figs. 5C, EV3B and EV4B, Movies EV3 and EV4) and their duration (Fig. 5D). To confirm the specific role of the intraciliary pool of CDC42 in actin polymerization within cilia, we used cilia-targeted CDC42-WT and CDC42-DN variants. We observed that cilia-targeted CDC42-DN reduced the frequency of ciliary actin polymerization events compared to CDC42-WT in *Bbs4*[KO/KO] MEFs (Figs. 5E and EV5A,B). In some instances, we observed growing F-actin patches prior to ectocytosis. This suggests that actin microfilaments could exert pushing forces to release the ectosomes from cilia (Corral-Serrano et al, 2020; Footer et al, 2007). Accordingly, the actin polymerization events were rare and only short-lived upon the CDC42 inhibition, indicating that CDC42 induces and sustains ciliary F-actin structures within cilia to trigger ectocytosis.

Overall, we showed in this study that the loss of the retrograde cargo adaptor BBSome leads to the hyperactivation of CDC42 within cilia, both in the steady state and during the SHH signaling. Hyperactive CDC42 triggers excessive F-actin mediated ectocytosis, which promotes cilia shortening, the typical phenotype linked to BBS (Chiuso et al, 2023; Hernandez-Hernandez et al, 2013; Prasai et al, 2020; Uytingco et al, 2019). Cilia length is regulated also by other factors, including PIP(2) levels, EP4 signaling, and endocytosis (Ansari et al, 2024; Patnaik et al, 2019; Saito et al, 2017; Stilling et al, 2022). The relationship between these regulatory mechanisms and their contribution to the phenotypes associated with BBS remains to be fully elucidated.

The signaling pathways triggered by some plasma membrane-localized GPCRs lead to the activation of the RHO GTPases and consequent actin remodeling on the whole cell level (Muller et al, 2020). A similar mechanism could also control actin polymerization within cilia. Activation of ciliary signaling pathways, e.g., SHH, PDGFRα, leads to a conformational change and ubiquitination of the GPCRs destined for ciliary exit (Schmid et al, 2018; Shinde et al, 2023; Shinde et al, 2020). When SMO enters the cilium, it enhances β-arrestin recruitment to GPR161 as an export signal from cilium (Pal et al, 2016). While the exact mechanism by which SMO enables GPR161 removal is unclear, it has been suggested that SMO and GPR161 may exit the cilium together as a bipartite receptor complex (Pal et al, 2016). In BBSome-deficient cells, while SMO can enter the cilium, it cannot be exported, similarly to GPR161 (Fig. 4A) (Nozaki et al, 2018). Given that ectocytosis is triggered by activated receptors unable to exit the cilium (Nager et al, 2017), it might be the cilia-

contained active SMO actually triggering the ectocytosis, leading to a concomitant removal of GPR161 within the bipartite complex.

Furthermore, GPR161 has constitutive activity and couples to Gα$_s$, which increases cAMP levels and PKA activity to represses SHH transduction (Mukhopadhyay et al, 2013). In the cytoplasm, this Gα$_s$ subunit of GPCRs activates PDZ-RhoGEF, a major CDC42 activator (Castillo-Kauil et al, 2020). PDZ-RhoGEF and other CDC42 activators have been detected in the ciliary proteomes (Kohli et al, 2017; Mick et al, 2015; van Dam et al, 2019). Therefore, the substantial accumulation of the ubiquitinated GPR161 within cilia in the BBSome-deficient cells might lead to aberrant triggering of the RHOGEFs and CDC42 through the activated Gα$_s$ proteins (Castillo-Kauil et al, 2020; Mukhopadhyay et al, 2013), eventually leading to ectocytosis. This scenario is supported by our findings that GPR161 signaling increases CDC42 activity in the cilia of BBSome-deficient cells but not WT cells, indicating a direct link between the GPCR signaling and CDC42 activation. Furthermore, this aligns with our and recent observation that mere overexpression of GPCRs is insufficient to activate ectocytosis (Nager et al, 2017).

Although the molecular etiology of particular BBS symptoms in tissues is largely unexplained, it is plausibly linked to defects in specific signaling pathways (Novas et al, 2015). The BBSome deficiency presents with short cilia and mislocalization of the BBSome-dependent cargoes, mostly G-protein coupled receptors (GPCRs). In cells lacking the BBSome, several GPCRs such as GPR161, GPR19, and D1 accumulate in the cilia (Domire et al, 2011; Nozaki et al, 2018; Stubbs et al, 2022), which corresponds to the well-described role of the BBSome in the retrograde transport (Lechtreck et al, 2013; Liu and Lechtreck, 2018; Nager et al, 2017; Ye et al, 2018). However, some other receptors disappear from cilia in the BBSome-deficient cells and tissues, such as SSTR3, MCHR1, and NPY2 in neurons of mouse models of BBS (Berbari et al, 2008; Loktev and Jackson, 2013; Stubbs et al, 2022). This was originally explained by the proposed role of the BBSome in the import of these receptors into the cilia (Jin et al, 2010). However, the BBSome-mediated ciliary cargo import has never been clearly documented on a molecular level. It is thus possible that the absence of these receptors from the cilia can be caused by the BBSome-deficiency indirectly via the enhanced ectocytosis which might remove these receptors in a by-stander manner. In this scenario, the CDC42-mediated ectocytosis in BBSome-deficient cells would be a pathological mechanism connected with the BBS pathology.

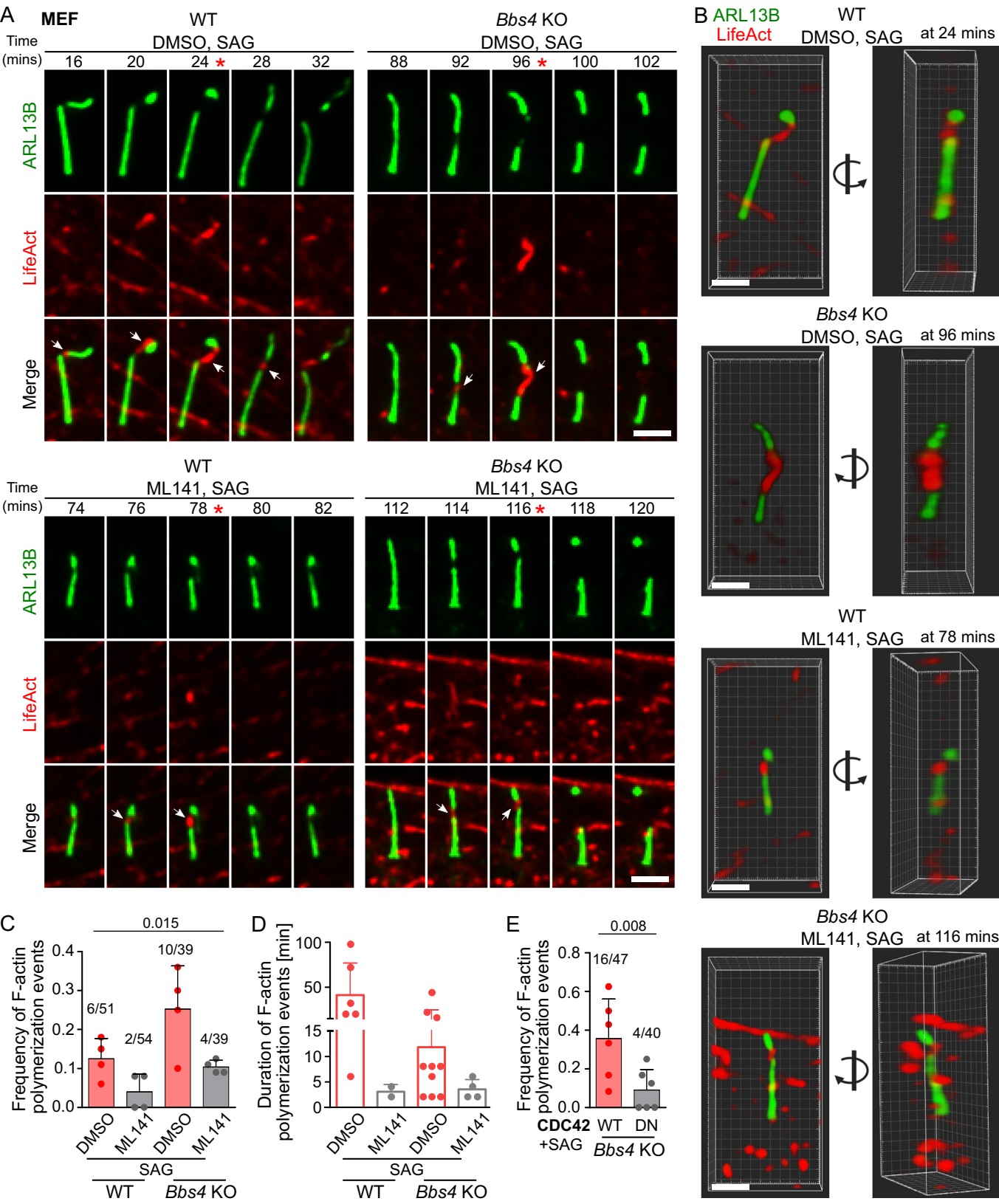

**Figure 5. CDC42 is required for actin polymerization inside the cilia.**

(A) Representative micrographs depicting ciliary ectocytosis and actin polymerization detected by live cell imaging of WT and *Bbs4*[KO/KO] MEFs expressing mNG-ARL13B and LifeAct-TagRFP. The cells were treated with SAG and DMSO (top) or ML141 (bottom) and imaged every 2 min for 2 h. White arrows point to the F-actin polymerization events. Red asterisks indicate the frames used for 3D visualization in (B). Cilia base at bottom. Scale bar, 2 μm. Maximum intensity projections of the z-stacks were done using Fiji ImageJ software and the intensities for both channels were adjusted post acquisition for better visualization. (B) 3D visualization of the F-actin patches and ciliary membrane in the WT and *Bbs4*[KO/KO] MEFs at the indicated time points of the live cell imaging in (A). Left micrographs show the front view of the cilium and right micrographs show a 90° rotated view in the direction indicated by the arrows. Scale bar, 2 μm. (C) The graph shows the fraction of cilia with actin polymerization events observed in WT and *Bbs4*[KO/KO] MEFs expressing mNG-ARL13B and LifeAct-TagRFP upon treatment with SAG and vehicle or ML141 during the live cell imaging in (A). The number of observed actin polymerization events out of the total count of imaged cilia from four independent live cell imaging experiments and mean with SD are shown. Statistical analysis was done using the contingency table and Chi-square test. (D) The plot depicts the duration of the observed actin polymerization events detected in cilia in the WT and *Bbs4*[KO/KO] MEFs expressing mNG-ARL13B and LifeAct-TagRFP upon treatment with SAG and vehicle or ML141 during the live cell imaging in (A). Means with SD are shown. (E) The graph shows the fraction of cilia with actin polymerization events observed in *Bbs4*[KO/KO] MEFs expressing mNG-ARL13B and LifeAct-TagRFP in the presence of WT or DN version of CDC42 upon treatment with SAG during the live cell imaging in (Fig. EV5). The number of observed actin polymerization events out of the total count of imaged cilia from six independent live cell imaging experiments and mean with SD are shown. Statistical analysis was done using the contingency table and two-sided Chi-square test. Data information: Statistical significance was calculated using the Chi-square test (C) and two-sided Chi-square test (E) and the obtained *p*-values are indicated. Source data are available online for this figure.

BBSome-deficient photoreceptors accumulate mislocalized proteins in their outer segments, which is a highly specialized cilium (Datta et al, 2015; Dilan et al, 2018; Masek et al, 2022). As the formation of the photoreceptor discs resembles modified ectocytosis (Spencer et al, 2020), it is possible that the BBSome deficiency interferes with this process, which leads to the dysfunction of photoreceptors and eventualy causes retinal dystrophy, one of the hallmarks of the BBS (Dilan et al, 2018; Niederlova et al, 2019).

The BBSome-deficiency leads to morphological changes also in motile cilia in the brain ependymal layer and in the airways (Shah et al, 2008; Zhang et al, 2011; Zhang et al, 2013), which are responsible for the flow of cerebrospinal fluid and the transport of mucus and protection from respiratory infections, respectively. The motile cilia of several examined *Bbs* KO mice formed bulges at the ciliary tip filled with vesicles, which was accompanied by altered cilia beating (Shah et al, 2008; Zhang et al, 2011). It is possible that the CDC42-mediated ectocytosis or a related process are triggered in the motile cilia upon the BBSome dysfunction.

Altogether, enhanced and altered ectocytosis in BBSome-deficient cells could contribute to the manifestation of BBS symptoms. Thus, the inhibition of the increased ectocytosis could be a novel potential therapeutic strategy in the BBS.

## Methods

### Reagents and tools table

| Reagent/Resource | Reference or Source | Identifier or Catalog Number |
|---|---|---|
| **Experimental models** | | |
| hTERT-RPE1 cells (*H. sapiens*) | Provided by Dr. Vladimir Varga, IMG CAS, Czech Republic | ATCC-CRL-4000 |
| *BBS4*[KO/KO] hTERT-RPE1 cells | Prasai et al, 2020 | |
| *BBS1*[KO/KO] hTERT-RPE1 cells | Prasai et al, 2020 | |
| *BBS9*[KO/KO] hTERT-RPE1 cells | Prasai et al, 2020 | |
| *BBS7*[KO/KO] hTERT- RPE1 cells | Prasai et al, 2020 | |

| Reagent/Resource | Reference or Source | Identifier or Catalog Number |
|---|---|---|
| Platinum Eco cells | Provided by Dr. Tomas Brdicka, IMG CAS, Czech Republic | |
| MEF cells (*M. musculus*) | Tsyklauri et al, 2021 | |
| ST2 cells (*M. musculus*) | Tsyklauri et al, 2021 | |
| **Recombinant DNA** | | |
| pMSCV-IRES-Thy 1.1 | Clontech | 17442 |
| CDC42 WT plasmid | Addgene, Subauste et al, 2000 | 12599 |
| CDC42 DN plasmid | Addgene, Subauste et al, 2000 | 12601 |
| CDC42 Raichu probe | Provided by Prof. Michiyuki Matsuda, Kyoto University Graduate School of Medicine, Japan. Yoshizaki et al, 2003 | |
| pLV-LifeAct-TagRFP | Provided by Dr. Zdenek Hodny, IMG CAS, Czech Republic | |
| pMSCV-ARL13b-mNeonGreen-puro | Provided by Dr. Vladimir Varga, IMG CAS, Czech Republic | |
| pSpCas9(BB)-2A-GFP (PX458) | Provided by Feng Zhang. Addgene | 48138 |
| pCMV-mNeonGreen-IFT88 | Provided by Dr. Raman Das (The University of Manchester). Toro-Tapia and Das, 2020 | |
| pMSCV-NPHP3(1-203)-Raichu-Cdc42 | This study | |
| pMSCV-NPHP3(1-203)-PAK-CFP-Cdc42 | This study | |
| pMSCV-GPR161-mCherry | This study | |
| pMSCV-NPHP3(1-203)-GFP-CDC42-WT | This study | |
| pMSCV-NPHP3(1-203)-GFP-CDC42-DN | This study | |
| pMSCV-IFT88-mNeonGreen | This study | |
| **Antibodies** | | |
| Mouse anti-acetylated tubulin | Provided by Dr. Vladimir Varga, IMG CAS, Czech Republic | |

| Reagent/Resource | Reference or Source | Identifier or Catalog Number |
|---|---|---|
| Rabbit anti-GPR161 | Provided by Dr. Saikat Mukhopadhyay, UT Southwestern Medical Center, USA | |
| Rabbit anti-ARL13B | Proteintech | 17711-1-AP |
| Rabbit anti-IFT88 | Proteintech | 13967-1-AP |
| Rabbit anti-mCherry | Thermofisher | PA534974 |
| Mouse anti-TSG101 | GeneTex | GTX70255 |
| Rat anti-CD9 | Santa-Cruz | sc-18869 |
| Mouse anti-SMO (E-5) | Santa-Cruz | sc-166685 |
| Rabbit anti-calnexin | Proteintech | 10427-2-AP |
| Mouse β-Actin | Sigma-Aldrich | A1978 |
| Anti-mouse Alexa Fluor 488 | Invitrogen | A11001 |
| Anti-rabbit Alexa Fluor 488 | Invitrogen | A11008 |
| Anti-mouse Alexa Fluor 555 | Invitrogen | A21422 |
| Anti-rabbit Alexa Fluor 555 | Invitrogen | A21428 |
| Anti-mouse Alexa Fluor 647 | Invitrogen | A21235 |
| Anti-rabbit Alexa Fluor 647 | Invitrogen | A21245 |
| Goat anti-mouse-HRP | Jackson | 115-035-14 |
| Goat anti-rabbit-HRP | Jackson | 111-035-144 |
| Goat anti-rat-HRP | Jackson | 112-035-003 |
| **Oligonucleotides and other sequence-based reagents** | | |
| *Ift88* sgRNA | This study | |
| *Ift88* KO sequencing primers | This study | |
| **Chemicals, Enzymes and other reagents** | | |
| Phalloidin Texas Red | Invitrogen | T7471 |
| ProLong™ Gold Antifade Mountant | Invitrogen | P36930 |
| ProLong™ Gold Antifade Mountant with DNA Stain DAPI | Invitrogen | P36941 |
| Dulbecco's modified Eagle's medium | Sigma | D6429 |
| Fetal bovine serum | Gibco | 10270-106 |
| Penicillin | BB Pharma | 15/156/69-A/C |
| Streptomycin | Sigma-Aldrich | 59137 |
| Gentamicin | Sandoz | |
| SAG | Sigma-Aldrich | 566660 |
| ML141 | Tocris Biosciences | 4266 |
| Y27632 | Sigma-Aldrich | Y0503 |
| RAC1 inhibitor | Tocris Biosciences | CAS1177865-17-6 |
| Cytochalasin D | Sigma-Aldrich | C8273 |
| CK666 | Sigma-Aldrich | 182515 |
| Goat serum | Sigma | G6767 |
| FluoroBrite DMEM | Gibco | A1896701 |

| Reagent/Resource | Reference or Source | Identifier or Catalog Number |
|---|---|---|
| Formaldehyde solution | Sigma-Aldrich | F8775 |
| Protease inhibitor mixture | Roche Applied Science | 5056489001 |
| Lipofectamine 2000 | Invitrogen | 52887 |
| **Software** | | |
| GraphPad Prism 5.0 | https://www.graphpad.com | |
| Fiji ImageJ | https://imagej.net/ij/ | |
| Imaris Image Analysis Software 9.9.1 | Bitplane, Oxford Instruments plc | |
| Huygens Professional v. 21.04 | Scientific Volume Imaging, The Netherlands | |
| **Other** | | |
| Thermo Scientific Heraeus Megafuge 8 centrifuge | Thermo Scientific | |
| SW32Ti rotor | Beckman Coulter | |
| Pierce™ BCA Protein Assay Kit | Thermo Scientific | |
| Vilber Fusion Solo S imaging system | Vilber | |
| Delta Vision Core microscope using the oil immersion objective (Plan-Apochromat 60× NA 1.42) | Delta Vision Core | |
| Leica Stellaris 8 Falcon confocal microscope | Leica | |
| CHOPCHOP | https://chopchop.cbu.uib.no Labun et al, 2019 | |
| FACSAria Ilu | BD Biosciences | |

## Antibodies, dyes, and reagents

Mouse anti-acetylated tubulin (IF, 1:50) was kindly provided by Dr. Vladimir Varga (Institute of Molecular Genetics of the Czech Academy of Sciences), rabbit anti-GPR161 (IF, 1:200) was kindly provided by Dr. Saikat Mukhopadhyay (UT Southwestern Medical Center). Rabbit anti-ARL13B (17711-1-AP; IF, 1:2000), rabbit anti-IFT88 (13967-1-AP; WB, 1:1000) and rabbit anti-Calnexin (10427-2-AP, WB 1:1000) were purchased from Proteintech. Rabbit anti-mCherry (PA534974; WB, 1:1000) was purchased from Thermofisher. Mouse anti-TSG101 [4A10] (GTX70255; WB, 1:1000) was purchased from GeneTex. Rat anti-CD9 (sc-18869, WB, 1:250) and mouse anti-SMO (sc-166685, IF, 1:50) was purchased from Santa-Cruz. Mouse anti-β-Actin (A1978, WB 1:5000) was purchased from Sigma-Aldrich.

Secondary antibodies for IF were as follows: anti-mouse Alexa Fluor 488 (Invitrogen, A11001; 1:1000), anti-rabbit Alexa Fluor 488 (Invitrogen, A11008; 1:1000), anti-mouse Alexa Fluor 555 (Invitrogen, A21422; 1:1000), anti-rabbit Alexa Fluor 555 (Invitrogen, A21428; 1:1000), anti-mouse Alexa Fluor 647 (Invitrogen, A21235; 1:1000) and anti-rabbit Alexa Fluor 647 (Invitrogen, A21245; 1:1000). Secondary antibodies for WB were as follows: goat anti-mouse-HRP (Jackson, 115-035-14, 1:1000), goat anti-rabbit-HRP (Jackson, 111-035-144, 1:1000) and goat anti-rat-HRP (Jackson, 112-035-003, 1:1000).

Phalloidin Texas Red (T7471; 1:250) and ProLong™ Gold antifade reagent with 4′,6-diamidino-2-phenylindole (DAPI) (P36941) or without (P36930) were purchased from Invitrogen.

## Cell cultures and treatments

Immortalized human retinal pigment epithelium cell line, hTERT-RPE1 (RPE1) (ATCC, CRL-4000) was kindly provided by Dr. Vladimir Varga (Institute of Molecular Genetics of the Czech Academy of Sciences). BBS4$^{KO/KO}$, BBS1$^{KO/KO}$, BBS9$^{KO/KO}$, and BBS7$^{KO/KO}$ RPE1 cell lines were established previously (Prasai et al, 2020). Phoenix Eco cells were kindly provided by Dr. Tomas Brdicka (Institute of Molecular Genetics of the Czech Academy of Sciences). Mouse embryonic fibroblast (MEFs) lines and mouse bone marrow derived mesenchymal ST2 cells were established previously (Tsyklauri et al, 2021). All cell lines were cultured in complete Dulbecco's modified Eagle's medium (DMEM, Sigma, D6429 - 500 mL) supplemented with 10% fetal bovine serum (FBS), 100 U/mL penicillin (BB Pharma), 100 µg/mL streptomycin (Sigma-Aldrich), and 40 µg/mL gentamicin (Sandoz). All cell lines are regularly tested for mycoplasma contamination.

Smoothened agonist—SAG (566660; Sigma-Aldrich) was used at a concentration of 200 nM. CDC42 inhibitor—ML141 (4266; Tocris Biosciences) was used at 50 µM for IF and at 20 µM for EV preparations, ROCK-1 inhibitor—Y27632 (Y0503; Sigma-Aldrich) was used at 20 µM, RAC1 inhibitor (CAS1177865-17-6; Tocris Biosciences) was used at 100 µM, Cytochalasin D (C8273; Sigma-Aldrich) was used at 1 µM and ARP 2/3 complex inhibitor—CK666 (182515; Sigma-Aldrich) was used at 100 µM concentration. The treatments were for 2 h if not indicated otherwise.

## Cloning, gene transfections, and deletion

GPR161 ORFs was amplified from cDNA obtained from RPE1 cells, appended with mCherry coding sequence at the C terminus using recombinant PCR, and cloned into pMSCV-IRES-Thy 1.1 vector (Clontech) using XhoI and ClaI restriction sites. CDC42 WT (12599) and DN (12601) plasmids were obtained from Addgene (Subauste et al, 2000). Ciliary targeting motif NPHP3 [1–203] (Mick et al, 2015) was fused to GFP-CDC42, WT and DN, N-terminally and sub-cloned into pMSCV-IRES-Thy 1.1 vector (Clontech) using EcoRI and ClaI restriction sites.

CDC42 Raichu probe was a kind gift from Prof. Michiyuki Matsuda (Kyoto University Graduate School of Medicine) (Yoshizaki et al, 2003). Raichu probe was fused N-terminally with the ciliary targeting motif and sub-cloned into pMSCV-IRES-Thy 1.1 vector using EcoRI and ClaI restriction sites. CFP-PAK-CDC42 was amplified from the CDC42 Raichu probe and N-terminally tagged with the ciliary targeting motif and sub-cloned into pMSCV-IRES-Thy 1.1 using EcoRI and ClaI restriction sites.

LifeAct-TagRFP was a kind gift from Dr. Zdenek Hodny (Institute of Molecular Genetics of the Czech Academy of Sciences). mNeonGreen-ARL13B was a kind gift from Dr. Vladimir Varga (Institute of Molecular Genetics of the Czech Academy of Sciences).

Ift88 ORF was amplified from pCMV-mNG-IFT88 kindly provided by Dr. Raman Das (The University of Manchester, UK) (Toro-Tapia and Das, 2020), appended with the mNeonGreen coding sequence at C terminus using recombinant PCR, and cloned

into pMSCV-Thy-IRES 1.1 vector (Clontech) using EcoRI and ClaI restriction sites.

The viral transduction of cell lines was done according to (Bino et al, 2022) with slight modifications. For viral particle production, Platinum Eco cells were seeded on 10 cm dish and allowed to reach confluency of 60–70%. 30 µg of plasmid DNA was transfected into cells using polyethyleneimine to generate retroviruses. Cells were incubated overnight and the media was changed to production media (DMEM + ATB + FBS) on the next day. MEFs and ST2 cells were transduced with 2 mL supernatant containing the viral particles along with 8 µg/mL polybrene and analyzed for transgene expression after 48 h. MEFs expressing CFP, YFP, mNeonGreen, GFP, mCherry, and/or TagRFP and ST2 cells expressing mNeonGreen were bulk sorted using a FACSAria IIu (BD Biosciences).

Ift88 knockout cell lines were generated using the CRISPR/Cas9 approach as done previously (Prasai et al, 2020). Single-guided RNA (sgRNA) targeting Ift88 gene was designed using the web tool CHOPCHOP (Labun et al, 2019). sgRNA was cloned into pSpCas9(BB)-2A-GFP (PX458) vector kindly provided by Feng Zhang (Addgene plasmid 48138) (Ran et al, 2013). sgRNA sequence with PAM motif (3′ end) for mouse Ift88 is: TCAATGGGAAGACCGATGACAGG.

ST2 cells were transfected with PX458 vector with specific Ift88 sgRNA using polyethyleneimine (Polysciences, Inc., 23966-2). After 48 h, cells expressing GFP were sorted as single cells in 96-well plates using the 488-nm laser on FACSAria IIu (BD Biosciences). Obtained clones were tested via immunoblotting for expression of IFT88 and confirmed by sequencing (Ift88 F: TGGATTGTTACTTGCC-TACCCT, Ift88 R: CAGGCAGAAGTAAACCACAGG).

## Preparation of cell lysates and extracellular vesicles and quantification

MEF cells expressing the GPR161-mCherry were cultured to near confluency in a two 15 cm dish per condition. The cells were starved overnight in pure DMEM. The next day the cells were washed with PBS, and incubated for 2.5 h in pure DMEM medium containing 200 nM SAG, and 20 µM ML141 or DMSO as vehicle. The medium was harvested and transferred to a 50 mL conical tube. The cells were washed with 10 mL PBS, and the wash was added to the collected medium. The medium was first centrifuged at $300 \times g$ for 10 min at 4 °C. The supernatant was transferred to a new 50 mL conical tube and centrifuged at $2000 \times g$ for 20 min at 4 °C. The resulting supernatant was transferred to a new 50 mL conical tube and centrifuged using fixed angle rotor in Thermo Scientific Heraeus Megafuge 8 centrifuged at $10,000 \times g$ for 40 min at 4 °C to produce the P10 pelet. The supernatant was then ultracentrifuged at $100,000 \times g$ for 90 min at 4 °C using the SW32Ti rotor. The pelet was washed in PBS and ultracentrifuged at $100,000 \times g$ for 90 min using the same rotor to produce the P100 pelet. The P100 pelet was lysed in the same volume of the Laemli buffer (~170 µL) and incubated for 10–15 min at 65 °C.

Alongside, the MEF cells were collected and lysed in lysis buffer (20 mM HEPES, pH 7.5, 150 mM NaCl, 2 mM EDTA, pH 8, 0.5% Triton X-100) supplemented with protease inhibitor mixture (complete, Roche Applied Science, catalog no. 05056489001). Total cell lysates (TCL) were cleared by centrifugation at $15,000 \times g$ for 15 min at 4 °C. Protein concentration was measured using the

Pierce™ BCA Protein Assay Kit (Thermo Scientific). The loading of the TCL and EV samples for SDS-PAGE (1.5 mm gels) was normalized to the TCL with the lowest concentration. The Western blotting was done according to (Prasai et al, 2020). The membranes were probed with the primary antibodies to mCherry, IFT88, TSG101, CD9, and Calnexin overnight and the next day incubated with the secondary antibodies coupled to HRP and developed using the Vilber Fusion Solo S imaging system (Vilber). The fold change values of the EV fractions of IFT88 and GRP161-mCherry were calculated using ImageJ as follows: ratio of (EV – background):(TCL – background). The fold change was normalized to the WT SAG DMSO value.

To assess the specificity of the IFT88 antibody, total cell lysates were prepared from WT MEFs, ST2 WT cells, ST2 *Ift88* KO cells, and their IFT88-mNeonGreen expressing counterparts, as described above. The Western blotting was done according to (Prasai et al, 2020). The membranes were probed with the primary antibodies to IFT88 (overnight) and β-Actin (1 h) and then incubated with the secondary antibodies coupled to HRP and developed using the Vilber Fusion Solo S imaging system (Vilber).

## Immunofluorescence

RPE1 or MEF cells were seeded on 12-mm coverslips and serum starved for 24 h. After respective treatments, cells were fixed (4% formaldehyde) and permeabilized (0.2% Triton X-100) for 10 min. Blocking was done using 5% goat serum (Sigma, G6767-100 mL) in PBS for 15 min and incubated with primary antibody (1% goat serum/PBS) and secondary antibody (PBS) for 1 h and 45 min, respectively, in a wet chamber. The cells were washed after each step in PBS three times. At last, the cells were washed in distilled H₂O, air-dried, and mounted using ProLong™ Gold antifade reagent with DAPI (P36941; Invitrogen).

## Fluorescence microscopy

Image acquisition was performed on the Delta Vision Core microscope using the oil immersion objective (Plan-Apochromat 60× NA 1.42) and filters for DAPI (435/48), FITC (523/36), TRITC (576/89), and Cy5 (632/22). Z-stacks were acquired at 1024 × 1024-pixel format and Z-steps of 0.2 μm. Z-stacks were analyzed using Fiji ImageJ software. Maximum intensity projections were used to quantify the frequency of ARL13B and GPR161 foci and GPR161 positive cilia and to measure the cilia length in the Fiji ImageJ software. The cilium length was measured with the line tool in Fiji ImageJ using either the single channel for acetylated tubulin (specifically Figs. 1B,E,F and EV1D,E), for ARL13B (Figs. 1J, EV1G and EV2A) or the merged channels for GPR161 with acetylated tubulin (Fig. 2E,G).

## Expansion microscopy

Expansion microscopy of primary cilia was done as described previously (Prasai et al, 2020). RPE1 cells were cultured on 12 mm coverslips and serum starved for 24 h. Coverslips were fixed with 4% formaldehyde/4% acrylamide in PBS overnight and then washed 2× with PBS. The gelation was performed by incubating coverslips face down with 45 μL of monomer solution (19% (W/W) sodium acrylate, 10% (W/W) acrylamide, 0.1% (W/W) N,N

′-methylenbisacrylamide in PBS supplemented with 0.5% TEMED and 0.5% APS), in a pre-cooled humid chamber. After 1 min on ice, chamber was incubated at 37 °C in the dark for 30 min. Samples in the gel were denatured in denaturation buffer (200 mM SDS, 200 mM NaCl, 50 mM Tris in ddH₂O) at 95 °C for 4 h. Gels were expanded in ddH₂O for 1 h and then cut into 1 × 1 cm pieces. Pieces of gel were incubated with primary antibodies diluted in 2% BSA in PBS overnight at RT. After staining, shrunk pieces of gel were incubated in ddH₂O for 1 h. After expansion, pieces of gel were incubated with secondary antibodies diluted in 2% BSA in PBS for 3 h at RT. Last expansion in ddH₂O with exchange every 20 min was for 1 h until pieces of gel reached full size. Samples were imaged in 35 mm glass bottom dishes (CellVis) pre-coated with poly-L-lysine. During imaging, gels were covered with ddH₂O to prevent shrinking. Expanded cells were imaged by confocal microscopy on Leica TCS SP8 using a 63× 1.4 NA oil objective with closed pinhole to 0.4 AU. Cilia images were acquired in Z-stacks at 0.1 μm stack size with pixel size 30–50 nm according to the cilia length. Images were de-convolved using Huygens Professional v. 21.04 software (Scientific Volume Imaging, Hilversum, Netherlands).

## Live cell imaging, data processing, and analysis

MEF cells expressing GPR161-mCherry or mNG-ARL13B and LifeAct-TagRFP were cultured on glass bottom dish (CellVis) until confluent. Cells were serum starved for 24 h for the induction of ciliogenesis. Before imaging, cells were treated with SAG and DMSO or CDC42 inhibitor, ML141 in FluoroBrite DMEM (Gibco, A1896701) and imaged immediately using Delta Vision Core microscope at 37 °C with 5% CO₂. Images were acquired using the oil immersion objective (Plan-Apochromat ×60, NA 1.42) in 1024 × 1024-pixel format and Z-steps of 0.2 μm. 7–8 imaging positions containing cilia were set.

Imaging of GPR161-mCherry expressing cells was performed using the filter for mCherry (632/60) for 100 min in 1 min intervals to acquire a time-lapse video (Fig. 4C). Imaging of mNG-ARL13B and LifeAct-TagRFP expressing cells was performed using filters for FITC (523/36) and TRITC (576/89) for 120 min in 2 min intervals to acquire a time-lapse video (Figs. 5A, EV3, and EV4). Finally, imaging of mNG-ARL13B and LifeAct-TagRFP in *Bbs4*^KO/KO MEFs expressing the CDC42 WT and DN variants was performed using the filters for FITC (523/36) and TRITC (576/89) for 120 min in 2 min intervals to acquire a time-lapse videos (Fig. EV5). Time-lapse videos were de-convolved using Huygens Professional v. 21.04 (Scientific Volume Imaging, Hilversum, Netherlands) using the classic maximum likelihood estimation (CMLE). Signal to noise ratio was set to 40 (Figs. 4C, 5A,B, EV3 and EV4) and to 11/23 (FITC/TRITC channels in EV5) with the area radius of 0.5 μm. One brick mode was used for the de-convolution and a maximum iteration of 40. De-convolved time-lapse videos were further analyzed using Fiji ImageJ. For 3D analysis and visualization Imaris viewer 9.9.1 (Bitplane, Oxford Instruments plc) was used. Movies were extracted from de-convolved time-lapse videos. Time-lapse video was cropped for the region of interest and time and animated with a rotation of 360°.

Cilium length in GPR161-mCherry expressing cells was estimated every 5 min during acquisition using the line tool in Fiji ImageJ software. The cilium length at each time point was then normalized to the length measured at 0 min.

## Fluorescence lifetime imaging microscopy, data processing, and analysis

MEFs were seeded on 12-mm coverslips and serum starved for 24 h. To activate SHH pathway, cells were incubated with the 200 nM SAG for 2 h prior fixation. To inhibit the Raichu probe, cells were incubated with ML141 or DMSO as control for 2 h prior fixation. The coverslips were fixed with 4% formaldehyde for 5 min, rinsed in PBS and water, air-dried and embedded in ProLong™ Gold antifade reagent and imaged on the same day. The FLIM-FRET measurements were done using Leica Stellaris 8 Falcon confocal microscope equipped with an oil immersion objective HC Plan-Apochromat 63× NA 1.4 oil, CS2 at room temperature. The acquisition and subsequent analysis were done using the built-in Phasor FLIM software tool (Digman et al, 2008). Donor CFP fluorescence was excited by a pulsed (40 MHz) white laser tuned at 440 nm, and emitted photons between 457 nm and 488 nm were collected (max. 500 photons per pixel) using the HyD X2 detector in photon counting mode. The acquisition was performed in $512 \times 512$-pixel format with pinhole 2.5 AU, at a speed of 400 Hz in bidirectional mode and 8-bits resolution. In each experiment, non-transfected cells were used to estimate the autofluorescence and the N-CFP-PAK-CDC42 expressing WT MEFs were used to estimate the lifetime of donor CFP only. These two parameters were used to estimate the FRET fraction and lifetime of the donor CFP (Raichu probe) in the phasor space (Digman et al, 2008). The values of these parameters were extracted from the region of interest including the analyzed cilium.

To correlate cilia length with donor lifetime, the cilia length was estimated using the signal from the N-Raichu-CDC42 probe during FRET-FLIM measurements, applying the line tool directly within the Leica LAS AF software platform and Phasor FLIM module (Fig. 3F).

### Statistical analysis

Statistical analysis was performed using GraphPad Prism Version 5.04 and 9. The statistical tests are indicated in the respective Figure legends. The following statistical tests were used and are indicated in the respective Figure legends: Mann–Whitney test, Paired t-test, Chi-square test, Pearson correlation, and F-test for comparing slopes of the regression lines.

## Data availability

This study includes no data deposited in external repositories.

The source data of this paper are collected in the following database record: biostudies:S-SCDT-10_1038-S44319-024-00326-z.

## Peer review information

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

## Acknowledgements

This project has received funding from the Czech Science Foundation (21-21612S to MH), core funding provided by the Institute of Molecular Genetics of the Czech Academy of Sciences (RVO 68378050), project National Institute of Virology and Bacteriology (Programme EXCELES, LX22NPO5103 to OS) - funded by the European Union - Next Generation EU, Charles University Grant Agency (386321 to AP and OI) and EU Horizon 2020 Research and Innovation programme (MSCF 847693 to MSC).

We acknowledge the Light Microscopy Core Facility, IMG, Prague, Czech Republic, supported by MEYS – LM2023050 and RVO – 68378050-KAV-NPUI, for their support with the widefield and confocal imaging and data analysis presented herein.

## Author contributions

**Avishek Prasai**: Formal analysis; Funding acquisition; Investigation; Visualization; Methodology; Writing—original draft; Writing—review and editing. **Olha Ivashchenko**: Formal analysis; Funding acquisition; Investigation; Visualization; Methodology; Writing—original draft; Writing—review and editing. **Kristyna Maskova**: Formal analysis; Investigation; Writing—original draft; Writing—review and editing. **Sofiia Bykova**: Formal analysis; Investigation; Writing—review and editing. **Marketa Schmidt Cernohorska**: Formal analysis; Funding acquisition; Investigation; Writing—original draft; Writing—review and editing. **Ondrej Stepanek**: Formal analysis; Funding acquisition; Investigation; Writing—original draft; Writing—review and editing. **Martina Huranova**: Conceptualization; Resources; Data curation; Formal analysis; Supervision; Funding acquisition; Validation; Investigation; Visualization; Methodology; Writing—original draft; Project administration; Writing—review and editing.

Source data underlying figure panels in this paper may have individual authorship assigned. Where available, figure panel/source data authorship is listed in the following database record: biostudies:S-SCDT-10_1038-S44319-024-00326-z.

## Disclosure and competing interests statement

The authors declare no competing interests.

# Expanded View Figures

**Figure EV1.   Actin regulators regulate the cilia length in different manner.**

(A) Expansion microscopy of the cilia axoneme of the *BBS1*$^{KO/KO}$ and *BBS7*$^{KO/KO}$ RPE1 cells visualized by staining with antibody to acetylated tubulin. White arrows point to bulges at the cilia tips. Scale bar, 2 µm. (B) The graph shows the frequency of bulges at the cilia tips visualized by expansion microscopy of cilia in WT and *BBS4*$^{KO/KO}$ RPE1 cells. The number of observed bulges out of the total count of imaged cilia is shown. Statistical analysis was done using the contingency table and one-sided Chi-square test. (C) Representative micrographs of the cilia visualized by staining with antibody to acetylated tubulin (Ac-tub) and staining of actin cytoskeleton with Phalloidin-TexasRed in the WT RPE1 cells treated with inhibitors of ROCK1/RHOA (Y27632), CDC42 (ML141), RAC1 (*CAS*1177865-17-6 (CAS)), ARP2/3 (CK666), and actin polymerization (Cytochalasin D) for 2 h. Scale bar, 5 µm. (D) Quantification of the cilia length based on acetylated tubulin signal in the WT and *BBS4*$^{KO/KO}$ RPE1 cells treated with DMSO or CK666 or Cytochalasin D (CytoD) for 2 h as in (C). Medians with interquartile range from three independent experiments ($n = 220$–$320$ cilia). (E) Quantification of the cilia length based on acetylated tubulin signal in the WT and *BBS4*$^{KO/KO}$ RPE1 cells treated with vehicle or CAS for 2 h as in (C). Medians with interquartile range from three independent experiments ($n = 250$–$280$ cilia). (F) Representative micrographs of the cilia visualized by staining with antibody to acetylated tubulin (Ac-tub) and ARL13B in the WT and *BBS4*$^{KO/KO}$ RPE1 cells treated with DMSO or ML141 for 2 h. Scale bar, 5 µm. (G) Quantification of the cilia length in WT, *BBS4*$^{KO/KO}$, and *BBS4*$^{KO/KO}$ RPE1 cells expressing YFP-BBS4 treated with DMSO or ML141 for 2 h and stained with antibodies to acetylated tubulin (Ac-tub) and ARL13B. Medians with interquartile range from three independent experiments ($n = 200$–$250$ cilia). (H) Quantification of the frequency of ARL13B foci at the cilia tips in WT, *BBS4*$^{KO/KO}$, and *BBS5*$^{KO/KO}$ RPE1 cells expressing YFP-BBS4 treated with DMSO or ML141 for 2 h in (G). Mean and SD of three independent experiments ($n = 200$–$250$ cilia). (I) Intensity plot profiles of the Ac-tub and ARL13B signal measured from the cilia base (B) to the cilia tip (T) in *Bbs4* KO MEFs treated with ML141 in Fig. 1I. The ARL13B signal extends (~0.5 µm) beyond the Ac-tub signal. (J) Representative chromatogram analysis of DNA sequencing data for WT ST2 and *Ift88* KO ST2 cell clones reveals a homozygous mutation characterized by the insertion of a T nucleotide between bases 248 and 249 of the *Ift88* gene (NM_009376.3, arrow). The position of the sgRNA and PAM sequence used for targeting is indicated. (K) Representative Western blot from total cell lysates prepared from WT MEFs, SAG/DMSO treated GPR161-mCherry WT MEFs (experiments #1–4 in Fig. 1K–M), WT and *Ift88* KO ST2 cells and *Ift88* KO ST2 cells expressing the IFT88-mNeonGreen and probed with antibodies to IFT88 and β-actin. The arrow highlights the IFT88 band (red star) corresponding to the expected size of 100 kDa, which is absent in the *Ift88* KO ST2 cells. Data information: Statistical significance was calculated using the one-sided Chi-square test (B), two-tailed Mann–Whitney test (D, E, G) and two-tailed paired t-test (H) and the obtained p-values are indicated. Merged micrographs show nuclei staining with DAPI—blue (F).

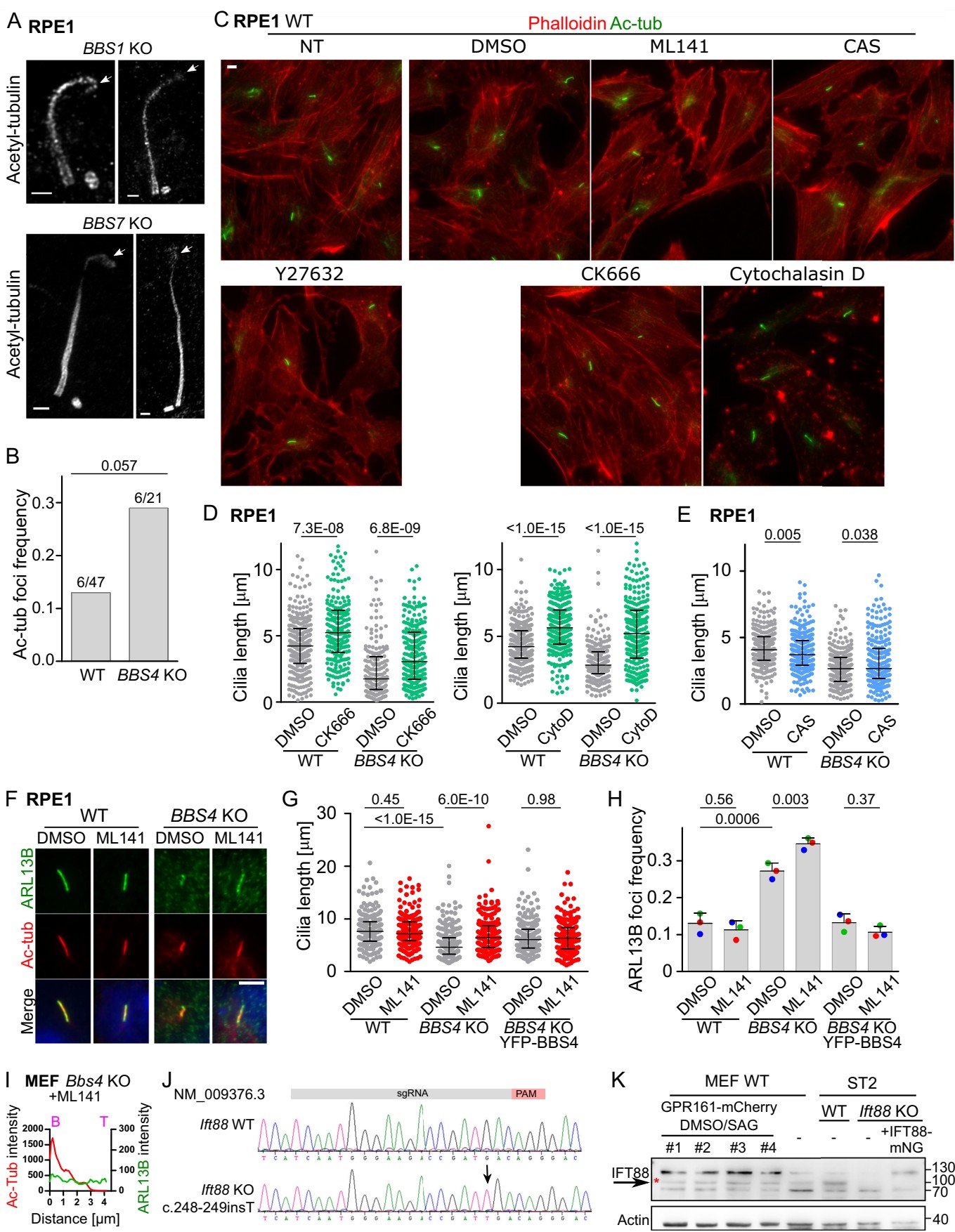

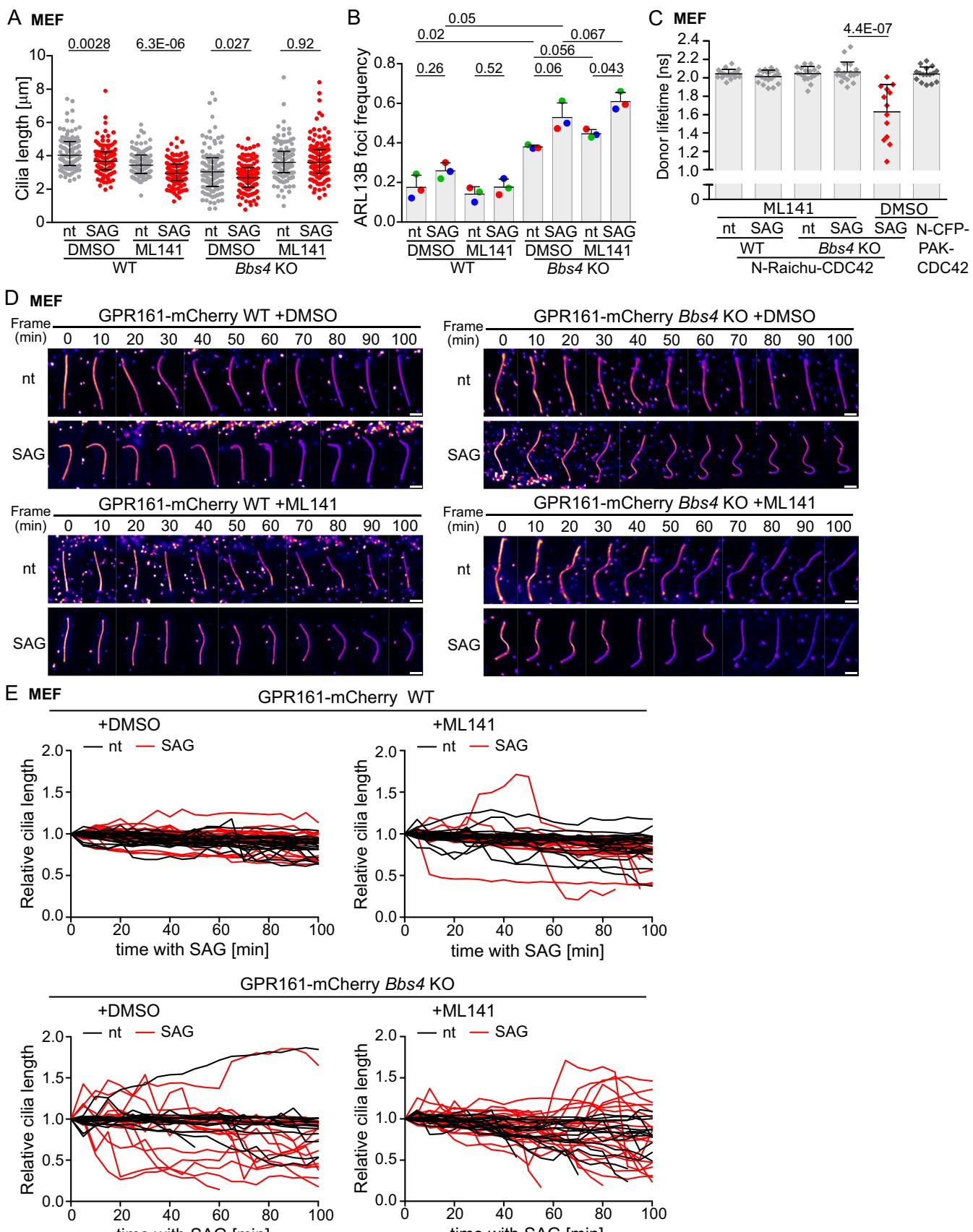

◄

**Figure EV2. CDC42 hyperactivation drives cilia shortening in BBSome-deficient cells upon SHH triggering.**

(A) Quantification of the cilia length, in non-treated (nt) and SAG induced—2 h, WT and $Bbs4^{KO/KO}$ MEFs concomitantly treated with DMSO or ML141 and stained with antibodies to acetylated tubulin (Ac-tub) and ARL13B. Medians with interquartile range from three independent experiments ($n = 90$–130 cilia). (B) Quantification of the frequency of ARL13B foci at the cilia tips in non-treated (nt) and SAG induced—2 h, WT and $Bbs4^{KO/KO}$ MEFs concomitantly treated with DMSO or ML141 in (A). Mean and SD of three independent experiments ($n = 90$–130 cilia). (C) The donor lifetime values extracted from the FRET-FLIM analysis of cilia as a region of interest in non-treated (nt) and SAG induced—2 h, WT and $Bbs4^{KO/KO}$ MEFs expressing N-Raichu-CDC42 treated with ML141 or DMSO and values measured for the no-FRET control expressed in WT MEFs treated with DMSO. Mean and SD of three (WT) and four (KO, no-FRET) independent experiments ($n = 13$–20 cilia). (D) Representative micrographs of the cilia in non-treated (nt) and SAG induced WT and $Bbs4^{KO/KO}$ MEFs expressing GPR161-mCherry, pre-treated for 30 min with DMSO or ML141 and imaged every 1 min for 100 min. Frames were extracted from the live cell videos (17–30 cilia per condition in two independent experiments). Scale bar, 2 μm. (E) Plots depict the variable dynamics of the length of the individual cilia in non-treated (nt) and SAG induced WT and $Bbs4^{KO/KO}$ MEFs expressing GPR161-mCherry, pre-treated for 30 min with DMSO or ML141 and imaged every 1 min for 100 min in (D). The length of the cilium was normalized to the length measured at time 0 min. In total 17–30 cilia were monitored per condition in two independent experiments. Data information: Statistical significance was calculated using the two-tailed Mann–Whitney test (A, C) and two-tailed paired t-test (B) and the obtained p-values are indicated.

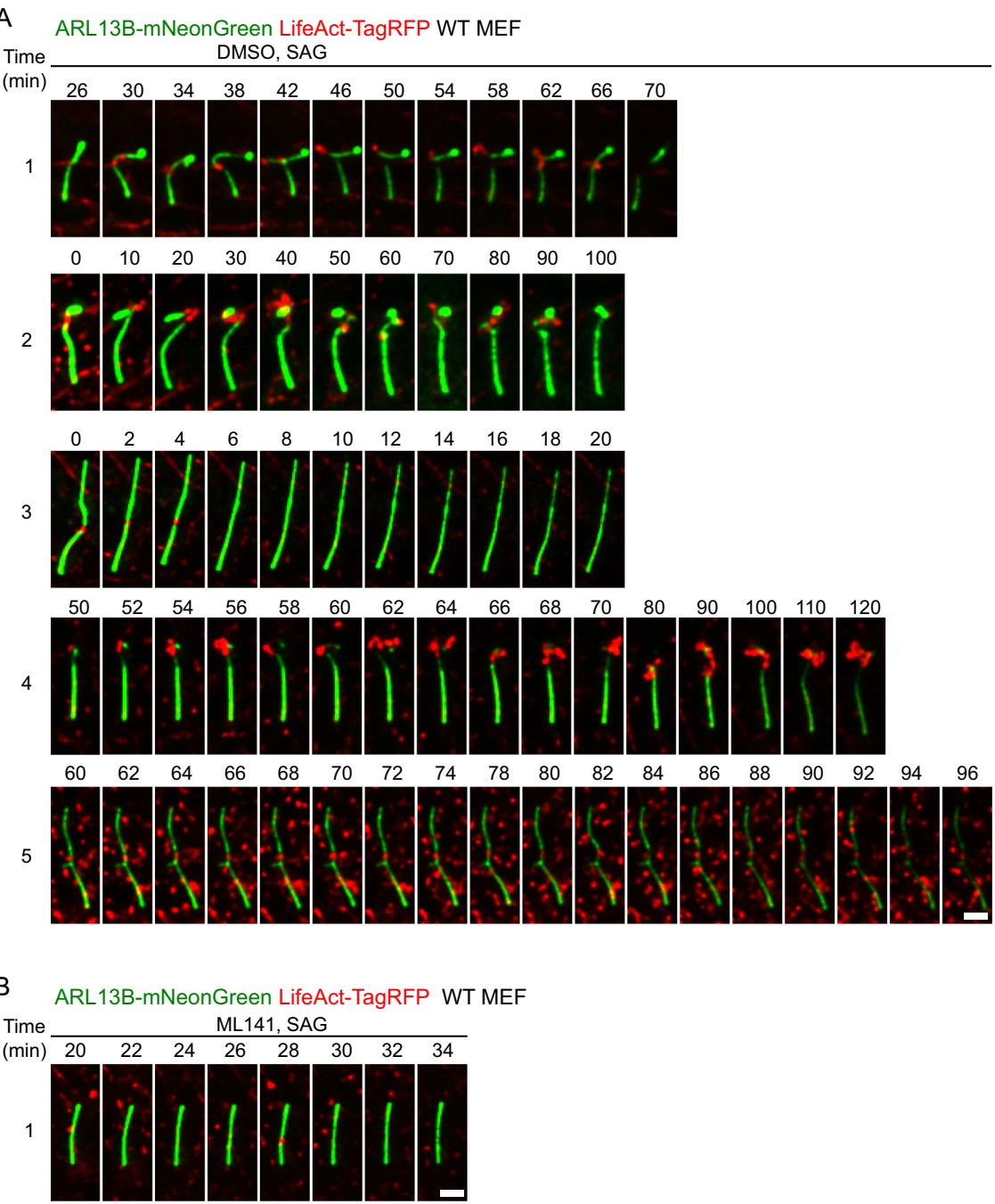

**Figure EV3. CDC42 is required for actin polymerization in cilia in WT cells.**

(A) Representative micrographs show F-actin polymerization events observed in cilia in WT MEFs expressing mNeonGreen-ARL13B (green) and LifeAct-TagRFP (red) treated with SAG and DMSO and imaged for 2 h. Frames were extracted from time-lapse videos (51 in total, 6 actin polymerization events, four independent experiments). Scale bar, 2 μm. (B) Representative micrographs show F-actin polymerization events observed in cilia in WT MEFs expressing mNeonGreen-ARL13B (green) and LifeAct-TagRFP (red) treated with SAG and ML141 and imaged for 2 h. Frames were extracted from time-lapse videos (54 in total, 2 actin polymerization events, four independent experiments). Scale bar, 2 μm. Maximum intensity projections of the z-stacks were done using Fiji ImageJ software and the intensities for both channels were adjusted post acquisition for better visualization.

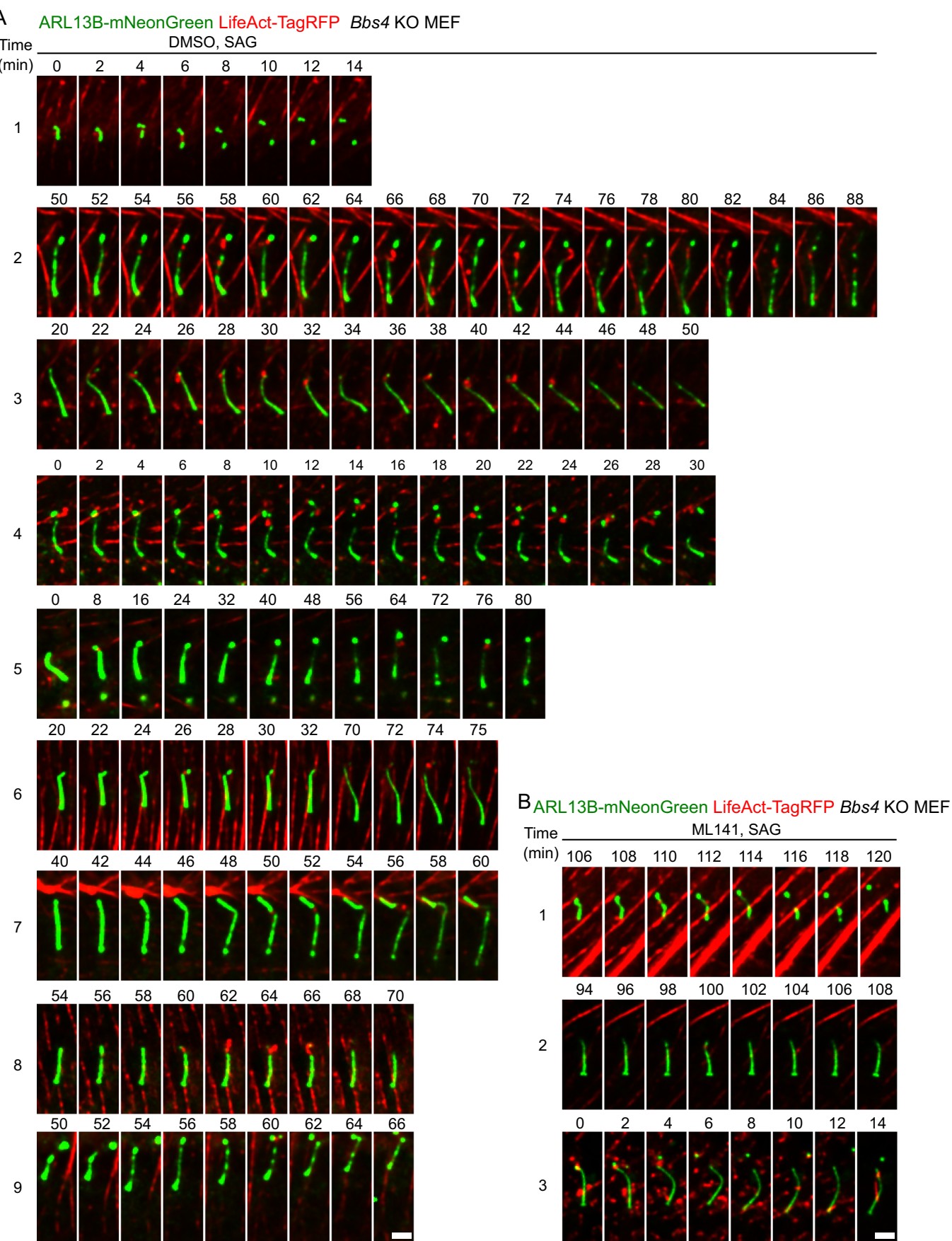

**Figure EV4. CDC42 is required for actin polymerization in cilia in BBSome-deficient cells.**

(A) Representative micrographs show F-actin polymerization events observed in cilia in *Bbs4*[KO/KO] MEFs expressing mNeonGreen-ARL13B (green) and LifeAct-TagRFP (red) treated with SAG and DMSO and imaged for 2 h. Frames were extracted from time-lapse videos (39 in total, 10 actin polymerization events, four independent experiments). Scale bar, 2 μm. (B) Representative micrographs show F-actin polymerization events observed in cilia in *Bbs4*[KO/KO] MEFs expressing mNeonGreen-ARL13B (green) and LifeAct-TagRFP (red) treated with SAG and ML141 and imaged for 2 h. Frames were extracted from time-lapse videos (39 in total, 4 actin polymerization events, four independent experiments). Scale bar, 2 μm. Maximum intensity projections of the z-stacks were done using Fiji ImageJ software and the intensities for both channels were adjusted post acquisition for better visualization.

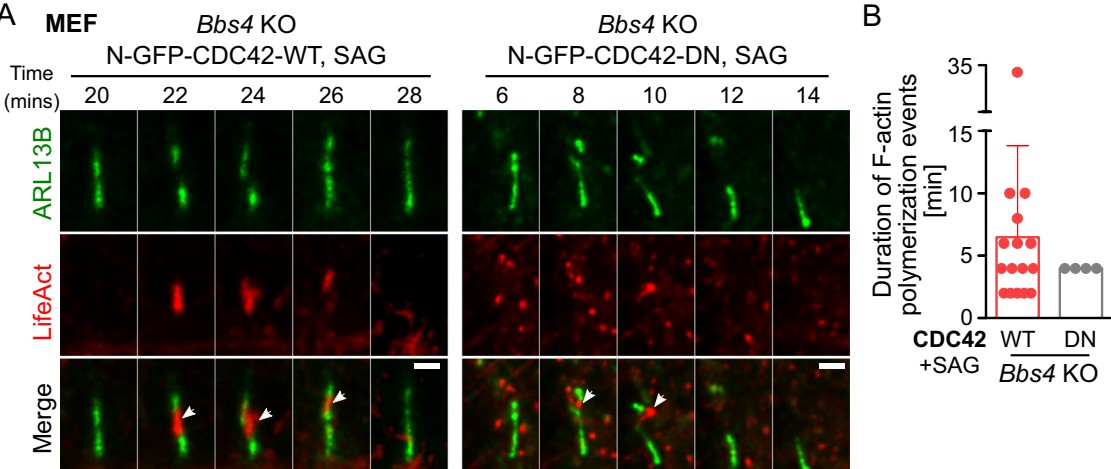

**Figure EV5.  Activity of cilia localized CDC42 is required for actin polymerization in cilia in BBSome-deficient cells.**

(A) Representative micrographs depicting ciliary ectocytosis and actin polymerization detected by live cell imaging of *Bbs4*^KO/KO^ MEFs expressing mNG-ARL13B and LifeAct-TagRFP in the presence of WT or DN version of CDC42. The cells were treated with SAG and imaged every 2 min for 2 h. White arrows point to the F-actin polymerization events. Cilia base at bottom. Scale bar, 2 μm. Maximum intensity projections of the z-stacks were done using Fiji ImageJ software and the intensities for both channels were adjusted post acquisition for better visualization. (B) The plot depicts the duration of the observed actin polymerization events detected in cilia in the *Bbs4*^KO/KO^ MEFs expressing mNG-ARL13B and LifeAct-TagRFP in the presence of WT or DN version of CDC42 upon treatment with SAG during live cell imaging—six independent experiments in each condition (A). Means with SD are shown.

