## [Peer Review File · EMBO Reports]

BBSome-deficient cells activate intraciliary CDC42 to trigger actin-dependent ciliary ectocytosis.

Avishek Prasai, Olha Ivashchenko, Kristyna Maskova, Sofii Bykova, Marketa Schmidt Cernohorska, Ondrej Stepanek, and Martina Huranova

Corresponding author(s): Martina Huranova (martina.huranova@img.cas.cz)

Review Timeline:

Submission Date:	4th Sep 23
Editorial Decision:	24th Oct 23
Appeal Received:	20th Nov 23
Editorial Decision:	19th Dec 23
Revision Received:	22nd May 24
Editorial Decision:	24th Jul 24
Revision Received:	2nd Sep 24
Editorial Decision:	4th Oct 24
Revision Received:	20th Oct 24
Accepted:	25th Oct 24

Transaction Report:

Dear Dr. Huranova

Thank you for the submission of your research manuscript to EMBO reports. We have now received the full set of referee reports on it, which are copied below.

I am sorry to say, that the evaluation of your manuscript is not a positive one. As you will see, while all three referees acknowledge that the findings are potentially interesting, they all raise a number of largely overlapping concerns. It appears that the referees are concerned that the study, as it stands, fails to provide convincing evidence that ciliary shortening in BBS4 KO cells is caused by excessive Cdc42-dependent ectocytosis. The referees also raise concerns regarding the conceptual advance provided over earlier work on Cdc42 and the reliance on overexpressed proteins.

Given the nature of these concerns, the amount of work required to address them, the uncertain outcome of these experiments, and the fact that EMBO reports can only invite revision of papers that receive enthusiastic support from a majority of referees, I am sorry to say that we cannot offer to publish your manuscript.

I am sorry to disappoint you on this occasion, and hope that the referee comments will be helpful in your continued work in this area.

Yours sincerely

Referee #1:

Prasai et al. set out to identify factors involved in ciliary ectocytosis and cilia shortening of BBSome-deficient cells. Their findings revealed that BBSome-deficient cells exhibit shorter cilia and accumulations of membrane proteins in the cilia. The hypothesis proposed by the researchers suggests that continuous ectocytosis is responsible for the observed shorter cilia in BBSome-deficient cells. Furthermore, they demonstrated that the reduction in cilia length can be rescued by inhibiting ROCK1 and CDC42 and actin polymerization factors (Cytochalasin D and ARP2/3 inhibitors). Subsequently, they established that CDC42 is hyperactivated in BBSome-deficient cells and is crucial for actin polymerization within the cilia.

The major issues:

Ectocytosis is an intriguing topic and this paper provided new findings. However, the current level of the paper lacks new mechanistic findings for publication in EMBO Reports. For example, RhoA inhibitors mediated-rescues of cilia length in BBSome-deficient cells were already reported by Victor Hernandez-Hernandez et al. 2013, Hum Mol Genet. In the same paper, they showed that BBSome regulates actin polymerization. Prasai et al. just added that an increase in actin polymerization in BBSome-deficient cells requires CDC42. The link between CDC42 and actin polymerization is already well known.

1- "We hypothesized that the cilia shortening in the BBSome-deficient cell lines, previously observed by others [9, 10, 12] and us [11] (Fig. 1A and B) is caused by continuous ectocytosis."

Although the hypothesis is intriguing, for it to be substantiated that constitutive ectocytosis causes the shortening of ciliary length in BBSome-deficient cells, they must present additional, independent evidence supporting this hypothesis.

2- In the absence of BBSome, when cells face difficulty in retrieving ciliary membrane proteins, they initiate an alternative pathway, such as ciliary ectocytosis, to eliminate these proteins from cilia. The mechanism that triggers this alternative pathway in BBSome-deficient cells raises intriguing questions. How does BBSome activate CDC42-via direct or indirect activation?

3- Is the accumulation of membrane proteins within cilia adequate to activate ectocytosis? I need to note here Nachury and colleagues showed that ectocytosis could not be attributed to the overexpression of ciliary membrane proteins (Nager et al. 2017). Prasai et al. might offer additional insights into these queries. Notably, they conducted experiments involving the overexpression of GPR161-mCherry in BBSome-deficient cells, suggesting a potential avenue for further exploration in their research.

The minor issues:

- 1- there is a type mistake: buldge-like structures, it should be bulge-like structures
- 2- Figure 1I. the beginning of cilia should be marked, thus readers can better view the position of accumulations.

Referee #2:

In their present manuscript, Martina Huranova and colleagues have investigated the mechanisms involved in the shortening of primary cilia observed in cells invalidated for the expression of BBSome subunits. Testing several molecules targeting Rho GTPases, they provide strong evidence that ML141, an inhibitor of CDC42, was able to increase ciliary length in BBS4 KO RPE1 cells and MEFs (mouse embryonic fibroblasts). They further investigate the role of CDC42 in BBS4 KO MEFs in the context of the activation of the Hedgehog signaling pathway. The presented data show that ML141 treatment and cilia-targeted dominant negative form of CDC42 increase ciliary length in these cells as well the SAG-induced decrease of ciliary length. Using a FRET sensor, they also nicely show increased ciliary CDC42 activity which leads to increased actin polymerization within cilia. Altogether these data indicate that the observed decreased ciliary length upon loss of BBS4 is linked to overactivation of CDC42 within cilia leading to actin polymerization ciliary ectosome formation. Altogether, the data presented are of high quality and interesting. I however have several concerns which are listed below.

Main concerns:

- the main hypothesis raised by the authors, based on the literature and their data, is that shortening of cilia in BBS4 KO cells is linked to increased production of ciliary-derived vesicles (ectosomes) caused by abnormal activation of CDC42 and actin polymerization. It is therefore somehow surprising that ectosome release was not quantified in this study. This can be easily quantified from the live cell imaging data. It can be also quantified by purification of secreted vesicles in the different conditions. The assumption that ciliary foci of membrane proteins correlate with ectosome release should be validated.
- the authors used RPE1 clones KO for BBS subunits that they have described in a previous study. They used WT RPE1 cells as a control but it is not clear if those cells correspond to the parental cell line or to "WT" clones isolated in the same time as KO clones. By the way, in their previous paper, they generated more appropriate control cell lines in which they re-expressed corresponding tagged-BBS subunit in the generated BBS KO cells. Those control cells should be used to validate that some of the phenotypes described here including ARL13B and GPR161 foci, are not linked to a single clone effect.
- It is quite surprising that, while RPE1 KO cells were used in Fig.1 to identify the effect of ML141 on ciliary length, they were not used thereafter, the authors shifted to KO MEFs. It would be important to use the two cell models in parallel to strengthen the data. In addition, ciliary tip foci which are considered by the authors as forming ectosomes, were quantified using ARL13B staining in RPE1 cells while GPR161 was used in MEFs. Quantifications should be done for these two markers in the two cell lines. By the way, the authors also used overexpressed GPR161 or ARL13B in KO MEFs (Figs 4 and 5) which is problematic based on the known effects of the overexpression of ciliary GPCRs and small GTPases on ciliary length. Except in case of live cell studies, the use of stainings of endogenously expressed proteins must be done in parallel.
- based on their present and previous works, the authors claim that the mechanism they have identified account for BBSome KO conditions. However, in the present study, they focused on BBS4 and the results presented in Fig.1 indicate that ML141 does not significantly increase ciliary length in BBS1 KO cells. The authors must clarify this point.

Specific points:

- Fig.1C: Arrows are supposed to point to "bulge-like structures" but these structures do not appear clearly in the provided pictures. A ciliary membrane marker may be used instead of acetylated-alpha-tubulin (Ac-Tub). This phenotype should be quantified.
- Fig.1D: The use of Ac-Tub staining to measure ciliary length could be problematic in case of acetylation defects. Did the authors double check using ARL13B staining? Similarly, on which staining was based the quantification of ciliary length in Fig.1J?
- Fig.1H and Fig.1J: Statistics are lacking for the comparison between WT vs BBS4 KO (DMSO).
- Fig.2A: The ciliary GPR161 staining is very faint in WT cells. If this is the case it might have masked the detection of ciliary foci in WT cells. ARL13B should be used too (see above).
- Fig.2C: Statistics between WT and BBS4 KO nt must be provided to see if there is a difference in the GPR161 foci.
- Fig.2F: Enlarged views of cilia should also be shown for WT cells similarly as for BBS4 KO cells.
- Fig.2G: Effect of WT and DN CDC42 ciliary expression should be used to address the points raised in Fig.1? What could be their effects on ciliary length (WT vs BBS4 KO)?
- Fig.3: Does increased CDC42 activity result in ectosome and/or foci formation?
- Fig.4A: Experiments have been performed in BBS4 KO MEFs but what about ciliary distribution of overexpressed GPR161 in WT MEF? Do these foci ever formed ectosomes (live cell analyses in 4C)?
- Fig. 5: As mentioned above, results in Fig.5 were generated overexpressing ARL13B. It would be important to quantify foci for endogenous ARL13B similarly as for GPR161. Again ectosome production was not quantified.

Referee #3:

The manuscript by Prasai et al. entitled "BBSome-deficient cells activate intraciliary CDC42 to trigger actin-dependent ciliary ectocytosis" aims to reveal CDC42 as a key driver of ciliary ectocytosis. The authors describe that CDC42 triggers actin polymerization, ciliary ectocytosis, and cilia shortening in some BBSome-deficient cells, whereas inhibition of CDC42 in BBSome-deficient cells decreases ciliary actin polymerization.

The authors have extensively studied the role of CDC42 in BBSome-deficient cells but the role of ciliary CDC42 under physiological conditions in WT cells remains poorly understood. Also, the specificity in the cilium vs. cytoplasm is not clear for all experiments.

Before being considered for publication, the following points need to be addressed:

Major issues

- Sheu et al., Cell 2022 demonstrated that ciliary 5-HTR6 stimulation activates a non-canonical Gαq/11-RhoA pathway, which modulates nuclear actin polymerization. The authors need to test whether nuclear actin is also remodeled in their context. Furthermore, the literature also needs to be mentioned and discussed in the introduction and discussion
- Figure 1G: what is shown in the right column? WT only on the left?
- Always show mean or median +/- S.D. (e.g., Figure 1H, 2B, 2C etc.)
- Ref. 31 has only shown this connection for renal epithelial cells. The authors should also show experimentally in WT MEFs that CDC42 play a dual role and also controls ciliogenesis through the exocyst context. This should also be analyzed in Bbs4-KO cells to reveal that reduced ciliary length under basal conditions is not due to CDC42 controlling exocyst activity.
- Figure 2C: The frequency of GPR161-specific foci at the ciliary tip is not different between SAG and non-treated cells in Bbs4-KO cells but the text gives different interpretation. Please rephrase.
- Figure 2E: Why are Bbs4-KO cells Shh-responsive with respect to ciliary length? According to the GPR161 data, GPR161 accumulates in the cilium and does not exit the cilium in Bbs4-KO cells.
- Figure 2H: The comparison to WT (non-transfected vs. DN expressing) cells for the GPR161 foci analysis is missing. This needs to be included to reveal whether ciliary CDC42 also plays a role in WT cells.
- Figure 3C/D: to demonstrate the specificity of the sensor and put the SAG responses into context, the authors need to apply the CDC42 inhibitor to demonstrate the dynamic range of the sensor in the cilium. Furthermore, in Figure 3C, only one cilium shows a higher CDC42 activity under basal conditions in Bbs4-KO cells. Also here, the relative difference upon addition of the inhibitor should be determined to see larger changes in Bbs4-KO cilia compared to WT cilia
- Figure 4A/B: perform comparable experiments in WT cells.
- Figure 4C: according to the text, this figure shows no difference in WT ciliary length upon SAG treatment whereas Figure 2G shows a significant decrease in ciliary length. Please comment on these differences! This makes the interpretation of Figure 4C for Bbs4-KO cells rather difficult.
- Figure 5A: from the pictures, it looks like that the actin dynamics in the cytosol are much more pronounced in Bbs4-KO cells upon CDC42 inhibition than in WT cells. The authors need to perform these experiments using the ciliary targeted DN-CDC42 to demonstrate specificity.
- The authors have not demonstrated ectocytosis of a GPCR in BBSome-deficient cells. As the regulation of ectocytosis is the key finding of the manuscript, the authors need to demonstrate that one of the GPCRs, which is rather absent from primary cilia in BBSome-deficient cells, is exported via ectocytosis.

Minor issues

- Add Forsythe and Beales, 2013 to reference 7
- Figure 1(A): stained with ...antibody. (The word antibody is missing). Change throughout the text.
- Discussion: "Using cilia-targeted dominant negative CDC42 mutant, we revealed that intraciliary CDC42 triggers ciliary ectocytosis that controls the overall ciliogenesis process in the BBSome-deficient cells." Rephrase as overall ciliogenesis might also be regulated by cytoplasmic CDC42 activity (see above).
- Discussion: following sentences - the authors have not shown that only ciliary and not cytoplasmic CDC42 results in cilia shortening in BBSome-deficient cells (also see questions above).
- The information given in the Material & Methods part is rather limited and does not allow to fully understand the experiments performed. It would be great if the authors could elaborate a bit more how the experiments have been performed. The same applies to the figure legends, which are also rather short and lack information.

** As a service to authors, EMBO Press provides authors with the ability to transfer a manuscript that one journal cannot offer to publish to another journal, without the author having to upload the manuscript data again. To transfer your manuscript to another EMBO Press journal using this service, please click on

Link Not Available

Prasai et al. EMBOR-2023-58108V2-Q

Point-by-point response

Referee #1:

Prasai et al. set out to identify factors involved in ciliary ectocytosis and cilia shortening of BBSome-deficient cells. Their findings revealed that BBSome-deficient cells exhibit shorter cilia and accumulations of membrane proteins in the cilia. The hypothesis proposed by the researchers suggests that continuous ectocytosis is responsible for the observed shorter cilia in BBSome-deficient cells. Furthermore, they demonstrated that the reduction in cilia length can be rescued by inhibiting ROCK1 and CDC42 and actin polymerization factors (Cytochalasin D and ARP2/3 inhibitors). Subsequently, they established that CDC42 is hyperactivated in BBSome-deficient cells and is crucial for actin polymerization within the cilia.

The major issues:

Ectocytosis is an intriguing topic and this paper provided new findings. However, the current level of the paper lacks new mechanistic findings for publication in EMBO Reports. For example, RhoA inhibitors mediated-rescues of cilia length in BBSome-deficient cells were already reported by Victor Hernandez-Hernandez et al. 2013, Hum Mol Genet. In the same paper, they showed that BBSome regulates actin polymerization. Prasai et al. just added that an increase in actin polymerization in BBSome-deficient cells requires CDC42. The link between CDC42 and actin polymerization is already well known.

We thank to this Referee for reading and commenting on our manuscript. It seems that we failed to sufficiently explain the major point of our study, which is the newly identified mechanism behind the ciliary ectocytosis in the BBSome deficient cells. Whereas, the link between CDC42 and actin polymerization in the cytoplasm is indeed already well known, we are not aware of a single report about the role of intraciliary CDC42.

Major mechanistic advances of our study are:

- 1. We showed for the first time that actin polymerization within the cilia is regulated by CDC42, a RHO GTPase family member.*
- 2. We found that ciliary CDC42 is hyperactivated during SHH signalling in the BBSome-deficient cells.*
- 3. We showed that intraciliary CDC42 activity facilitates ectocytosis of excessive GPCRs, when their retrograde transport is defective, which occurs in ciliopathies, such as Bardet-Biedl Syndrome.*

Thus, we are submitting a first study showing that intraciliary activity of a Rho family member is substantially enhanced by defective retrograde transport, which is a hallmark of Bardet-Biedl Syndrome. We believe that altogether, these pivotal observations warrant sufficient novelty for publication in EMBO Reports. We will revise our manuscript to make these advances more clear.

We are familiar with the data published by Hernandez-Hernandez et al. 2013 suggesting that the overall cellular activity of RHOA is enhanced by the BBSome deficiency. Although, Hernandez-Hernandez et al. reported differences in the cytosolic actin cytoskeleton in the BBSome-deficient cells, they did not address the actin inside the cilia at all, because the very first reports about the F-actin in the cilia appeared a couple of years later (Nager et al. 2017 PMID: 28017328, Phua et al. 2017 PMID:

28086093). Of note, there are still very few studies focusing on the ciliary actin filaments and this phenomenon is still poorly understood.

We were not able to reproduce the key observation by Hernandez-Hernandez et al. in RPE1 cells (Figure 1 here). To study the role of the RHOA pathway, Hernandez-Hernandez et al. used the ROCK1 inhibitor (downstream effector of RHOA) to rescue the cilia length in the BBSome-deficient cells. These data (Fig. 7B-E in Hernandez-Hernandez et al.) are in line with our observations (Fig. 1D-E in our manuscript), as both studies showed that ROCK1 inhibition prolongs cilia in WT as well as in the BBSome deficient cells, indicating that the effect of ROCK1 inhibition is general and not specific to the BBSome deficiency (which differs strikingly from the CDC42 inhibition). Overall, these data do not support the scenario that RHOA hyperactivation in the BBSome-deficient cells is responsible for the cilia shortening, although they do not exclude this possibility either.

The most plausible scenario is that the ROCK1 inhibition (as well as less specific cytochalasin treatment or Arp2/3 inhibition – Supplemental Fig. 1B-C in our manuscript) leads to the remodelling of the cortical actin, which is known to be involved in ciliogenesis (Pan et al. 2007 PMID: 17488776, Rangel et al. 2019 PMID: 30718762). However, in our study, we focused on the intraciliary actin, which has been addressed only by very few studies so far. We showed that overall chemical, but also ciliary-specific genetic inhibition of CDC42 activity rescues the cilia length exclusively in the BBSome deficient cells, but not in WT cells, which does support the scenario that CDC42 hyperactivation is the major cause of cilia shortening in the BBSome cells.

Figure 1. Relative levels of the GTP loaded RHOA in starved and non-starved RPE1 cells. RHO-GTP levels were measured using a commercial kit.

1- "We hypothesized that the cilia shortening in the BBSome-deficient cell lines, previously observed by others [9, 10, 12] and us [11] (Fig. 1A and B) is caused by continuous ectocytosis."

Although the hypothesis is intriguing, for it to be substantiated that constitutive ectocytosis causes the shortening of ciliary length in BBSome-deficient cells, they must present additional, independent evidence supporting this hypothesis.

The pivotal work in the field of ciliary ectocytosis (Nager et al. 2017, Cell PMID: 28017328) showed the constitutive ectocytosis in cells with defects in the retrograde ciliary transport, including ARL6 (BBS3)-deficient cells. BBS3 is a Bardet-Biedl Syndrom-associated gene), although the symptoms of BBS3-deficient patients are different than of those with the BBSome deficiency (Niederlova et al. 2019 PMID: 31283077). However, as all the Refs asked for actually showing and quantifying the ectocytosis in our cellular system, we will revise our manuscript accordingly. We will:

1. Refer to the published work (Nager et al. 2017 PMID: 28017328).
2. Show the qualitative evidence (live cell imaging documenting the ectocytosis).
3. Isolate and quantify the ectosomes in the medium of cultures of WT and BBSome-deficient cells. We will also analyse their content (GPCRs). The respective protocols have been established and used in multiple labs (Nager et al. 2017 PMID: 28017328, Volz et al. 2021 PMID: 34580290, Loukil et al. 2021 PMID: 33846249).

2- In the absence of BBSome, when cells face difficulty in retrieving ciliary membrane proteins, they initiate an alternative pathway, such as ciliary ectocytosis, to eliminate these proteins from cilia. The mechanism that triggers this alternative pathway in BBSome-deficient cells raises intriguing questions. How does BBSome activate CDC42-via direct or indirect activation?

We are not showing that the BBSome activates CDC42. Actually, we are showing the very opposite: CDC42 is activated by the absence of the BBSome, suggesting that the not the BBSome, but the BBSome-deficiency activates CDC42. After studying the current literature, the link between the BBSome-deficiency and ciliary CDC42 activation is not that surprising. It has been shown that in the cytoplasm, the GPCR signaling (namely the Gas subunit) activate PDZ-RhoGEF, the major CDC42 activator (Castillo-Kauil et al. 2020, PMID: 33023908). Intriguingly, we found out that PDZ-RhoGEF as well as CDC42 are present in the ciliary proteome data, which were previously published (Mick et al. 2015 PMID: 26585297, Kohli et al. 2017 PMID: 28710093, van Dam et al. 2019 PMID: 31095607). Thus, it is very plausible that the BBSome deficiency causing the defective retrieval of activated GPCRs (Nager et al. 2017 PMID: 28017328) leads to hyperactivation of Gas (Mukhopadhyay et al. 2013 PMID: 23332756), which activates PDZ-RhoGEF, which in turn activates intraciliary CDC42. However, there are multiple CDC42 activators present in the ciliary proteomes (Mick et al. 2015 PMID: 26585297, Kohli et al. 2017 PMID: 28710093, van Dam et al. 2019 PMID: 31095607), which might cause functional redundancies in this process and difficulties to address this experimentally. However, this scenario is supported by our data showing that GPR161 signaling increases the CDC42 activity in the cilia of BBS-deficient, but not WT cells, showing a direct link between the GPCR signaling and CDC42 activation. Of note, GPR161 is associated with the Gas subunit (Mukhopadhyay et al. 2013 PMID: 23332756).

We will include this plausible explanation in the Discussion of the revised manuscript.

3- Is the accumulation of membrane proteins within cilia adequate to activate ectocytosis? I need to note here Nachury and colleagues showed that ectocytosis could not be attributed to the overexpression of ciliary membrane proteins (Nager et al. 2017). Prasai et al. might offer additional insights into these queries. Notably, they conducted experiments involving the overexpression of GPR161-mCherry in BBSome-deficient cells, suggesting a potential avenue for further exploration in their research.

We are not sure whether the accumulation of any membrane proteins within cilia triggers ectocytosis. Our data suggest that the accumulation of signaling GPCRs in the cilia triggers ectocytosis, which was also previously shown by Nager et al. 2017 PMID: 28017328. We are not sure whether this point is meant by the Ref as an issue in the current version of the manuscript or just an indication of a potential follow-up research.

Anyway, we will revise the manuscript pursuing this direction. First, we have several time-lapse movies, where we observed ectocytosis and we will include them in the revised manuscript (Figure 2. here). Because these events are relatively rarely captured (because of the delay between individual images in the time-lapse movies), it is difficult to quantify it, but they clearly document that the accumulation of GPR161 in the ciliary tips precedes the ectocytosis.

GPR161-mCherry *Bbs4* KO

ectocytosis #1

ectocytosis #2

Figure 2. Two examples of the ectosome release (cyan circles) from the cilia upon SHH induction in the *BBS4* deficient MEFs expressing GPR161-mCherry. Series of frames acquired every second. We will include more of these movies in the revised manuscript.

It should be noted that the overexpression of GPR161 in MEFs prolongs the cilia as we insert more material into the cilia. Moreover, the overexpression of mCherry-GPR161 doubles the frequency of the GPR161 foci increased in both WT and *Bbs4* KO cells lines (from 0.1 to 0.2 in WT and from 0.2 to 0.4 in the KO) (Figure 3 here). The ectocytosis might be enhanced in these lines by the GPR161 overexpression, but probably not substantially, as the ectocytosis is not able to fully compensate for the increase in the ciliary material, which leads to the cilia prolongation.

Figure 3. Quantification of the GPR161-mCherry foci in the WT and *Bbs4* deficient MEFs.

The minor issues:

1- there is a type mistake: buldge-like structures, it should be bulge-like structures

Thank you for spotting this. It will be corrected

2- Figure 1I. the beginning of cilia should be marked, thus readers can better view the position of accumulations.

We will fix this.

Referee #2:

In their present manuscript, Martina Huranova and colleagues have investigated the mechanisms involved in the shortening of primary cilia observed in cells invalidated for the expression of BBSome subunits. Testing several molecules targeting Rho GTPases, they provide strong evidence that ML141, an inhibitor of CDC42, was able to increase ciliary length in BBS4 KO RPE1 cells and MEFs (mouse embryonic fibroblasts). They further investigate the role of CDC42 in BBS4 KO MEFs in the context of the activation of the Hedgehog signaling pathway. The presented data show that ML141 treatment and cilia-targeted dominant negative form of CDC42 increase ciliary length in these cells as well the SAG-induced decrease of ciliary length. Using a FRET sensor, they also nicely show increased ciliary CDC42 activity which leads to increased actin polymerization within cilia. Altogether these data indicate that the observed decreased ciliary length upon loss of BBS4 is linked to overactivation of CDC42 within cilia leading to actin polymerization ciliary ectosome formation. Altogether, the data presented are of high quality and interesting. I however have several concerns which are listed below.

We thank to the reviewer for appreciating the quality of our data, for finding our data interesting, and for the comments.

Main concerns:

- the main hypothesis raised by the authors, based on the literature and their data, is that shortening of cilia in BBS4 KO cells is linked to increased production of ciliary-derived vesicles (ectosomes) caused by abnormal activation of CDC42 and actin polymerization. It is therefore somehow surprising that ectosome release was not quantified in this study. This can be easily quantified from the live cell imaging data. It can be also quantified by purification of secreted vesicles in the different conditions. The assumption that ciliary foci of membrane proteins correlate with ectosome release should be validated.

The ectocytosis events are infrequent and very fast. Moreover, the ectosome often diffuses away rapidly. Thus, the events are easy to be missed during the time-lapse movies, where we had to capture images every two minutes – to minimize phototoxicity and to monitor multiple fields of views (i.e., cilia) in two different fluorescence channels at the same time. This allowed us to acquire 40-50 time-lapse movies for individual cilia in each condition in total in three independent experiments (Figure 5 in the manuscript). For similar reasons, there are very few time-lapse videos documenting the ectocytosis published so far.

However, we have movies with much faster image acquisition, which sporadically captured ectocytosis/ectosome release using ARL13B-mNG or GPR161-mCherry. These movies show that the formation of the ciliary foci precedes the ectosome release. We will include these movies in the revised version of the manuscript (Figure 2 here).

GPR161mCherry *Bbs4* KO

ectocytosis #1

ectocytosis #2

Figure 2, same as above. Examples of the ectosome release (cyan circles) from the cilia upon SHH induction in the *BBS4* deficient MEFs expressing GPR161-mCherry. Series of frames acquired every second.

Since all the reviewers asked for quantification of ectocytosis, we will isolate and quantify the ectosomes released to the medium of particular cell-line cultures and we will assess their content. There are established protocols for this assay (Nager et al. 2017 PMID: 28017328, Volz et al. 2021 PMID: 34580290, Loukil et al. 2021 PMID: 33846249).

- the authors used RPE1 clones KO for BBS subunits that they have described in a previous study. They used WT RPE1 cells as a control but it is not clear if those cells correspond to the parental cell line or to "WT" clones isolated in the same time as KO clones. By the way, in their previous paper, they generated more appropriate control cell lines in which they re-expressed corresponding tagged-BBS subunit in the generated BBS KO cells. Those control cells should be used to validate that some of the phenotypes described here including ARL13B and GPR161 foci, are not linked to a single clone effect.

*The WT RPE1 cell line is the parental cell line, we will adjust the manuscript to make this clear. It is very difficult to generate CRISPR/Cas9 knockouts in the RPE1 cells (Prasai et al. PMID: 32759308). We managed to generate, reconstitute and validate one KO clone for BBS1 and BBS4 genes. We have additional data showing that RPE1 cells lacking BBS9 behave in the same way as BBS4 and BBS1 KO cells and we will add them to the revised manuscript (Figure 4 here). Since three KO lines lacking different BBSome subunits show the same phenotype, we believe that these data are even more convincing/robust than having three KO clones of a single BBS gene. It should be also noted that we confirmed the observations with RPE1 cells using an independent cellular model - primary WT and *Bbs4* KO MEF cells (Figure 1I-J of the manuscript).*

The cilia-shortening in these BBS-deficient RPE1 lines was previously shown to be caused by the BBSome-deficiency using the reconstituted lines (Prasai. et al. 2020 PMID: 32759308). However, this Ref. is right that we did not perform the CDC42 inhibitor treatments in these lines. For this reason, we will perform the proposed experiments with the reconstituted BBS4 KO BBS4-YFP RPE-1 cells (Prasai et al. 2020 PMID: 32759308) as suggested.

Figure 4. Correction of the panel F of the Figure 1 in the manuscript.

- It is quite surprising that, while RPE1 KO cells were used in Fig.1 to identify the effect of ML141 on ciliary length, they were not used thereafter, the authors shifted to KO MEFs. It would be important to use the two cell models in parallel to strengthen the data. In addition, ciliary tip foci which are considered by the authors as forming ectosomes, were quantified using ARL13B staining in RPE1 cells while GPR161 was used in MEFs. Quantifications should be done for these two markers in the two cell lines. By the way, the authors also used overexpressed GPR161 or ARL13B in KO MEFs (Figs 4 and 5) which is problematic based on the known effects of the overexpression of ciliary GPCRs and small GTPases on ciliary length. Except in case of live cell studies, the use of stainings of endogenously expressed proteins must be done in parallel.

The RPE1 cells were used for the initial screen of the actin modulators and for a couple of follow-up experiments, which pointed to the role of CDC42 in the cilia shortening and ectocytosis. After we established the MEFs from the Bbs4 KO mice (Tsyklauri et al. PMID: 33426789), we reproduced the key findings from the RPE1 model and switched to MEFs for the additional experiments, because MEFs, but not RPE1 cells, are responsive to Hedgehog signaling at the transcriptional level (Ptch, Gli1 expression) in our hands. As no GPR161 signaling experiments are done on RPE1 cells, we believe that staining for GPR161 in RPE1 cells is not required and would not add much to the manuscript. We believe that using one cell line for the initial discovery, reproducing the initial discovery in another line, which is more suitable for signaling experiments, and then doing the follow-up experiments using this second line is a standard approach in the field. We will clarify this in the revised version of the manuscript.

*We are aware of the potential overexpression artifacts, therefore the non-time-lapse experiments with KO MEFs in the manuscript rely on the staining of endogenous GPR161 and ARL13B (Figure 1 and 2 in the manuscript). The only exception is the experiment showing the GPR161-mCherry foci in Figure 4A, but this experiment largely overlaps with the experiments shown in Figure 2F-I, in which we stained the endogenous GPR161 and obtained essentially very similar results. Nevertheless, we will repeat the experiment in Figure 4A with endogenous GPR161, which would provide a side-by-side comparison of the endogenous and exogenous GPR161 behavior under the same conditions. In addition, we will quantify the foci using the endogenous ARL13B in the MEFs, and include these data to corroborate the *in vivo* imaging analysis in Figure 5 of the manuscript.*

- based on their present and previous works, the authors claim that the mechanism they have identified account for BBSome KO conditions. However, in the present study, they focused on BBS4

and the results presented in Fig.1 indicate that ML141 does not significantly increase ciliary length in BBS1 KO cells. The authors must clarify this point.

Although the rescue of the cilium length by CDC42 inhibition in the BBS1 KO cells is not as impressive as in the BBS4 KO cells, it is still statistically significant and going in the same direction (unlike in WT cells where the CDC42 inhibition slightly reduces the cilia length). We will include additional data in the revised manuscript showing that RPE1 cells lacking BBS9 exhibit the same phenomenon (Figure 4 here), indicating that this effect applies generally to the BBSome-deficiency.

Specific points:

Fig.1C: Arrows are supposed to point to "bulge-like structures" but these structures do not appear clearly in the provided pictures. A ciliary membrane marker may be used instead of acetylated-alpha-tubulin (Ac-Tub). This phenotype should be quantified.

The micrographs showing the cilia visualized using the expansion microscopy are supposed to illustrate the morphology of cilia in RPE1 cells and the presence of bulges. We have to note that this approach is not quantitative since the acquisition time of the whole volume of a single expanded cilium is quite long. We quantified the frequency of the bulges/foci as requested by this Ref. using the Ac-tub staining (Figure 5 here). You can see that the AcTub foci are more frequent in the BBS1 or BBS4 KO cells, but the statistical power is too low at this number of analyzed cilia.

Initially, we tried the ARL13B staining, but the signal was rather weak, especially in the BBS KO cells.

We will adjust the levels in micrographs in the Figure 1 to visualize the bulge-like structures more clearly. We will also include the quantification of the bulge-like structures in the revised manuscript as supplementary data. Moreover, we will revise the manuscript to make clear that this is more an illustration than a quantitative approach.

Figure 5. Plot shows the frequency of the AcTub foci we observed at the ciliary axoneme visualized using the expansion microscopy approach.

Fig.1D: The use of Ac-Tub staining to measure ciliary length could be problematic in case of acetylation defects. Did the authors double check using ARL13B staining? Similarly, on which staining was based the quantification of ciliary length in Fig.1J?

We measured the cilium length by staining the Ac-Tub only in the initial screen of the specific effects of the RHO GTPase inhibitors (Fig. 1D in the manuscript). In the follow-up experiments, we measured the cilia length using either the ARL13B staining (including Fig. 1J in the manuscript) or the overlay of

the GPR161 and Ac-tub signal when assessing the effect of SAG treatment. We will clarify this in the respective sections of the revised manuscript.

Fig.1H and Fig.1J: Statistics are lacking for the comparison between WT vs BBS4 KO (DMSO).

The statistical comparison between WT and KO will be added (Figure 6 here).

Figure 6. Revision of the panel H-left and J-right of the Figure 1 in the manuscript. Paired t test.

Fig.2A: The ciliary GPR161 staining is very faint in WT cells. If this is the case it might have masked the detection of ciliary foci in WT cells. ARL13B should be used too (see above).

As mentioned above, we will perform this additional experiment.

Fig.2C: Statistics between WT and BBS4 KO nt must be provided to see if there is a difference in the GPR161 foci.

The statistical comparison between WT and KO will be added (Figure 7 here).

Figure 7. Statistical analysis of the frequency of the GPR161 foci under the different conditions shown in Figure 2C of the manuscript.

Fig.2F: Enlarged views of cilia should also be shown for WT cells similarly as for BBS4 KO cells.

We will show this, as suggested (Figure 8 here).

Figure 8. Correction of the panel F in the Figure 2 in the manuscript.

Fig.2G: Effect of WT and DN CDC42 ciliary expression should be used to address the points raised in Fig.1? What could be their effects on ciliary length (WT vs BBS4 KO)?

The experiment in Fig. 2G shows that the expression of the DN CDC42 ciliary mutant prolongs the cilia length in the Bbs4KO, but not in WT, cells. These results very nicely correspond to the results with the chemical inhibition of CDC42 in Fig. 1F and J. The major conclusion here is that the chemical as well as genetic inhibition of CDC42 has the same effect, which largely excludes possible off-target effects. Moreover, it indicates that it is the ciliary pool of CDC42 which is responsible for the ciliary shortening in BBS4KO cells.

We will emphasize this in the revised version of the manuscript.

Fig.3: Does increased CDC42 activity result in ectosome and/or foci formation?

Increased CDC42 activity was observed in Bbs4KO cells treated with SAG results (Fig. 3C-D). The same condition leads to slightly increased GPR161 foci frequency (Fig. 2C). However, blocking the CDC42 activity by expressing DN CDC42 variant increases the foci frequency, whereas the expression of WT CDC42 reduces the foci frequency (Fig. 3H), indicating that these foci are removed by ectocytosis in a CDC42-dependent manner. We will also include time-lapse movies showing that the foci formation precedes ectocytosis (Figure 2 here).

GPR161mCherry *Bbs4* KO

ectocytosis #1

ectocytosis #2

Figure 2, as above. Examples of the ectosome release (cyan circles) from the cilia upon SHH induction in the *BBS4* deficient MEFs expressing GPR161-mCherry. Series of frames acquired every second.

Overall, our data suggest that the increase of CDC42 activity is triggered as an alternative pathway to remove signaling ciliary GPCRs, when they cannot be removed by the retrograde transport as is the case for SAG-treated *Bbs4*KO cells. Active CDC42 does not induce the foci, but rather removes the foci by ectocytosis.

If this question pointed at whether we see a correlation between CDC42 activity and foci on a level of single cilia in Fig. 3C-D, we cannot say, as we could not include an additional fluorescence channel without interfering with the analysis of the FRET NPHP3-Raichu-CDC42 probe. However, we speculate that the cilia with high CDC42 activity undergoing ectocytosis. We will address this by reanalysing our data to correlate the cilia length with the CDC42 reporter activity.

Fig.4A: Experiments have been performed in *BBS4* KO MEFs but what about ciliary distribution of overexpressed GPR161 in WT MEF? Do these foci ever formed ectosomes (live cell analyses in 4C)?

We will add the respective images and quantifications of the WT MEFs as we have these data (Figure 9 here). We will provide the images from the time-lapse experiments documenting that the foci formation precedes the ectosome release (Figure 2 here, above).

Figure 9. Micrographs and quantification of the GPR161-mCherry foci both in the WT and *Bbs4* deficient MEFs.

Fig. 5: As mentioned above, results in Fig.5 were generated overexpressing ARL13B. It would be important to quantify foci for endogenous ARL13B similarly as for GPR161. Again ectosome production was not quantified.

As mentioned above, we will perform the endogenous ARL13B foci analysis to corroborate the live cell data. As mentioned above, we will isolate and quantify the ectosomes.

Referee #3:

The manuscript by Prasai et al. entitled "BBSome-deficient cells activate intraciliary CDC42 to trigger actin-dependent ciliary ectocytosis" aims to reveal CDC42 as a key driver of ciliary ectocytosis. The authors describe that CDC42 triggers actin polymerization, ciliary ectocytosis, and cilia shortening in some BBSome-deficient cells, whereas inhibition of CDC42 in BBSome-deficient cells decreases ciliary actin polymerization.

The authors have extensively studied the role of CDC42 in BBSome-deficient cells but the role of ciliary CDC42 under physiological conditions in WT cells remains poorly understood. Also, the specificity in the cilium vs. cytoplasm is not clear for all experiments.

We thank to the Referee for carefully reading our manuscript and for their comments. The roles of RHO GTPases and actin in cilium are only emerging. Our study mostly focus on BBS-deficient cells for two major reasons. First, we studied these cells to better understand the pathological mechanisms of the Bardet-Biedl Syndrome. Second, we believed that using cells with defective retrograde ciliary transport of GPCRs is a good start to understand the general mechanisms of actin regulation in the cilia. As it has been shown before, some GPCRs do not have the retrieval sequence and are not removed from the cilia by the retrograde transport even in healthy cells (Nager et al. 2017 PMID: 28017328). These receptors, such as neuronal NPY2R, are removed by ectocytosis in WT cells in a manner, which recapitulates ectocytosis of other GPCRs in cells with their defective retrieval (Nager et al. 2017 PMID: 28017328). For this reason, it is very likely that CDC42 is involved in the removal of these triggered receptors via ectocytosis even in WT cells. Of course this warrants further investigation. We will discuss this point in the revised version of the manuscript.

Our initial experiments were performed using a chemical inhibitor of CDC42, which is obviously not selective between cytoplasmic and ciliary CDC42. This was the reason, why we used the DN CDC42 which is targeted to the cilia. Since the results of the general CDC42 inhibition and the genetic cilia-specific CDC42 inhibition lead to very similar observations, we can conclude that both of them induce the observed phenotypes primarily via the inhibition of the ciliary CDC42. Moreover, our cilia-specific probe enabled us to observe the induction of the CDC42 activity in the cilia in the BBS KO cells treated with SAG. This is in line with the CDC42-inhibition experiments, which show the strongest phenotype also under these conditions. We will investigate the cytoplasm vs. cilia specificity during the revision by addressing the specific comments of this Ref.

Before being considered for publication, the following points need to be addressed:

Major issues

- Sheu et al., Cell 2022 demonstrated that ciliary 5-HTR6 stimulation activates a non-canonical Gαq/11-RhoA pathway, which modulates nuclear actin polymerization. The authors need to test whether nuclear actin is also remodeled in their context. Furthermore, the literature also needs to be mentioned and discussed in the introduction and discussion

We are thankful for mentioning the impressive study by Sheu et al. We will discuss this study in the revised manuscript. We believe that although the remodeling of the nuclear actin via 5-HTR6 ciliary signaling is an interesting research direction, it is beyond the scope of our manuscript as we mostly focus on cilia-proximal events and GPR161 signaling. However, to answer this question, we did not observe any apparent differences of the signal in the nucleus in the experiments, where we performed the F-actin staining in various conditions.

- Figure 1G: what is shown in the right column? WT only on the left?

Fig. 1G is an illustration how the cilia with and without ARL13B foci look like. The primary read-out of this experiment is the quantification shown in Figure 1H. Both panels of Fig. 1G show the WT condition. We agree that the current labelling is confusing, although we do not consider this imperfection as a major issue with the manuscript. Anyway, we will change this in the revised version of the manuscript to make it clear, see the Figure 10 here.

Figure 10. Correction of the panel G in the Figure 1 in the manuscript.

- Always show mean or median +/- S.D. (e.g., Figure 1H, 2B, 2C etc.)

Will be done accordingly, see the examples below, Figure 11 here.

Figure 11. The suggested corrections, the plots will show mean, SD and individual data points. From left Figure 1H, 2B, 2C.

- Ref. 31 has only shown this connection for renal epithelial cells. The authors should also show experimentally in WT MEFs that CDC42 play a dual role and also controls ciliogenesis through the exocyst context. This should also be analyzed in Bbs4-KO cells to reveal that reduced ciliary length under basal conditions is not due to CDC42 controlling exocyst activity.

We agree with the Ref. that this hypothesis is worth further investigation. We will use the exocyst inhibitor (endosydin 2) to assess how the exocyst inhibition alters the cilia length both in WT and Bbs4 KO MEFs +/- SAG treatment. This will clarify the role of exocyst in our cell line models in the context of the BBSome-dependent ciliary trafficking.

- Figure 2C: The frequency of GPR161-specific foci at the ciliary tip is not different between SAG and non-treated cells in Bbs4-KO cells but the text gives different interpretation. Please rephrase.

Thank you for spotting this error. We will rephrase it accordingly.

- Figure 2E: Why are Bbs4-KO cells Shh-responsive with respect to ciliary length? According to the GPR161 data, GPR161 accumulates in the cilium and does not exit the cilium in Bbs4-KO cells.

We thank the reviewer for this point, we will clarify and discuss this in the text. We observed cilia shortening in the Bbs4 KO cells treated with SAG, which was blocked by the chemical inhibition of CDC42 (Fig. 2E) or by the expression of its genetic inhibitor, DN CDC42 (Fig. 2G). We did not observe the accumulation of GPR161, as we did not quantify the GPR161 staining intensity in these cells. We observed increased foci formation (Fig. 2C, Fig. 4B). In the revised manuscript, we will include data showing that the foci formation precedes the ectocytosis. This is also in line with the previous study, which showed cilia shortening induced by somatostatin treatment of the BBS3/ARL6 deficient cells (Nager et al. 2017 PMID: 28017328). The axoneme shortening/destabilization is caused by the ectosome formation and its subsequent shedding from the ciliary membrane.

Why we did not see clear depletion of GPR161 in SAG-treated Bbs4 KO cells? The removal of the receptors by ectocytosis is probably much less quantitative than the retrograde transport (Fig. 2I, Nager et al. 2017 PMID: 28017328). Thus, new GPR161 molecules are imported to the cilia, which could replenish the ectocytosed GPR161. It seems that CDC42 might be a limiting factor for this process as the expression of cilia-targeted WT CDC42 enhances the ectocytosis, which is manifested as observed depletion of GPR161 foci as well as decreased frequency of GPR161+ cilia (Fig. 2H-I), whereas the expression of DN CDC42 has the opposite effect. We will clarify this in the revised manuscript.

- Figure 2H: The comparison to WT (non-transfected vs. DN expressing) cells for the GPR161 foci analysis is missing. This needs to be included to reveal whether ciliary CDC42 also plays a role in WT cells.

We have these data (Figure 12 here) and we will include them in the revised version of the manuscript. The data are consistent with the model that GPR161 is removed from the cilia via the BBSome-dependent retrograde transport in SAG-treated WT cells and thus, CDC42 is dispensable for this process there.

Figure 12. Analysis of the GPR161 foci in the WT cells expressing either WT or DN variant of CDC42, experiment

in Figure 2, panels H-I.

- Figure 3C/D: to demonstrate the specificity of the sensor and put the SAG responses into context, the authors need to apply the CDC42 inhibitor to demonstrate the dynamic range of the sensor in the cilium. Furthermore, in Figure 3C, only one cilium shows a higher CDC42 activity under basal conditions in Bbs4-KO cells. Also here, the relative difference upon addition of the inhibitor should be determined to see larger changes in Bbs4-KO cilia compared to WT cilia

We are thankful for suggesting this validation experiment. We will perform the FRET-FLIM analysis using the ML141 inhibitor.

As can be seen in the quantification of the CDC42 ciliary activity (Fig. 3D), there is a variation between individual cilia in the Bbs4 KO cells upon SAG treatment. We included images of three cilia per condition to provide a fair and representative documentation of our observation. Overall, there is a significant difference between WT and Bbs4 KO cells under the basal conditions, although the difference is not as pronounced as in the case of SAG treatment (Fig. 3D). We speculate that those cilia with high CDC42 activity might be those undergoing ectocytosis. As mentioned above, we will address it by correlating the cilia length with CDC42 reporter activity.

- Figure 4A/B: perform comparable experiments in WT cells.

We have these data, which will be included in the revised manuscript. We will also add the representative micrographs, see the Figure 9 here.

Figure 9, same as above. Micrographs and quantification of the GPR161-mCherry foci both in the WT and Bbs4 deficient MEFs.

- Figure 4C: according to the text, this figure shows no difference in WT ciliary length upon SAG treatment whereas Figure 2G shows a significant decrease in ciliary length. Please comment on these differences! This makes the interpretation of Figure 4C for Bbs4-KO cells rather difficult.

The cilia shortening triggered by SAG in WT MEFs is probably caused by removal of specific receptors, including the GPR161, from cilia via endocytosis. In addition, the SHH pathway is mitogenic and its activation can stimulate cilia resorption at the ciliary pocket. Our observations thus suggest that SAG induced cilia shortening can be counteracted by overloading the ciliary compartment either generally with cargo/membrane material or specifically with GPR161 receptor acting inhibitory to SHH activation.

Our over-expression system thus allows us to neglect the contribution of endocytosis to cilia shortening (Saito et al 2017 PMID: 28607034) and offers an advantage to examine the effect of CDC42 inhibition specifically on the cilia shortening mediated by ectocytosis.

We will comment on this apparent discrepancy and explain our observations in the revised version of the manuscript.

- Figure 5A: from the pictures, it looks like that the actin dynamics in the cytosol are much more pronounced in Bbs4-KO cells upon CDC42 inhibition that in WT cells. The authors need to perform these experiments using the ciliary targeted DN-CDC42 to demonstrate specificity.

During the time lapse, the LifeAct signal bleaches over time. The presented panels show the LifeAct in different time points after the beginning of the image acquisition. For the presentation purposes, the LifeAct signal levels are adjusted so that the F-actin is visible in the later time points. Together with cell to cell variability (which we observed using LifeAct or phalloidin staining), this might lead to the impression of differential dynamics of cytosolic F-actin. However, we did not observe any dramatic differences in actin dynamics in WT versus Bbs4 KO cells, which can be seen in the additional movies, shown as supplementary files. We will provide the original sequences and also the ones with boosted LifeAct intensities and comment on the overall F-actin behaviour in the text.

In addition, we will validate these data by performing analogical experiments using the cilia-targeted DN-CDC42.

- The authors have not demonstrated ectocytosis of a GPCR in BBSome-deficient cells. As the regulation of ectocytosis is the key finding of the manuscript, the authors need to demonstrate that one of the GPCRs, which is rather absent from primary cilia in BBSome-deficient cells, is exported via ectocytosis.

The constitutive (signal-independent) ectocytosis of GPCRs (SSTR3) in BBS mutants has been already demonstrated by the pivotal work in the field of the ciliary ectocytosis: Nager et al. 2017 PMID: 28017328. We will emphasize it in the revised manuscript.

We will strengthen this part of our manuscript by adding additional results. As mentioned above, we will provide movies showing that the formation of GPR161 foci precedes the ectocytosis (Figure 2 here) to demonstrate that GPR161 indeed gets ectocytosed.

GPR161mCherry Bbs4 KO

ectocytosis #1

ectocytosis #2

Figure 2, as above. Examples of the ectosome release (cyan circles) from the cilia upon SHH induction in the BBS4 deficient MEFs expressing GPR161-mCherry. Series of frames acquired every second.

As mentioned above, we will isolate and quantify ectosomes in the culture media of different cell lines and we will analyze their content with the emphasis on the GPCRs.

Minor issues

- Add Forsythe and Beales, 2013 to reference 7

We will correct this.

- Figure 1(A): stained with ...antibody. (The word antibody is missing). Change throughout the text.

We will correct this.

- Discussion: "Using cilia-targeted dominant negative CDC42 mutant, we revealed that intraciliary CDC42 triggers ciliary ectocytosis that controls the overall ciliogenesis process in the BBSome-deficient cells." Rephrase as overall ciliogenesis might also be regulated by cytoplasmic CDC42 activity (see above).

We will rephrase it.

- Discussion: following sentences - the authors have not shown that only ciliary and not cytoplasmic CDC42 results in cilia shortening in BBSome-deficient cells (also see questions above).

We will discuss the potential role(s) of cytoplasmic CDC42 and other RHO family GTPases in the cilia shortening of the BBSome-deficient cells.

- The information give in the Material & Methods part is rather limited and does not allow to fully understand the experiments performed. It would be great if the authors could elaborate a bit more how the experiments have been performed. The same applies to the figure legends, which are also rather short and lack information.

We agree with the Ref. We will extend our Material & Methods and Figure Legend sections to increase the clarity and to make it compatible with the journal policies.

Dear Dr. Huranova

Thank you for your letter asking us to reconsider our decision and invite revision of your manuscript. I apologize for my delayed response but I have now carefully read your letter and your point-by-point response to the referee concerns and preliminary revision plan.

Taking all information into account and given that you propose to address all referee concerns, we have no objection to reconsider a strengthened manuscript that is revised along the lines specified in your response. Further evidence and quantification of ectocytosis, as outlined, will be an important addition to strengthen your conclusions.

While we usually invite authors to resubmit their revised manuscript de novo, I have made a formal decision here inviting you to revise your study for potential publication in EMBO Reports. This allows a smooth and faster handling of your revised manuscript, as you can already format it according to EMBO Reports guidelines. You will also be contacted by our source data coordinator Hannah Sonntag and have time to prepare all source data while you are working on the revision.

Acceptance of the manuscript will depend on a positive outcome of a second round of review. It is EMBO Reports policy to allow a single round of revision only and acceptance or rejection of the manuscript will therefore depend on the completeness of your responses included in the next, final version of the manuscript.

We realize that it is difficult to revise to a specific deadline. In the interest of protecting the conceptual advance provided by the work, we recommend a revision within 3 months (March 19). Please discuss the revision progress ahead of this time with the editor if you require more time to complete the revisions.

I am also happy to discuss the revision further via e-mail or a video call, if you wish.

*****IMPORTANT NOTE:

We perform an initial quality control of all revised manuscripts before re-review. Your manuscript will FAIL this control and the handling will be delayed IN CASE the following APPLIES:

- 1) A data availability section providing access to data deposited in public databases is missing. If you have not deposited any data, please add a sentence to the data availability section that explains that.
- 2) Your manuscript contains statistics and error bars based on $n=2$. Please use scatter blots in these cases. No statistics should be calculated if $n=2$.

When submitting your revised manuscript, please carefully review the instructions that follow below. Failure to include requested items will delay the evaluation of your revision.*****

- 1) a .docx formatted version of the manuscript text (including legends for main figures, EV figures and tables). Please make sure that the changes are highlighted to be clearly visible.
- 2) individual production quality figure files as .eps, .tif, .jpg (one file per figure). Please download our Figure Preparation Guidelines (figure preparation pdf) from our Author Guidelines pages <https://www.embopress.org/page/journal/14693178/authorguide> for more info on how to prepare your figures.
- 3) a .docx formatted letter INCLUDING the reviewers' reports and your detailed point-by-point responses to their comments. As part of the EMBO Press transparent editorial process, the point-by-point response is part of the Review Process File (RPF), which will be published alongside your paper.
- 4) a complete author checklist, which you can download from our author guidelines (<<https://www.embopress.org/page/journal/14693178/authorguide>>). Please insert information in the checklist that is also

reflected in the manuscript. The completed author checklist will also be part of the RPF.

5) Please note that all corresponding authors are required to supply an ORCID ID for their name upon submission of a revised manuscript (<<https://orcid.org/>>). Please find instructions on how to link your ORCID ID to your account in our manuscript tracking system in our Author guidelines (<<https://www.embopress.org/page/journal/14693178/authorguide#authorshipguidelines>>)

6) We replaced Supplementary Information with Expanded View (EV) Figures and Tables that are collapsible/expandable online. A maximum of 5 EV Figures can be typeset. EV Figures should be cited as 'Figure EV1, Figure EV2' etc... in the text and their respective legends should be included in the main text after the legends of regular figures.

7) Please note that a Data Availability section at the end of Materials and Methods is now mandatory. In case you have no data that requires deposition in a public database, please state so instead of refereeing to the database. See also < <https://www.embopress.org/page/journal/14693178/authorguide#dataavailability>>. Please note that the Data Availability Section is restricted to new primary data that are part of this study.

Additional information on source data and instruction on how to label the files are available <<https://www.embopress.org/page/journal/14693178/authorguide#sourcedata>>.

10) Figure legends and data quantification:
The following points must be specified in each figure legend:

- the name of the statistical test used to generate error bars and P values,
 - the number (n) of independent experiments (please specify technical or biological replicates) underlying each data point,
 - the nature of the bars and error bars (s.d., s.e.m.)
- If the data are obtained from n {less than or equal to} 5, show the individual data points in addition to the SD or SEM.
- If the data are obtained from n {less than or equal to} 2, use scatter blots showing the individual data points.

See also the guidelines for figure legend preparation:
<https://www.embopress.org/page/journal/14693178/authorguide#figureformat>

11) Our journal encourages inclusion of *data citations in the reference list* to directly cite datasets that were re-used and obtained from public databases. Data citations in the article text are distinct from normal bibliographical citations and should directly link to the database records from which the data can be accessed. In the main text, data citations are formatted as follows: "Data ref: Smith et al, 2001" or "Data ref: NCBI Sequence Read Archive PRJNA342805, 2017". In the Reference list, data citations must be labeled with "[DATASET]". A data reference must provide the database name, accession number/identifiers and a resolvable link to the landing page from which the data can be accessed at the end of the reference. Further instructions are available at <<https://www.embopress.org/page/journal/14693178/authorguide#referencesformat>>.

12) All Materials and Methods need to be described in the main text. We would encourage you to use 'Structured Methods', our new Materials and Methods format. According to this format, the Materials and Methods section should include a Reagents and

Tools Table (listing key reagents, experimental models, software and relevant equipment and including their sources and relevant identifiers) followed by a Methods and Protocols section in which we encourage the authors to describe their methods using a step-by-step protocol format with bullet points, to facilitate the adoption of the methodologies across labs. More information on how to adhere to this format as well as downloadable templates (.doc or .xls) for the Reagents and Tools Table can be found in our author guidelines: <<https://www.embopress.org/page/journal/14693178/authorguide#manuscriptpreparation>>. An example of a Method paper with Structured Methods can be found here: <<https://www.embopress.org/doi/10.15252/msb.20178071>>.

13) As part of the EMBO publication's Transparent Editorial Process, EMBO Reports publishes online a Review Process File to accompany accepted manuscripts. This File will be published in conjunction with your paper and will include the referee reports, your point-by-point response and all pertinent correspondence relating to the manuscript.

Kind regards,

Prasai et al. EMBOR-2023-58108V2-Q

Point-by-point response

Referee #1:

Prasai et al. set out to identify factors involved in ciliary ectocytosis and cilia shortening of BBSome-deficient cells. Their findings revealed that BBSome-deficient cells exhibit shorter cilia and accumulations of membrane proteins in the cilia. The hypothesis proposed by the researchers suggests that continuous ectocytosis is responsible for the observed shorter cilia in BBSome-deficient cells. Furthermore, they demonstrated that the reduction in cilia length can be rescued by inhibiting ROCK1 and CDC42 and actin polymerization factors (Cytochalasin D and ARP2/3 inhibitors). Subsequently, they established that CDC42 is hyperactivated in BBSome-deficient cells and is crucial for actin polymerization within the cilia.

The major issues:

Ectocytosis is an intriguing topic and this paper provided new findings. However, the current level of the paper lacks new mechanistic findings for publication in EMBO Reports. For example, RhoA inhibitors mediated-rescues of cilia length in BBSome-deficient cells were already reported by Victor Hernandez-Hernandez et al. 2013, Hum Mol Genet. In the same paper, they showed that BBSome regulates actin polymerization. Prasai et al. just added that an increase in actin polymerization in BBSome-deficient cells requires CDC42. The link between CDC42 and actin polymerization is already well known.

We thank to this Referee for reading and commenting on our manuscript. It seems that we failed to sufficiently explain the major point of our study, which is the newly identified mechanism behind the ciliary ectocytosis in the BBSome deficient cells. Whereas, the link between CDC42 and actin polymerization in the cytoplasm is indeed already well known, we are not aware of a single report about the role of intraciliary CDC42.

Major mechanistic advances of our study are and we stressed them in the manuscript:

- 1. We showed for the first time that actin polymerization within the cilia is regulated by CDC42, a RHO GTPase family member.*
- 2. We found that ciliary CDC42 is hyperactivated during SHH signalling in the BBSome-deficient cells.*
- 3. We showed that intraciliary CDC42 activity facilitates ectocytosis of excessive GPCRs, when their retrograde transport is defective, which occurs in ciliopathies, such as Bardet-Biedl Syndrome.*

Thus, we are submitting a first study showing that intraciliary activity of a Rho family member is substantially enhanced by defective retrograde transport, which is a hallmark of Bardet-Biedl Syndrome. We believe that altogether, these pivotal observations warrant sufficient novelty for publication in EMBO Reports. We revised our manuscript to make these advances more clear.

We are familiar with the data published by Hernandez-Hernandez et al. 2013 suggesting that the overall cellular activity of RHOA is enhanced by the BBSome deficiency. Although, Hernandez-Hernandez et al. reported differences in the cytosolic actin cytoskeleton in the BBSome-deficient cells, they did not address the actin inside the cilia at all, because the very first reports about the F-actin in the cilia appeared a couple of years later (Nager et al. 2017 PMID: 28017328, Phua et al. 2017 PMID:

28086093). Of note, there are still very few studies focusing on the ciliary actin filaments and this phenomenon is still poorly understood.

We were not able to reproduce the key observation by Hernandez-Hernandez et al. in RPE1 cells (Figure 1 here). To study the role of the RHOA pathway, Hernandez-Hernandez et al. used the ROCK1 inhibitor (downstream effector of RHOA) to rescue the cilia length in the BBSome-deficient cells. These data (Fig. 7B-E in Hernandez-Hernandez et al.) are in line with our observations (Fig. 1D-E in our manuscript), as both studies showed that ROCK1 inhibition prolongs cilia in WT as well as in the BBSome deficient cells, indicating that the effect of ROCK1 inhibition is general and not specific to the BBSome deficiency (which differs strikingly from the CDC42 inhibition). Overall, these data do not support the scenario that RHOA hyperactivation in the BBSome-deficient cells is responsible for the cilia shortening, although they do not exclude this possibility either.

The most plausible scenario is that the ROCK1 inhibition (as well as less specific cytochalasin treatment or Arp2/3 inhibition – Supplemental Fig. 1B-C in our manuscript) leads to the remodelling of the cortical actin, which is known to be involved in ciliogenesis (Pan et al. 2007 PMID: 17488776, Rangel et al. 2019 PMID: 30718762). However, in our study, we focused on the intraciliary actin, which has been addressed only by very few studies so far. We showed that overall chemical, but also ciliary-specific genetic inhibition of CDC42 activity rescues the cilia length exclusively in the BBSome deficient cells, but not in WT cells, which does support the scenario that CDC42 hyperactivation is the major cause of cilia shortening in the BBSome cells.

Figure 1. Relative levels of the GTP loaded RHOA in starved and non-starved RPE1 cells. RHO-GTP levels were measured using a commercial kit.

1- "We hypothesized that the cilia shortening in the BBSome-deficient cell lines, previously observed by others [9, 10, 12] and us [11] (Fig. 1A and B) is caused by continuous ectocytosis."

Although the hypothesis is intriguing, for it to be substantiated that constitutive ectocytosis causes the shortening of ciliary length in BBSome-deficient cells, they must present additional, independent evidence supporting this hypothesis.

The pivotal work in the field of ciliary ectocytosis (Nager et al. 2017, Cell PMID: 28017328) showed the constitutive ectocytosis in cells with defects in the retrograde ciliary transport, including ARL6 (BBS3)-deficient cells. BBS3 is a Bardet-Biedl Syndrom-associated gene), although the symptoms of BBS3-deficient patients are different than of those with the BBSome deficiency (Niederlova et al. 2019 PMID: 31283077). However, as all the Refs asked for actually showing and quantifying the ectocytosis in our cellular system, we revised our manuscript accordingly.

1. *We refer to the published work (Nager et al. 2017 PMID: 28017328).*
2. *We show the qualitative evidence (live cell imaging documenting the ectocytosis).*
3. *We isolated the cilia derived vesicles present in the medium of cultures of WT and BBSome-deficient cells. We analysed their content (GPR161, IFT88). We made use of the protocols established and used in multiple labs (Nager et al. 2017 PMID: 28017328, Volz et al. 2021 PMID: 34580290, Loukil et al. 2021 PMID: 33846249).*

2- In the absence of BBSome, when cells face difficulty in retrieving ciliary membrane proteins, they initiate an alternative pathway, such as ciliary ectocytosis, to eliminate these proteins from cilia. The mechanism that triggers this alternative pathway in BBSome-deficient cells raises intriguing questions. How does BBSome activate CDC42-via direct or indirect activation?

We are not showing that the BBSome activates CDC42. Actually, we are showing the very opposite: CDC42 is activated by the absence of the BBSome, suggesting that not the BBSome, but the BBSome-deficiency activates CDC42. After studying the current literature, the link between the BBSome-deficiency and ciliary CDC42 activation is not that surprising. It has been shown that in the cytoplasm, the GPCR signaling (namely the Gas subunit) activate PDZ-RhoGEF, the major CDC42 activator (Castillo-Kaul et al. 2020, PMID: 33023908). Intriguingly, we found out that PDZ-RhoGEF as well as CDC42 are present in the ciliary proteome data, which were previously published (Mick et al. 2015 PMID: 26585297, Kohli et al. 2017 PMID: 28710093, van Dam et al. 2019 PMID: 31095607). Thus, it is very plausible that the BBSome deficiency causing the defective retrieval of activated GPCRs (Nager et al. 2017 PMID: 28017328) leads to hyperactivation of Gas (Mukhopadhyay et al. 2013 PMID: 23332756), which activates PDZ-RhoGEF, which in turn activates intraciliary CDC42. However, there are multiple CDC42 activators present in the ciliary proteomes (Mick et al. 2015 PMID: 26585297, Kohli et al. 2017 PMID: 28710093, van Dam et al. 2019 PMID: 31095607), which might cause functional redundancies in this process and difficulties to address this experimentally. However, this scenario is supported by our data showing that GPR161 signaling increases the CDC42 activity in the cilia of BBS-deficient, but not WT cells, showing a direct link between the GPCR signaling and CDC42 activation. Of note, GPR161 is associated with the Gas subunit (Mukhopadhyay et al. 2013 PMID: 23332756).

We included this plausible explanation in the Discussion of the revised manuscript.

3- Is the accumulation of membrane proteins within cilia adequate to activate ectocytosis? I need to note here Nachury and colleagues showed that ectocytosis could not be attributed to the overexpression of ciliary membrane proteins (Nager et al. 2017). Prasai et al. might offer additional insights into these queries. Notably, they conducted experiments involving the overexpression of GPR161-mCherry in BBSome-deficient cells, suggesting a potential avenue for further exploration in their research.

First, we are not sure whether this point is meant by the Ref as an issue in the manuscript or just an indication of a potential follow-up research.

It should be noted that the overexpression of GPR161 in MEFs prolongs the cilia as we insert more material into the cilia. Moreover, the overexpression of mCherry-GPR161 doubles the frequency of the GPR161 foci increased in both WT and Bbs4 KO cells lines (from 0.1 to 0.2 in WT and from 0.2 to 0.4 in the KO) (Figure 2 here). The ectocytosis might be enhanced in these lines by the GPR161

overexpression, but probably not substantially, as the ectocytosis is not able to fully compensate for the increase in the ciliary material, which leads to the cilia prolongation.

We commented on the issue of the overexpression and ectocytosis in the revised manuscript.

Figure 2. Quantification of the GPR161-mCherry foci in the WT and Bbs4 deficient MEFs.

The minor issues:

1- there is a type mistake: buldge-like structures, it should be bulge-like structures

Thank you for spotting this. It was corrected

2- Figure 1I. the beginning of cilia should be marked, thus readers can better view the position of accumulations.

We fixed this. We mention in figure legends that the cilia base is at the bottom of the micrographs when needed.

Referee #2:

In their present manuscript, Martina Huranova and colleagues have investigated the mechanisms involved in the shortening of primary cilia observed in cells invalidated for the expression of BBSome subunits. Testing several molecules targeting Rho GTPases, they provide strong evidence that ML141, an inhibitor of CDC42, was able to increase ciliary length in BBS4 KO RPE1 cells and MEFs (mouse embryonic fibroblasts). They further investigate the role of CDC42 in BBS4 KO MEFs in the context of the activation of the Hedgehog signaling pathway. The presented data show that ML141 treatment and cilia-targeted dominant negative form of CDC42 increase ciliary length in these cells as well the SAG-induced decrease of ciliary length. Using a FRET sensor, they also nicely show increased ciliary CDC42 activity which leads to increased actin polymerization within cilia. Altogether these data indicate that the observed decreased ciliary length upon loss of BBS4 is linked to overactivation of CDC42 within cilia leading to actin polymerization ciliary ectosome formation. Altogether, the data presented are of high quality and interesting. I however have several concerns which are listed below.

We thank to the reviewer for appreciating the quality of our data, for finding our data interesting, and for the comments.

Main concerns:

- the main hypothesis raised by the authors, based on the literature and their data, is that shortening of cilia in BBS4 KO cells is linked to increased production of ciliary-derived vesicles (ectosomes) caused by abnormal activation of CDC42 and actin polymerization. It is therefore somehow surprising that ectosome release was not quantified in this study. This can be easily quantified from the live cell imaging data. It can be also quantified by purification of secreted vesicles in the different conditions. The assumption that ciliary foci of membrane proteins correlate with ectosome release should be validated.

As suggested by all reviewers, we performed ultracentrifugation of the ciliary vesicles. We optimized the protocol and made a pulse chase for 2.5 hours, which allowed us to neglect the significant contribution of the cellular EVs to the P100 in the BBSome deficient MEFs upon long starvation as shown recently (Volz et al. 2021 PMID: 34580290). We observed increased production of cilia-derived EVs in the Bbs4 deficient MEFs in comparison to WT MEFs. These data are now included into the manuscript as Figure 1K, L, M.

Additionally, we quantified the live cell imaging data acquired in the Figure 4C, because of the 1 min time frames, which allowed us to match the budding bulges to the released ectosomes from the cilium. The ectocytosis events are rather infrequent and the quantification requires a lot of time-lapse movies, therefore we consider the measurement of the length and foci as a complementary and more robust approach. We included the examples of the movies showing budding of the GPR161 from the cilia as Figure 4C (see the examples also below in Figure 3) and the quantification as 4D.

We appreciate this concern and believe that the three experimental approaches—ultracentrifugation of EVs, quantification of time-lapse movies, and cilia length and foci quantification—collectively support the main hypothesis and findings in our manuscript.

GPR161mCherry *Bbs4* KO

ectocytosis #1

ectocytosis #2

Figure 3. Examples of the ectosome release (cyan circles) from the cilia upon SHH induction in the *BBS4* deficient MEFs expressing GPR161-mCherry. Series of frames acquired every minute.

- the authors used RPE1 clones KO for BBS subunits that they have described in a previous study. They used WT RPE1 cells as a control but it is not clear if those cells correspond to the parental cell line or to "WT" clones isolated in the same time as KO clones. By the way, in their previous paper, they generated more appropriate control cell lines in which they re-expressed corresponding tagged-BBS subunit in the generated BBS KO cells. Those control cells should be used to validate that some of the phenotypes described here including ARL13B and GPR161 foci, are not linked to a single clone effect.

The WT RPE1 cell line is the parental cell line, we will adjust the manuscript to make this clear. It is very difficult to generate CRISPR/Cas9 knockouts in the RPE1 cells (Prasai et al. PMID: 32759308). We managed to generate, reconstitute and validate one KO clone for BBS1 and BBS4 genes. We have additional data showing that RPE1 cells lacking BBS9 behave in the same way as BBS4 and BBS1 KO cells and we added them to the revised manuscript (Figure 4 here). Since three KO lines lacking different BBSome subunits show the same phenotype, we believe that these data are even more convincing/robust than having three KO clones of a single BBS gene. It should be also noted that we confirmed the observations with RPE1 cells using an independent cellular model - primary WT and Bbs4 KO MEF cells (Figure 1I-J of the manuscript).

The cilia-shortening in these BBS-deficient RPE1 lines was previously shown to be caused by the BBSome-deficiency using the reconstituted lines (Prasai. et al. 2020 PMID: 32759308). However, this Ref. is right that we did not perform the CDC42 inhibitor treatments in these lines. For this reason, we performed the proposed experiments with the reconstituted BBS4 KO BBS4-YFP RPE-1 cells (Prasai et al. 2020 PMID: 32759308) as suggested and these data were included into the manuscript as Expanded View Figure 1G, H.

Figure 4. Correction of the panel F of the Figure 1 in the manuscript.

- It is quite surprising that, while RPE1 KO cells were used in Fig.1 to identify the effect of ML141 on ciliary length, they were not used thereafter, the authors shifted to KO MEFs. It would be important to use the two cell models in parallel to strengthen the data. In addition, ciliary tip foci which are considered by the authors as forming ectosomes, were quantified using ARL13B staining in RPE1 cells while GPR161 was used in MEFs. Quantifications should be done for these two markers in the two cell lines. By the way, the authors also used overexpressed GPR161 or ARL13B in KO MEFs (Figs 4 and 5) which is problematic based on the known effects of the overexpression of ciliary GPCRs and small GTPases on ciliary length. Except in case of live cell studies, the use of stainings of endogenously expressed proteins must be done in parallel.

The RPE1 cells were used for the initial screen of the actin modulators and for a couple of follow-up experiments, which pointed to the role of CDC42 in the cilia shortening and ectocytosis. After we established the MEFs from the Bbs4 KO mice (Tsyklauri et al. PMID: 33426789), we reproduced the key findings from the RPE1 model and switched to MEFs for the additional experiments, because MEFs, but not RPE1 cells, were responsive to Hedgehog signaling at the transcriptional level (Ptch, Gli1 expression) in our hands. As no GPR161 signaling experiments are done on RPE1 cells, we believe that staining for GPR161 in RPE1 cells is not required and would not add much to the manuscript. We believe that using one cell line for the initial discovery, reproducing the initial discovery in another line, which is more suitable for signaling experiments, and then doing the follow-up experiments using this second line is a standard approach in the field. We clarified this in the revised version of the manuscript.

We are aware of the potential overexpression artifacts, therefore the non-time-lapse experiments with KO MEFs in the manuscript rely on the staining of endogenous GPR161 and ARL13B (Figure 1 and 2 in the manuscript). The only exception is the experiment showing the GPR161-mCherry foci in Figure 4A, but this experiment largely overlaps with the experiments shown in Figure 2F-I, in which we stained the endogenous GPR161 and obtained essentially very similar results. Nevertheless, we quantified the foci using the endogenous ARL13B in the MEFs, and included these data into the manuscript as Expanded View Figure 2A, B.

- based on their present and previous works, the authors claim that the mechanism they have identified account for BBSome KO conditions. However, in the present study, they focused on BBS4 and the results presented in Fig.1 indicate that ML141 does not significantly increase ciliary length in BBS1 KO cells. The authors must clarify this point.

Although the rescue of the cilium length by CDC42 inhibition in the BBS1 KO cells is not as impressive as in the BBS4 KO cells, it is still statistically significant and going in the same direction (unlike in WT cells where the CDC42 inhibition slightly reduces the cilia length). We included additional data in the revised manuscript showing that RPE1 cells lacking BBS9 exhibit the same phenomenon (Figure 4 here), indicating that this effect applies to the deficiency of any of the three BBSome subunits. The reason why this effect was not very pronounced in the BBS1-deficient line as in BBS4 or BBS9 deficient lines could be biological differences between these three proteins, technical variation between the three lines, or stochasticity. In any case, these results support our overall conclusions.

Specific points:

Fig.1C: Arrows are supposed to point to "bulge-like structures" but these structures do not appear clearly in the provided pictures. A ciliary membrane marker may be used instead of acetylated-alpha-tubulin (Ac-Tub). This phenotype should be quantified.

The micrographs showing the cilia visualized using the expansion microscopy are supposed to illustrate the morphology of cilia in RPE1 cells and the presence of bulges. We have to note that this approach is not quantitative since the acquisition time of the whole volume of a single expanded cilium is quite long. We quantified the frequency of the bulges/foci as requested by this Ref. using the Ac-tub staining (Figure 5 here). You can see that the AcTub foci are more frequent in the BBS4 KO cells, but the statistical power is too low at this number of analyzed cilia.

Initially, we tried the ARL13B staining, but the signal in the expansion microscopy was rather weak, especially in the BBS KO cells.

We adjusted the levels in micrographs and used different LUT in ImageJ in the Figure 1 to visualize the bulge-like structures more clearly. We also made enlarged insets showing mainly the ciliary tip. We included the quantification of the bulge-like structures in the revised manuscript as Expanded View Figure 1B. Moreover, we revised the manuscript to make clear that this is more an illustration than a quantitative approach.

Figure 5. Plot shows the frequency of the AcTub foci we observed at the ciliary axoneme in RPE1 cells visualized using the expansion microscopy approach.

Fig.1D: The use of Ac-Tub staining to measure ciliary length could be problematic in case of acetylation defects. Did the authors double check using ARL13B staining? Similarly, on which staining was based the quantification of ciliary length in Fig.1J?

We measured the cilium length by staining the Ac-Tub only in the initial screen of the specific effects of the RHO GTPase inhibitors (Fig. 1D in the manuscript). In the follow-up experiments, we measured

the cilia length using either the ARL13B staining (including Fig. 1J in the manuscript) or the overlay of the GPR161 and Ac-tub signal when assessing the effect of SAG treatment. We clarified this in the respective sections of the revised manuscript.

Fig.1H and Fig.1J: Statistics are lacking for the comparison between WT vs BBS4 KO (DMSO).

We added this statistical comparison between WT and KO as requested (Figure 6 here).

Figure 6. Revision of the panel H-left and J-right of the Figure 1 in the manuscript.

Fig.2A: The ciliary GPR161 staining is very faint in WT cells. If this is the case it might have masked the detection of ciliary foci in WT cells. ARL13B should be used too (see above).

We performed this additional experiment, which is now included into the manuscript as Expanded View Figure 2A, B.

Fig.2C: Statistics between WT and BBS4 KO nt must be provided to see if there is a difference in the GPR161 foci.

The statistical comparison between WT and KO was provided (Figure 7 here).

Figure 7. Statistical analysis of the frequency of the GPR161 foci under the different conditions shown in Figure 2C of the manuscript.

Fig.2F: Enlarged views of cilia should also be shown for WT cells similarly as for BBS4 KO cells.

We included these views, as suggested (Figure 8 here).

Figure 8. Correction of the panel F in the Figure 2 in the manuscript.

Fig.2G: Effect of WT and DN CDC42 ciliary expression should be used to address the points raised in Fig.1? What could be their effects on ciliary length (WT vs BBS4 KO)?

The experiment in Fig. 2G shows that the expression of the DN CDC42 ciliary mutant prolongs the cilia length in the Bbs4KO, but not in WT, cells. These results very nicely correspond to the results with the chemical inhibition of CDC42 in Fig. 1F and J. The major conclusion here is that the chemical as well as genetic inhibition of CDC42 has the same effect, which largely excludes possible off-target effects. Moreover, it indicates that it is the ciliary pool of CDC42 which is responsible for the ciliary shortening in BBS4KO cells.

We emphasized this in the revised version of the manuscript.

Fig.3: Does increased CDC42 activity result in ectosome and/or foci formation?

Increased CDC42 activity was observed in Bbs4 KO cells treated with SAG results (Fig. 3C-D). The same condition leads to slightly increased GPR161 foci frequency (Fig. 2C). However, blocking the CDC42 activity by expressing DN CDC42 variant increases the foci frequency, whereas the expression of WT CDC42 reduces the foci frequency (Fig. 3H), indicating that these foci are removed by ectocytosis in a CDC42-dependent manner. We will also include time-lapse movies showing that the foci formation precedes ectocytosis (Figure 2 here).

GPR161mCherry *Bbs4* KO

ectocytosis #1

ectocytosis #2

Figure 2, as above. Examples of the ectosome release (cyan circles) from the cilia upon SHH induction in the *BBS4* deficient MEFs expressing GPR161-mCherry. Series of frames acquired every second.

Overall, our data suggest that the increase of CDC42 activity is triggered as an alternative pathway to remove signaling ciliary GPCRs, when they cannot be removed by the retrograde transport as is the case for SAG-treated *Bbs4* KO cells. Active CDC42 does not induce the foci, but rather removes the foci by ectocytosis.

If this question pointed at whether we see a correlation between CDC42 activity and foci on a level of single cilia in Fig. 3C-D, we cannot say, as we could not include an additional fluorescence channel without interfering with the analysis of the FRET NPHP3-Raichu-CDC42 probe. However, we reanalyzed our data to correlate the cilia length with the CDC42 reporter activity both in WT and *Bbs4* KO cells. We found that the shorter cilia in *BBSome* deficient cells undergoing ectocytosis exhibit higher FLIM-FRET thus higher CDC42 activity. The correlation analysis is now included into the manuscript as Figure 3F.

Fig.4A: Experiments have been performed in *BBS4* KO MEFs but what about ciliary distribution of overexpressed GPR161 in WT MEF? Do these foci ever formed ectosomes (live cell analyses in 4C)?

We added the respective images and quantifications of the WT MEFs (Figure 9 here). We provided the representative images from the time-lapse experiments documenting that the foci formation precedes the ectosome release (Figure 2 here, above).

Figure 9. Micrographs and quantification of the GPR161-mCherry foci both in the WT and *Bbs4* deficient MEFs.

Fig. 5: As mentioned above, results in Fig.5 were generated overexpressing ARL13B. It would be important to quantify foci for endogenous ARL13B similarly as for GPR161. Again ectosome production was not quantified.

As mentioned above, we quantified the frequency of endogenous ARL13B foci to corroborate the live cell imaging assays and included these data in the manuscript as Expanded View Figure 2A, B. Additionally, we isolated and quantified the ectosomes under the tested conditions, now included in the manuscript as Figure 1K, L, M.

Referee #3:

The manuscript by Prasai et al. entitled "BBSome-deficient cells activate intraciliary CDC42 to trigger actin-dependent ciliary ectocytosis" aims to reveal CDC42 as a key driver of ciliary ectocytosis. The authors describe that CDC42 triggers actin polymerization, ciliary ectocytosis, and cilia shortening in some BBSome-deficient cells, whereas inhibition of CDC42 in BBSome-deficient cells decreases ciliary actin polymerization.

The authors have extensively studied the role of CDC42 in BBSome-deficient cells but the role of ciliary CDC42 under physiological conditions in WT cells remains poorly understood. Also, the specificity in the cilium vs. cytoplasm is not clear for all experiments.

We thank to the Referee for carefully reading our manuscript and for their comments. The roles of RHO GTPases and actin in cilium are only emerging. Our study mostly focus on BBS-deficient cells for two major reasons. First, we studied these cells to better understand the pathological mechanisms of the Bardet-Biedl Syndrome. Second, we believed that using cells with defective retrograde ciliary transport of GPCRs is a good start to understand the general mechanisms of actin regulation in the cilia. As it has been shown before, some GPCRs do not have the retrieval sequence and are not removed from the cilia by the retrograde transport even in healthy cells (Nager et al. 2017 PMID: 28017328). These receptors, such as neuronal NPY2R, are removed by ectocytosis in WT cells in a manner, which recapitulates ectocytosis of other GPCRs in cells with their defective retrieval (Nager et al. 2017 PMID: 28017328). For this reason, it is very likely that CDC42 is involved in the removal of these triggered receptors via ectocytosis even in WT cells. Of course this warrants further investigation. We will discuss this point in the revised version of the manuscript.

Our initial experiments were performed using a chemical inhibitor of CDC42, which is obviously not selective between cytoplasmic and ciliary CDC42. This was the reason, why we used the DN CDC42 which is targeted to the cilia. Since the results of the general CDC42 inhibition and the genetic cilia-specific CDC42 inhibition lead to very similar observations, we can conclude that both of them induce the observed phenotypes primarily via the inhibition of the ciliary pool of CDC42. Moreover, our cilia-specific FRET probe enabled us to observe the induction of the CDC42 activity in the cilia in the BBS KO cells treated with SAG. This is in line with the CDC42-inhibition experiments, which show the strongest phenotype also under these conditions. We investigated the cytoplasm vs. cilia specificity during the revision by addressing the specific comments of this Ref.

Before being considered for publication, the following points need to be addressed:

Major issues

- Sheu et al., Cell 2022 demonstrated that ciliary 5-HTR6 stimulation activates a non-canonical Gαq/11-RhoA pathway, which modulates nuclear actin polymerization. The authors need to test whether nuclear actin is also remodeled in their context. Furthermore, the literature also needs to be mentioned and discussed in the introduction and discussion

We are thankful for mentioning the impressive study by Sheu et al. We will discuss this study in the revised manuscript. We believe that although the remodeling of the nuclear actin via 5-HTR6 ciliary signaling is an interesting research direction, it is beyond the scope of our manuscript as we mostly focus on cilia-proximal events and GPR161 signaling. However, to answer this question, we did not observe any apparent differences of the signal in the nucleus in the experiments, where we performed the F-actin staining in various conditions

- Figure 1G: what is shown in the right column? WT only on the left?

Fig. 1G is an illustration how the cilia with and without ARL13B foci look like. The primary read-out of this experiment is the quantification shown in Figure 1H. Both panels of Fig. 1G show the WT condition. We agree that the current labelling is confusing, although we do not consider this imperfection as a major issue with the manuscript. We changed this in the revised version of the manuscript to make it clear, Figure 10 here.

Figure 10. Correction of the panel G in the Figure 1 in the manuscript.

- Always show mean or median +/- S.D. (e.g., Figure 1H, 2B, 2C etc.)

Was done accordingly, see the examples below, Figure 11 here.

Figure 11. The suggested corrections, the plots will show mean, SD and individual data points. From left Figure 1H, 2B, 2C.

- Ref. 31 has only shown this connection for renal epithelial cells. The authors should also show experimentally in WT MEFs that CDC42 play a dual role and also controls ciliogenesis through the exocyst context. This should also be analyzed in Bbs4-KO cells to reveal that reduced ciliary length under basal conditions is not due to CDC42 controlling exocyst activity.

We agree with the Ref. that this hypothesis is worth further investigation. However, we emphasize that a thorough examination of the role of exocyst in cargo import and export, and its impact on ectocytosis, is beyond the scope of this manuscript. Instead, we conducted a preliminary investigation to see if cilia in MEFs respond similarly to exocyst inhibition in steady state, as observed in renal cells

in the study we referred to in the manuscript. Using the exocyst inhibitor endosidin 2, we observed a mild decrease in cilia length in WT MEFs, unlike the significant effects seen in IMCD3 cells. In *Bbs4* KO MEFs, no effect on cilia length was observed, Figure 12 here. Given that this area is still underexplored, we consider these findings as preliminary data for future research. Therefore, we suggest moving this hypothesis to the discussion section of the manuscript and not including these data in the main results.

Figure 12. A) Representative micrographs and B) analysis of the cilia length in the WT and *Bbs4* MEFs upon treatment with 200 μ M endosidin2 (EndoS2) and DMSO as vehicle for 2h. Cilia were visualized via staining with antibodies to ARL13B (green) and acetylated tubulin (Ac-tub, red). Scale bar, 5 μ m. Medians with interquartile range from three independent experiments ($n = 119 - 131$ cilia). Statistical significance was calculated using the two-tailed Mann-Whitney test. Merged micrographs show nuclei staining by DAPI – blue.

- Figure 2C: The frequency of GPR161-specific foci at the ciliary tip is not different between SAG and non-treated cells in *Bbs4*-KO cells but the text gives different interpretation. Please rephrase.

Thank you for spotting this error. We rephrased this accordingly.

- Figure 2E: Why are *Bbs4*-KO cells Shh-responsive with respect to ciliary length? According to the GPR161 data, GPR161 accumulates in the cilium and does not exit the cilium in *Bbs4*-KO cells.

*We thank the reviewer for this point, we will clarify and discuss this in the text. We observed cilia shortening in the *Bbs4* KO cells treated with SAG, which was blocked by the chemical inhibition of CDC42 (Fig. 2E) or by the expression of its genetic inhibitor, DN CDC42 (Fig. 2G). We did not observe the accumulation of GPR161, as we did not quantify the GPR161 staining intensity in these cells. We observed increased foci formation (Fig. 2C, Fig. 4B). The foci formation precedes the ectocytosis and cilia shortening, which is shown in the revised version of the manuscript. This is also in line with the previous study, which showed cilia shortening induced by somatostatin treatment of the *BBS3/ARL6* deficient cells (Nager et al. 2017 PMID: 28017328). The axoneme shortening/destabilization is caused by the ectosome formation and its subsequent shedding from the ciliary membrane.*

*Why we did not see clear depletion of GPR161 in SAG-treated *Bbs4* KO cells? The removal of the receptors by ectocytosis is probably much less quantitative than the retrograde transport (Fig. 2I, Nager et al. 2017 PMID: 28017328). Thus, new GPR161 molecules are imported to the cilia, which could replenish the ectocytosed GPR161. It seems that CDC42 might be a limiting factor for this*

process as the expression of cilia-targeted WT CDC42 enhances the ectocytosis, which is manifested as observed depletion of GPR161 foci as well as decreased frequency of GPR161+ cilia (Fig. 2H-I), whereas the expression of DN CDC42 has the opposite effect. We clarified this in the revised manuscript.

- Figure 2H: The comparison to WT (non-transfected vs. DN expressing) cells for the GPR161 foci analysis is missing. This needs to be included to reveal whether ciliary CDC42 also plays a role in WT cells.

We actually had these data and we included them now in the revised version of the manuscript (Figure 13 here). The data are consistent with the model that GPR161 is removed from the cilia via the BBSome-dependent retrograde transport in SAG-treated WT cells and thus, CDC42 is dispensable for this process there.

Figure 13. Analysis of the GPR161 foci in the WT cells expressing either WT or DN variant of CDC42, experiment in Figure 2, panels H-I.

- Figure 3C/D: to demonstrate the specificity of the sensor and put the SAG responses into context, the authors need to apply the CDC42 inhibitor to demonstrate the dynamic range of the sensor in the cilium. Furthermore, in Figure 3C, only one cilium shows a higher CDC42 activity under basal conditions in Bbs4-KO cells. Also here, the relative difference upon addition of the inhibitor should be determined to see larger changes in Bbs4-KO cilia compared to WT cilia

We are thankful for suggesting this validation experiment. We performed the FRET-FLIM analysis using the CDC42 inhibitor and we observed that the inhibitor decreases the FRET of the Raichu probe to lifetime values similar to those of the no FRET control (N-CFP-PAK-CDC42). We included these data into the manuscript as Expanded View Figure 2C.

As it can be seen in the quantification of the CDC42 ciliary activity (Fig. 3E now), there is a variation between individual cilia in the Bbs4 KO cells upon SAG treatment. We included images of three cilia per condition to provide a fair and representative documentation of our observation. Overall, there is a significant difference between WT and Bbs4 KO cells under the basal conditions, although the difference is not as pronounced as in the case of SAG treatment (Fig. 3E now). To test our hypothesis that cilia with high CDC42 activity undergo ectocytosis in Bbs4 KO cells, we correlated the cilia length with CDC42 reporter activity. These data are included into the revised version of the manuscript as Figure 3F.

- Figure 4A/B: perform comparable experiments in WT cells.

We had these data and included them in the revised manuscript. We also added the representative micrographs, see the Figure 9 here.

Figure 9, same as above. Micrographs and quantification of the GPR161-mCherry foci both in the WT and Bbs4 deficient MEFs.

- Figure 4C: according to the text, this figure shows no difference in WT ciliary length upon SAG treatment whereas Figure 2G shows a significant decrease in ciliary length. Please comment on these differences! This makes the interpretation of Figure 4C for Bbs4-KO cells rather difficult.

The cilia shortening triggered by SAG in WT MEFs is probably caused by removal of specific receptors, including the GPR161, from cilia via endocytosis. In addition, the SHH pathway is mitogenic and its activation can stimulate cilia resorption at the ciliary pocket. Our observations thus suggest that SAG induced cilia shortening can be counteracted by overloading the ciliary compartment either generally with cargo/membrane material or specifically with GPR161 receptor acting inhibitory to SHH activation.

Our over-expression system thus allows us to neglect the contribution of endocytosis to cilia shortening (Saito et al 2017 PMID: 28607034) and offers an advantage to examine the effect of CDC42 inhibition specifically on the cilia shortening mediated by ectocytosis.

We commented on this apparent discrepancy and explained our observations in the revised version of the manuscript.

- Figure 5A: from the pictures, it looks like that the actin dynamics in the cytosol are much more pronounced in Bbs4-KO cells upon CDC42 inhibition that in WT cells. The authors need to perform these experiments using the ciliary targeted DN-CDC42 to demonstrate specificity.

During the time lapse, the LifeAct signal bleaches over time. The presented panels show the LifeAct in different time points after the beginning of the image acquisition. For the presentation purposes, the LifeAct signal levels are adjusted so that the F-actin is visible in the later time points. Together with cell to cell variability (which we observed using LifeAct or phalloidin staining), this might lead to the impression of differential dynamics of cytosolic F-actin. However, we did not observe any dramatic differences in actin dynamics in WT versus Bbs4 KO cells, which can be seen in the additional movies, shown as expanded view files.

To validate the specificity of the cilia-targeted DN-CDC42 we performed time lapse imaging of the Bbs4 KO MEFs expressing either the WT (47 movies) or the DN (40 movies) version of the CDC42 targeted to cilia. We observed that the frequency of ciliary actin polymerization decreased in the cells expressing the DN version of CDC42. These data have been included into the manuscript as Fig. 5E and Fig. EV5A, B.

- The authors have not demonstrated ectocytosis of a GPCR in BBSome-deficient cells. As the regulation of ectocytosis is the key finding of the manuscript, the authors need to demonstrate that one of the GPCRs, which is rather absent from primary cilia in BBSome-deficient cells, is exported via ectocytosis.

The constitutive (signal-independent) ectocytosis of GPCRs (SSTR3) in BBS mutants has been already demonstrated by the pivotal work in the field of the ciliary ectocytosis: Nager et al. 2017 PMID: 28017328. We emphasized it in the revised manuscript.

We believe that we managed to strengthen this part of our manuscript by adding additional results. As mentioned above, we provided movies showing that the formation of GPR161 foci precedes the ectocytosis (Figure 2 here) to demonstrate that GPR161 indeed gets ectocytosed. Furthermore, we isolated the ectosomes present in the culture media and analyzed their content with the emphasis on the GPR161 and IFT88.

GPR161mCherry Bbs4 KO

ectocytosis #1

ectocytosis #2

Figure 2, as above. Examples of the ectosome release (cyan circles) from the cilia upon SHH induction in the BBS4 deficient MEFs expressing GPR161-mCherry. Series of frames acquired every second.

Minor issues

- Add Forsythe and Beales, 2013 to reference 7

We corrected this.

- Figure 1(A): stained with ...antibody. (The word antibody is missing). Change throughout the text.

We corrected this.

- Discussion: "Using cilia-targeted dominant negative CDC42 mutant, we revealed that intraciliary CDC42 triggers ciliary ectocytosis that controls the overall ciliogenesis process in the BBSome-deficient cells." Rephrase as overall ciliogenesis might also be regulated by cytoplasmic CDC42 activity (see above).

We rephrased it.

- Discussion: following sentences - the authors have not shown that only ciliary and not cytoplasmic CDC42 results in cilia shortening in BBSome-deficient cells (also see questions above).

We discussed the potential role(s) of cytoplasmic CDC42 and other RHO family GTPases in the cilia shortening of the BBSome-deficient cells.

- The information give in the Material & Methods part is rather limited and does not allow to fully understand the experiments performed. It would be great if the authors could elaborate a bit more how the experiments have been performed. The same applies to the figure legends, which are also rather short and lack information.

We agree with the Ref. We added more information to our Material & Methods. We also revised the Figure Legend section to increase the clarity and to make it compatible with the EMBO Reports journal guidelines.

Dear Dr. Huranova

Thank you for the submission of your revised manuscript to EMBO reports. I apologize for the delay in handling your manuscript but we have meanwhile received the full set of referee reports that is copied below.

As you will see, both referees find that the study has been significantly improved during revision but they also note a number of remaining concerns and consider the newly added data not fully conclusive. The referees feel that further data in support of the proposed production of ciliary-derived ectosomes/EVs need to be provided in order to make your data fully conclusive:

- Negative controls are needed to rule out that cell debris contaminates the EV isolation.
- Please check whether the short time frame of production is the cause for TSG101 not being detected.
- The specificity of the IFT88 antibody needs to be documented.
- Concerns regarding GPR161 overexpression and ciliary length that are raised by referee #2 and #3 need to be clarified.

I would like to give you the opportunity to address these remaining concerns in a second round of revision. Please also address all referee concerns in a point-by-point response.

As always, I am happy to discuss the revision further via e-mail or a video call, if you wish.

From the editorial side, there are also a number of things that need your attention:

- Your manuscript will be published in our Reports section, which requires to combine the Results and Discussion section.
 - Please reduce the number of keywords to 5.
 - Please update the 'Conflict of interest' paragraph to our new 'Disclosure and competing interests statement'. For more information see <https://www.embopress.org/page/journal/14693178/authorguide#conflictsofinterest>
 - Regarding the Author Contributions, we now use CRediT to specify the contributions of each author in the journal submission system. Therefore, please remove the Author Contributions from the manuscript file and make sure that the author contributions in our online manuscript tracking system are correct and up-to-date. The information you specified in the system will be automatically retrieved and typeset into the article. You can enter additional information in the free text box provided, if you wish.
 - The following funding sources are acknowledged in the manuscript, but missing in our online submission system: - Light Microscopy Core Facility, IMG, Prague, Czech Republic, supported by MEYS - LM2023050 and RVO - 68378050-KAV-NPUI. Please update the information in the online system, as this is the information that will be transferred to the journal homepage and PubMed.
 - Movies: file names and titles need to be Movie EV1, etc. The legends need to be removed from the manuscript file and provided in separate readme.txt files. Each movie should be zipped up together with its corresponding legend so that we have 1 folder uploaded per movie. Please note that the source file names need to have the correct naming as well - Movie EV1 etc.
 - Please add separate movie callouts in the manuscript text; currently these are just mentioned as "Movies".
 - Author Checklist: Please complete Column D by choosing the appropriate response from the pull-down menus.
 - Please change the titles in EV figure legends in the manuscript to "Figure EV1", etc. (instead of "Expanded View Figure 1").
 - Figure 1 contains a lot of information and panels and I feel that the panels are so close to each other that it is not so easy to distinguish one panel from each other and to find the panel letter on first glance. In particular for F-I and J-M. I suggest to increase the spacing between the panels somewhat, if possible.
 - Our production/data editors have asked you to clarify several points in the figure legends (see below). Please incorporate these changes in the manuscript and return the revised file with tracked changes with your final manuscript submission.
- A) Statistical test information. Only p-values that are actually shown in the figure panel(s) should (and must) be defined in the legends, all others should be removed from (or added to) the legend. Moreover, we ask for the specification of exact p-values:
- Please note that the exact p values are not provided in the legends of figures 1a, 1e-f, 1j; 2b, e, g; EV 1d, g; EV 2a, c.
- B) Replicates, error bars, and data presentation:
- Please note that the error bars are not defined in the legends of figure 4e.
 - Please note that the scale bar needs to be defined for figure 3c.

- Since July 1st we require that Materials and Methods need to be described in the main text using our 'Structured Methods' format. I know that you submitted your manuscript before that date but it would nevertheless be appreciated if you could fill and add the Reagents and Tools table, as described below.

Structured Methods: according to this format, the Methods section includes a Reagents and Tools Table (listing key reagents, experimental models, software and relevant equipment and including their sources and relevant identifiers) followed by a Methods and Protocols section describing the methods using a step-by-step protocol format. The aim is to facilitate adoption of the methodologies across labs. More information on how to adhere to this format as well as a downloadable template (.docx) for the Reagents and Tools Table can be found in our author guidelines:

- As a standard procedure, we edit the title and abstract of manuscripts to make them more accessible to a general readership. Please find the edited suggestion below my signature.

- Finally, EMBO Reports papers are accompanied online by

A) a short (1-2 sentences) summary of the findings and their significance,

B) 2-3 bullet points highlighting key results and

C) a schematic summary figure that provides a sketch of the major findings (not a data image).

Please provide the summary figure as a separate file in PNG or JPG format at a size of 550x300-600 pixels (width x height).

Please note that the size is rather small and that text needs to be readable at the final size. Please send us this information along with the revised manuscript.

With kind regards,

Martina Rembold, PhD

Senior Editor

EMBO reports

Abstract:

Bardet-Biedl syndrome (BBS) is a pleiotropic ciliopathy caused by dysfunction of the BBSome, a cargo adaptor essential for export of transmembrane receptors from cilia. Although actin-dependent ectocytosis has been proposed to compensate defective cargo retrieval, its molecular basis remains unclear, especially in relation to BBS pathology. In this study, we investigated how actin polymerization and ectocytosis are regulated within the cilium. Our findings reveal that ciliary CDC42, a RHO-family GTPase, triggers in situ actin polymerization, ciliary ectocytosis, and cilia shortening in BBSome-deficient cells. Activation of the Sonic Hedgehog pathway further enhances CDC42 activity specifically in BBSome-deficient cilia. Inhibition of CDC42 in BBSome-deficient cells decreases the frequency and duration of ciliary actin polymerization events, causing buildup of G protein coupled receptor 161 (GPR161) in bulges along the axoneme during Sonic Hedgehog signaling. Overall, our study identifies CDC42 as a key trigger of ciliary ectocytosis. Hyperactive ciliary CDC42 and ectocytosis and the resulting loss of ciliary material might contribute to BBS disease severity.-

Referee #2:

The reviewer thanks the authors for their detailed point by point response which partially answered to his/her comments on the first submitted version of their manuscript. Accordingly, in the new version, Martina Huranova and colleagues have included a set of new data. The main point raised by the three previous reviewers was to provide evidence that the observed differences in cilium length and the presence of what they call "foci" at the tip of cilia in BBS KO cells were indeed linked to ectocytosis, i.e. production of ciliary-derived ectosomes/EVs. The new data presented in Figs.1,4,5 aimed to solve this important point but raise some concerns.

Fig.1K,L,M:

The authors quantified the production of EVs by WT and Bbs4 KO cells treated or not with ML141. They used a widely used protocol consisting of sequential centrifugations which is known to allow isolation of all secreted small EVs (P100) including those originating from the plasma membrane (CD9+), endosomes (exosomes, TSG101+) and cilia.

Unfortunately, the way those EVs were analyzed is not accurate. The authors should have used negative controls to be sure that what they have purified are really EVs and not cell debris. Endoplasmic reticulum markers (calnexin) which are normally not found in EVs are commonly used to control the specificity of EV preps. This important control is lacking preventing clear interpretation of the shown results.

In addition, it is not clear why the authors have used GPR161-cherry stable cell lines treated with SAG to study production of ciliary EVs. All the data presented in Figure 1 were generated to characterize the BBS-associated cilia shortening in non-treated cells. The main problem with this experimental setting is that this leads to a protocol for EV purification in which only a 2.5 h production is analyzed since cells were washed in PBS before SAG stimulation. This is a very short time course for EV production likely explaining why TSG101 was not detected.

The lack of TSG101 is not expected since EV from endosomes (exosomes) are co-purified with the ciliary ones, which represent a minor fraction of total EVs, and TSG101 should have been therefore detected as a major marker of exosomes.

The fact that three different bands are observed with the IFT88 Ab raises some doubts about its specificity. Additional controls for this Ab must be provided. The authors should test other endogenous ciliary markers with trustable Abs. Finally, the analyses in L and M indicate that most of the observed differences are not statistically significant.

Figs.4&5:

The results presented in Fig.4 are based on stable cell lines overexpressing GPR161-mCherry. It is difficult to clearly understand the rationale for this choice since endogenous GPR161 can be easily followed and that overexpression of this key Hh signaling component is likely to lead to a major perturbation of Hh signaling as shown by the fact that overexpressed GPR161 is not removed from cilia upon SAG treatment. Recently published data showed that cilium length is tightly regulated or affected upon Hh activation due to differential ciliary cAMP levels linked to GPR161 removal and subsequent activation of ciliary prostaglandin receptors (PMID: 38856684). Indeed, likely due to the persistence of GPR161, cilia are not shortening after SAG treatment in WT cells overexpressing GPR161 as shown in Fig.4E. Based on these observations, the results obtained with the GPR161 overexpression system are questionable. Why the authors did not use the ARL13B-neonGreen fusion that they used in Fig.5? I am not sure if the quantification and statistical analyses provided in Figs. 4D and 5DE are accurate (only two independent experiments).

Additional points:

- The authors should more precisely define what they called and quantified as "foci". What is the difference between "foci" and a classical staining of a ciliary membrane marker at the ciliary tip? This is specifically needed in conditions such as in Fig.11 where Bbs4 KO cells in which Ac-tub staining is very short relative to ciliary membrane, how foci are defined in this case? This also raises the question of the marker that was used to quantify cilia length in these cells? Ac-tub or Arl13b?
- Regulation of cilium length is not only linked to ectocytosis. Several publications showed that the ciliary prostaglandin receptor EP4 positively regulates cilium length by increasing ciliary cAMP and IFT-mediated transport (PMID: 34927727). Recently, this pathway was shown to counterbalance the Hh-mediated modulation of ciliary length (PMID: 38856684). These data should be taken into account to avoid a biased ectocytosis-centric view of cilium length regulation. What if ciliary EP4 are lost in BBS conditions?

Referee #3:

In the revised manuscript by Prasai et al. entitled "BBSome-deficient cells activate intraciliary CDC42 to trigger actin-dependent ciliary ectocytosis", the authors managed to address most of the concerns.

However, there are still some points that have not been fully addressed:

- The authors mention that they have analyzed nuclear actin remodeling, but I somehow cannot find the results. These should be included at least in the supplemental data.

- In the first round, I raised a question to Figure 2E: Why are Bbs4-KO cells Shh-responsive with respect to ciliary length? According to the GPR161 data, GPR161 accumulates in the cilium and does not exit the cilium in Bbs4-KO cells. The authors stated that "we did not observe the accumulation of GPR161, as we did not quantify the GPR161 staining intensity in these cells. We observed increased foci formation (Fig. 2C, Fig. 4B)." Maybe there is a misunderstanding but clearly, Fig. 2F shows that GPR161 is not removed from Bbs4-KO cilia, demonstrating a defect in Hh signaling. In turn, this should impair Hh signaling and its downstream effects. If cilia shortening is SAG/Hh signaling dependent, why should this exert a similar effect in Bbs4-KO cells?

- In the first round, I raised a question to Figure 4C: according to the text, this figure shows no difference in WT ciliary length upon SAG treatment whereas Figure 2G shows a significant decrease in ciliary length. Please comment on these differences! This makes the interpretation of Figure 4C for Bbs4-KO cells rather difficult.

The authors stated that: "The cilia shortening triggered by SAG in WT MEFs is probably caused by removal of specific receptors, including the GPR161, from cilia via endocytosis. In addition, the SHH pathway is mitogenic and its activation can stimulate cilia resorption at the ciliary pocket. Our observations thus suggest that SAG induced cilia shortening can be

counteracted by overloading the ciliary compartment either generally with cargo/membrane material or specifically with GPR161 receptor acting inhibitory to SHH activation. Our over-expression system thus allows us to neglect the contribution of endocytosis to cilia shortening (Saito et al 2017 PMID: 28607034) and offers an advantage to examine the effect of CDC42 inhibition specifically on the cilia shortening mediated by ectocytosis."

Again, the author might not have understood my point: the authors described that Figure 4 shows no difference in cilia length in WT cells upon SAG treatment - what is the explanation for that? I am even more confused now than before.

- Figure 5C-E: Is this n = 1? Mean +/- S.D.?

Point-by-point response

Referee #2:

The reviewer thanks the authors for their detailed point by point response which partially answered to his/her comments on the first submitted version of their manuscript. Accordingly, in the new version, Martina Huranova and colleagues have included a set of new data. The main point raised by the three previous reviewers was to provide evidence that the observed differences in cilium length and the presence of what they call "foci" at the tip of cilia in BBS KO cells were indeed linked to ectocytosis, i.e. production of ciliary-derived ectosomes/EVs. The new data presented in Figs.1,4,5 aimed to solve this important point but raise some concerns.

We thank to the reviewer for insightful comments.

Fig.1K,L,M:

The authors quantified the production of EVs by WT and Bbs4 KO cells treated or not with ML141. They used a widely used protocol consisting of sequential centrifugations which is known to allow isolation of all secreted small EVs (P100) including those originating from the plasma membrane (CD9+), endosomes (exosomes, TSG101+) and cilia.

Unfortunately, the way those EVs were analyzed is not accurate. The authors should have used negative controls to be sure that what they have purified are really EVs and not cell debris.

Endoplasmic reticulum markers (calnexin) which are normally not found in EVs are commonly used to control the specificity of EV preps. This important control is lacking preventing clear interpretation of the shown results.

Although we believe that the CD9 and TSG101 showing no enrichment in the Bbs4 KO cells is a sufficient control controlling also for the cell debris, we probed the EV fractions with the antibody to Calnexin as suggested by this Reviewer. We observed that the P100 fraction is largely devoid of ER/cell debris in comparison to EV fraction, and there was no detectable enrichment in Bbs4 KO cells relative to WT cells. The data are now included in the Figure 1K and respective Source data file.

In addition, it is not clear why the authors have used GPR161-cherry stable cell lines treated with SAG to study production of ciliary EVs. All the data presented in Figure 1 were generated to characterize the BBS-associated cilia shortening in non-treated cells. The main problem with this experimental setting is that this leads to a protocol for EV purification in which only a 2.5 h production is analyzed since cells were washed in PBS before SAG stimulation. This is a very short time course for EV production likely explaining why TSG101 was not detected.

The lack of TSG101 is not expected since EV from endosomes (exosomes) are co-purified with the ciliary ones, which represent a minor fraction of total EVs, and TSG101 should have been therefore detected as a major marker of exosomes.

We need to use a brief pulse of SHH activation because the long-term cultivation of cells in the presence of the ML141 inhibitor leads to the cell death, making the typical overnight protocol unsuitable regardless of using GPR161-mCherry transgenic or nontransgenic lines. Moreover, since the formation of ciliary ectosomes is triggered by SAG, it could be expected that the ratio of newly formed ciliary ectosomes to the continuously produced EVs and cellular debris will decrease upon a prolonged incubation.

The SAG-activated GPR161-mCherry expressing cell lines enabled us to detect GPR161+ ectosomes after such a short pulse using the anti-mCherry antibody. Thus, this setup enables us to see the accumulation of ciliary proteins IFT88 and GPR161, but not classical exosome proteins CD9 and TSG101 or a debris-born ER marker in the EV fraction of the SAG treated Bbs4 KO cells. It is difficult for us to imagine that the data could be even more convincing. We believe that establishing this protocol based on a short pulse is one of the important advances of our study and that it may become a standard for studying ciliary ectosomes.

We are not aware of any study that ever measured the ratio between exosomes and ciliary ectosomes. Anyway, this ratio likely varies with experimental conditions, cell type, sensitivity of the detection of particular protein markers, and, as mentioned above, timing. As we used the SAG-treated serum-starved ciliated MEFs in the two-hour and 30 minutes pulse and observed the particularly strong IFT88 and GPR161 signal in the SAG treated Bbs4 KO cell, we believe that our experimental system is unique in multiple parameters and the ratio between the ciliary ectosomes and exosomes cannot be easily predicted.

We believe that the low TSG101 signal is caused by the brief duration of the EV production in combination with the suboptimal performance of the antibody. Importantly, even published studies (Nager et al. 2017 PMID: 28017328, Volz et al. 2021 PMID: 34580290) do not show a particularly strong TSG101 signal in EV fractions obtained from overnight protocols using serum-starved ciliated cells. Thus, we consider our results consistent with existing research data.

The fact that three different bands are observed with the IFT88 Ab raises some doubts about its specificity. Additional controls for this Ab must be provided. The authors should test other endogenous ciliary markers with trustable Abs. Finally, the analyses in L and M indicate that most of the observed differences are not statistically significant.

We validated the IFT88 antibody using Ift88-deficient ST2 mouse cells that we generated in the lab for a different project. Since ST2 cells, like MEFs, are of mesenchymal origin, the staining profile of IFT88 under the conditions used for EV production analysis is nearly identical in both non-transfected cell types. This experiment showed that the middle band corresponds to IFT88 and the lower and upper bands might be non-specific. These data have now been included in the manuscript as Figure EV1J-K.

Altogether, we have two different antibodies – validated anti-IFT88 (endogenous) and anti-GPR161-mCherry (exogenous) for the detection of the ciliary ectosomes. We are convinced that this evidence is sufficient for our conclusions, considering the amount of ectocytosed material in our pulse study is very limited.

Concerning the statistical significance, the data show that SAG induces the shedding of ciliary material using the two markers. In the case of extracellular GPR161-mCherry, we observed on average ~1.72 fold increase in KO cells with a strong statistical significance ($p=0.0054$). Upon the CDC42 inhibition, this effect was largely lost (~1.32 fold increase, $p=0.23$). In the case of IFT88 we observed ~ 3.61 fold increase and statistical significance of $p=0.08$. Again, this effect was lost upon CDC42 inhibition (~0.9 fold change, $p=0.31$). Thus, one of the two markers shows undisputed statistical significance, whereas the other marker shows a relatively low $p=0.08$, which is however, above the commonly used arbitrary threshold of 0.05. The p values for other comparisons are not essential for the interpretation of the results and are shown just for overview – we were motivated by the previous comment of this Reviewer asking us to show p -values for additional comparisons in

some other figures. On a side note, the *p*-values between SAG-treated KO cells with and without the CDC42 inhibitor were also relatively low (0.053 and 0.13).

*It is well established that the *p*-value-based frequentist statistics is not suitable for the experimental biological research (e.g., PMID: 29302092, PMID: 27695640). Moreover, there is still a confusion that the *p* value is being used consciously or unconsciously for two different purposes in the scientific publishing – for dichotomous hypothesis testing and as a descriptive statistical parameter enhancing the intuitive interpretation of the data. The strict usage of the frequentist statistics for experimental biological research is harmful for many reasons and leads to various ways of *p*-hacking. In accord with the currently prevailing opinion, we use *p*-value as a descriptive parameter and we do not establish any threshold for significance in the sense of hypothesis testing. We believe this is in line with the EMBO Reports policy saying: "...must state (...) the actual *P* value for each test (not merely 'significant' or '*P* < 0.05')". Our approach is supported by multiple articles, from which we select three examples:*

1. Wasserstein and Lazar (*The ASA's statement on *P*-values: Context, process, and purpose. Am Stat. 2016 70:129–33.*) wrote: "Scientific conclusions and business or policy decisions should not be based only on whether a *p*-value passes a specific threshold."

2. Greenland et al. (PMID: 27209009) wrote: "Nonetheless, the *P* value can be viewed as a continuous measure of the compatibility between the data and the entire model used to compute it, ranging from 0 for complete incompatibility to 1 for perfect compatibility, and in this sense may be viewed as measuring the fit of the model to the data. Too often, however, the *P* value is degraded into a dichotomy in which results are declared "statistically significant" if *P* falls on or below a cut-off (usually 0.05) and declared "nonsignificant" otherwise."

3. Amerheim et al. (PMID: 30894741) wrote: "When *P* values are reported, they will be given with sensible precision (for example, *P* = 0.021 or *P* = 0.13) — without adornments such as stars or letters to denote statistical significance and not as binary inequalities (*P* < 0.05 or *P* > 0.05). Decisions to interpret or to publish results will not be based on statistical thresholds."

*Perhaps, this Reviewer has a similar opinion on the *p*-value usage in our field, as they asked us in the previous round of review to show additional *p*-values which were not calculated in the original manuscript. Such requests would not be meaningful in the case of the strict utilization of *p*-value for the dichotomous hypothesis testing, as we did not formulate any hypotheses around these comparisons beforehand.*

*According to its definition, the *p*=0.08 in the IFT88 immunoblot data means that if there was actually no accumulation of extracellular IFT88 induced by the SAG treatment in the KO cells vs WT cells (and if all other assumptions of the statistical test are true), there would be a chance of 8% to observe the same or larger difference as we measured. For us, these results, together with the GPR161 analysis, are very convincing to draw the conclusion that SAG does induce the ciliary ectocytosis in the KO cells and that this is prevented by CDC42 inhibition. However, we are providing the *p* values (and source data) that the reader could easily interpret the results according to their experience.*

Finally, we would like to stress that our analysis goes beyond the common practise in this field as we, to our best knowledge, are the first ones to provide any quantification and statistical analysis of the production of ciliary ectosomes in different conditions. The previous studies presented data originating from a single representative experiment without showing reproducibility (Nager et al. 2017 PMID: 28017328, Loukil et al. 2021 PMID: 33846249).

Figs.4&5:

The results presented in Fig.4 are based on stable cell lines overexpressing GPR161-mCherry. It is difficult to clearly understand the rationale for this choice since endogenous GPR161 can be easily followed and that overexpression of this key Hh signaling component is likely to lead to a major perturbation of Hh signaling as shown by the fact that overexpressed GPR161 is not removed from cilia upon SAG treatment. Recently published data showed that cilium length is tightly regulated or affected upon Hh activation due to differential ciliary cAMP levels linked to GPR161 removal and subsequent activation of ciliary prostaglandin receptors (PMID: 38856684).

Indeed, likely due to the persistence of GPR161, cilia are not shortening after SAG treatment in WT cells overexpressing GPR161 as shown in Fig.4E. Based on these observations, the results obtained with the GPR161 overexpression system are questionable. Why the authors did not use the ARL13B-neonGreen fusion that they used in Fig.5?

We utilized overexpression of GPR161 to be able to complement the data obtained by staining the endogenous GPR161 and connect the ectocytosis to cilia shortening in vivo. Exogenous expression of the BBSome-dependent cargoes, such as GPR161, is established in the field (Nager et al. 2017 PMID: 28017328). We are not sure what this reviewer means "by endogenous GPR161 can be easily followed" in the live cell imaging experiments. Even if we had an antibody binding to the extracellular part of GPR161, binding of the antibody might influence GPR161 signaling, dynamics, localization etc. which would interfere with the live cell imaging assay.

In our study, we focus specifically on actin dependent regulation of ectocytosis. Here, our overexpression system with the high rate of GPR161 ectocytosis in the BBSome deficient cells allowed us to monitor the role of CDC42 in this process in live cells. Furthermore, overexpression and saturation of cilia with GPR161-mCherry enabled us also to examine the capacity of cilia to handle and remove cargoes through ectocytosis. The Ref #1 in the first round of reviews was actually interested in this point and appreciated this experimental system.

We are aware that the overexpression might interfere with the signaling. However, SMO was normally imported to cilia upon SHH activation in the WT MEFs overexpressing GPR161, documenting that the SAG-mediated signaling is not completely OFF, so we added this information to the Figure 4A. The increased import of GPR161 might explain why we do not see cilia shortening in this condition. While, we see that SHH triggering leads to cilia shortening in WT cells in our study, this is not always the case in the mentioned study (PMID: 38856684). This suggests that the SHH and EP4 signaling might control cilium length in a context-dependent manner and that other factors regulating cilium length might contribute to the final outcome.

The reason why we performed these experiments using tagged GPR161 was that we wanted to monitor the trafficking/ectocytosis of the specific cargo that we know is targeted to the ciliary ectosomes. If we used tagged ARL13B, the reviewers might have asked whether the ARL13B+ ectosomes contain GPR161.

We would like to stress here that we did not study the mechanism of cilia shortening in SAG-treated WT cells. We studied the mechanism of cilia shortening in the BBSome-deficient cells, which is clearly different from the WT cells (CDC42-dependent) and which is induced by SAG independently of GPR161 over-expression (Fig. 2E, G). The major conclusions of the study are based on complementary and consistent data obtained from cell models with endogenous GPR161 and cells over-expressing GPR161.

We clarified the rationale behind our experimental setup and acknowledge the limitations of this system in the revised version of the manuscript.

I am not sure if the quantification and statistical analyses provided in Figs. 4D and 5DE are accurate (only two independent experiments).

The data described in Fig. 4D originate from 2 independent experiments, whereas the experiments shown in Fig. 5D-E originate from 4 independent experiments. Doing the live cell imaging experiments, we were primarily interested in whether an event occurs in a given cell or not (binary output). Since capturing these events is rare, such live cell imaging experiments including multiple conditions – 8 in Fig. 4D, are very demanding. For this reason, we decided to perform two independent experiments, pooled the cells from these experiments and performed the statistical analysis for the binary outcome on a single cell level. We consider this approach appropriate for this case, as Nager et al. (2017) quantifies the ciliary foci formation also as a cumulative frequency even without indicating the number of independent experiments. Phua et al. (2017) analyzed live-cell imaging experiments to monitor cilia length during its resorption in a similar way and in some cases pooled data from as few as 2 independent experiments.

Additional points:

- The authors should more precisely define what they called and quantified as "foci". What is the difference between "foci" and a classical staining of a ciliary membrane marker at the ciliary tip? This is specifically needed in conditions such as in Fig.1I where Bbs4 KO cells in which Ac-tub staining is very short relative to ciliary membrane, how foci are defined in this case? This also raises the question of the marker that was used to quantify cilia length in these cells? Ac-tub or Arl13b?

The foci are defined by an increase of the ARL13B signal (spanning around 0.5 μ m) with a concomitant drop in the Ac-tub signal as now shown in the Fig. 1G (right) of the revised manuscript and described in the text. This approximate size fits with the approximate size of the foci and ectosomes observed during the live cell imaging (Fig. 4C). We present similar line scan for the Bbs4 KO MEFs treated with ML141 (Fig. 1I) as a Fig. EV1I to show that the ciliary membrane can extend around 0.5 μ m and more beyond the axoneme indicating the sustained ectocytosis under this condition.

We measured the cilia length using either the ARL13B staining (including Fig. 1I-J, Fig. EV1G, Fig. EV2A) or the merge of the GPR161 and Ac-tub signal when assessing the effect of SAG treatment (Fig. 2E, G) to include even the GPR161+ foci at the terminus. In Fig. 3F, we measured the cilia length based on the signal of the N-Raichu-CDC42 probe during the FRET-FLIM measurements. We clarified this in the Methods of the revised manuscript.

- Regulation of cilium length is not only linked to ectocytosis. Several publications showed that the ciliary prostaglandin receptor EP4 positively regulates cilium length by increasing ciliary cAMP and IFT-mediated transport (PMID: 34927727). Recently, this pathway was shown to counterbalance the Hh-mediated modulation of ciliary length (PMID: 38856684). These data should be taken into account to avoid a biased ectocytosis-centric view of cilium length regulation. What if ciliary EP4 are lost in BBS conditions?

This comment was not raised during the first round of revisions. Moreover, one of these articles was published very recently, after we have finished our study.

Our study specifically focuses on the regulation of intraciliary actin polymerization during ectocytosis in BBSome-deficient cells. We acknowledged other mechanisms contributing to the regulation of cilium length in the revised manuscript.

We agree with the reviewer that the interplay between EP4 and the BBSome and BBSome-dependent pathways would be an interesting area of research. However, it would go beyond the current scope of our manuscript as we find it unlikely that EP4 mislocalization could be the main mechanism behind the cilia shortening in the BBSome-deficient cells for the following reasons:

1. Similar to GPR161, the ciliary localization of EP4 is facilitated by Tubby (Wu et al 2021, PMID: 34385262). The loss of the BBSome does not affect Tubby-mediated import of GPR161, suggesting that EP4 import is independent of the BBSome, though this has yet to be confirmed.

2. The BBSome-mediated export of GPR161 operates independently of EP4 as was demonstrated in EP4 knockout MEFs upon SHH activation (Ansari et al 2024, PMID: 38856684).

3. The expression of EP4 greatly varies across tissues, whereas the cilia shortening has been observed in a variety of different BBSome-deficient cell types such as mouse renal medullary cells (Hernandez-Hernandez et al 2013, PMID: 23716571), mouse fibroblasts and medaka fish neural tube cells (Chiuso et al 2023, PMID: 36744302), human RPE1 cells (Prasai et al 2020, PMID: 32759308) and mouse olfactory sensory neuron cells (Uytingco et al 2019, PMID: 30665891).

4. The mechanism of SAG-induced cilia shortening is different in WT and the BBSome-deficient cells. Only in the latter case, it is CDC42-dependent, occurs independently of GPR161-overexpression, and correlates with the intraciliary actin polymerization events (Fig. 5C, E) and ectosome formation (Fig. 4D) as captured by live cell microscopy.

Referee #3:

In the revised manuscript by Prasai et al. entitled "BBSome-deficient cells activate intraciliary CDC42 to trigger actin-dependent ciliary ectocytosis", the authors managed to address most of the concerns.

We thank to this Referee for acknowledging that we have addressed most of the concerns; the detailed feedback indeed helped us to improve our manuscript.

However, there are still some points that have not been fully addressed:

- The authors mention that they have analyzed nuclear actin remodeling, but I somehow cannot find the results. These should be included at least in the supplemental data.

As mentioned in the first round of the revisions, the remodeling of the nuclear actin via ciliary signaling is an interesting and still unexplored research direction, however it is beyond the scope of our manuscript as we focused on cilia-proximal events, such as the regulation of intraciliary actin polymerization. In the previous point-by-point response, we mentioned that "we did not observe any apparent differences of the signal in the nucleus in the experiments, where we performed the F-actin staining in various conditions". Here, we present examples of F-actin patterning in the nuclear area (Figure 1). Since we neither anticipated nor observed striking differences between the tested

conditions, we did not pursue this aspect further. Additionally, the LifeAct probe may not be ideal for detecting the changes in nuclear actin remodeling due to its strong cytoplasmic signal. A more targeted approach, such as using antibodies against nuclear actin or actin remodeling factors (e.g.,

adducin – Sheu et al 2022, PMID: 36055200), would be more effective in exploring nuclear actin remodeling during SHH signaling in greater detail. We prefer keeping the images of F-actin patterning in the Point-by-Point section of this manuscript, which would be published along the article anyway. Alternatively, we can publish these images as Supplemental material. We defer this decision to the editor.

Figure 1. Examples of the F-actin patterning (red) in the nucleus area depicted by the DAPI staining (grey) in non-treated (nt) and SAG (SAG 2h) treated WT and Bbs4 KO MEFs expressing the LifeAct-TagRFP. Scale bar, 5 μ m.

- In the first round, I raised a question to Figure 2E: Why are Bbs4-KO cells Shh-responsive with respect to ciliary length? According to the GPR161 data, GPR161 accumulates in the cilium and does not exit the cilium in Bbs4-KO cells.

The authors stated that "we did not observe the accumulation of GPR161, as we did not quantify the GPR161 staining intensity in these cells. We observed increased foci formation (Fig. 2C, Fig. 4B)." Maybe there is a misunderstanding but clearly, Fig. 2F shows that GPR161 is not removed from Bbs4-KO cilia, demonstrating a defect in Hh signaling. In turn, this should impair Hh signaling and its downstream effects. If cilia shortening is SAG/Hh signaling dependent, why should this exert a similar effect in Bbs4-KO cells?

Although we observed cilia shortening in both cell lines, we believe that it results from different mechanisms, as the cilia in WT and Bbs4 KO cells respond differently to the CDC42 inhibition. Our data show that in Bbs4 KO cells, but not in the WT cells, the cilia shortening is significantly influenced by ectocytosis. In WT cells, the decrease in cilium length during signaling would be likely caused by endocytosis of PTCH1, GPR161 and other signaling molecules (Pal et al. 2016, PMID: 27002170, Zue et al. 2014, PMID: 24925320) and/or EP4 signaling in some contexts (Ansari et al 2024, PMID: 38856684). In BBSome deficiency, GPR161 exits the cilium in ectosomes. We believe that new

GPR161 molecules are imported to the cilia from the cytoplasm resulting in the comparable signal intensities along the cilium. However, as the cilia is shortened in these conditions the total amount of ciliary GPR161 is very likely decreased. This is conceptually intriguing, as it implies that the abrupt drops in cAMP levels due to the sudden removal of GPR161 via ectosomes are sufficient to maintain decent levels of SHH signaling in the BBSome-deficient conditions (Zhang et al. 2012 PMID: 22228099, Nager et al. 2017 PMID: 28017328). We discuss this issue in the revised manuscript.

- In the first round, I raised a question to Figure 4C: according to the text, this figure shows no difference in WT ciliary length upon SAG treatment whereas Figure 2G shows a significant decrease in ciliary length. Please comment on these differences! This makes the interpretation of Figure 4C for Bbs4-KO cells rather difficult.

The authors stated that: "The cilia shortening triggered by SAG in WT MEFs is probably caused by removal of specific receptors, including the GPR161, from cilia via endocytosis. In addition, the SHH pathway is mitogenic and its activation can stimulate cilia resorption at the ciliary pocket. Our observations thus suggest that SAG induced cilia shortening can be counteracted by overloading the ciliary compartment either generally with cargo/membrane material or specifically with GPR161 receptor acting inhibitory to SHH activation. Our over-expression system thus allows us to neglect the contribution of endocytosis to cilia shortening (Saito et al 2017 PMID: 28607034) and offers an advantage to examine the effect of CDC42 inhibition specifically on the cilia shortening mediated by endocytosis."

Again, the author might not have understood my point: the authors described that Figure 4 shows no difference in cilia length in WT cells upon SAG treatment - what is the explanation for that? I am even more confused now than before.

The major difference in these experiments is that Figure 2G shows fixed cells expressing endogenous GPR161, yet in presence of cilia targeted WT CDC42, after a 2 hour incubation in a CO2 incubator, whereas the Figure 4C, E is the continuous live cell imaging of GPR161-expressing cells in the microscopy chamber. We believe that the differences in the WT cells could be caused by two factors.

1. The overexpression of GPR161. The increased import of GPR161-mCherry likely saturates cilia both in WT and BBSome deficient cells and therefore the signal of the tagged version does not decrease so apparently as its endogenous counterpart (Ye et al 2018, PMID: 29483145). Moreover, the overexpression of GPR161 likely increases cAMP levels, which might inhibit EP4 signaling (Ansari et al 2024, PMID: 38856684).

2. The imaging chamber might not completely mimic the stable conditions of the CO2 incubator, which, together with some level of phototoxicity, might lead to some alterations of the cell physiology.

It is important to mention that we did not study the mechanism of cilia shortening in SAG-treated WT cells. We studied the mechanism of cilia shortening in BBSome-deficient cells, which is clearly different from the WT cells (CDC42-dependent) and which is induced by SAG independently of GPR161 over-expressing cells (Fig. 2E, G and also Fig. 3F). We observed the cilia shortening in the BBSome-deficient cells in both type of experiments (Figure 2E, G and 4E), which is important for the validity of our conclusions.

We discuss the discrepancy between the Figure 2E/G and 4E in WT cells in the revised version of the manuscript.

- Figure 5C-E: Is this $n = 1$? Mean \pm S.D.?

The number of independent experiments for the movie acquisition experiments is indicated in the figure legends and is larger than $n=1$:) (Fig. 5C,D: $n=4$; Fig. 5E: $n=6$). We included the individual values of the estimated fractions of actin polymerization events within the individual imaging sessions. Additionally, in panel D, the bars now represent both the mean and standard deviation in the revised manuscript.

Manuscript number: EMBOR-2023-58108V4

Title: BBSome-deficient cells activate intraciliary CDC42 to trigger actin-dependent ciliary ectocytosis.

Author(s): Martina Huranova, Avishek Prasai, Olha Ivashchenko, Kristyna Maskova, Sofii Bykova, Marketa Cernohorska, and Ondrej Stepanek

Dear Dr. Huranova

Thank you for your patience while we have reviewed and further discussed your revised manuscript with the referees and with the editorial team. As you will see from the reports below, both referees have remaining concerns on the interpretation of your data, in particular regarding the fact that Bbs4 KO cells are still Shh-responsive and why ciliary length is different when cells are fixed versus live imaging.

Referee 2 agreed with the concerns from Referee 3 and also noted again that your results rely on using the stable GPR161-Cherry line, which you used not only for live cell imaging but also for the production of EVs (Fig. 1F) as well as on fixed cells (Fig. 4A, B) for ciliary foci quantification in experiments which are not based on live cell analysis and do not require expression of a fluorescent GPR161.

Therefore, the argumentation that "We are not sure what this reviewer means "by endogenous GPR161 can be easily followed" in the live cell imaging experiments." seems not to fully address the concern raised by Referee 2 during the previous round of peer review:

"Indeed, likely due to the persistence of GPR161, cilia are not shortening after SAG treatment in WT cells overexpressing GPR161 as shown in Fig.4E" and "overexpression of this key Hh signaling component is likely to lead to a major perturbation of Hh signaling as shown by the fact that overexpressed GPR161 is not removed from cilia upon SAG treatment. Recently published data showed that cilium length is tightly regulated or affected upon Hh activation due to differential ciliary cAMP levels linked to GPR161 removal and subsequent activation of ciliary prostaglandin receptors (PMID: 38856684)."

As indicated, I discussed these concerns further with the referees, who did not oppose publication but the remaining concerns need to be discussed in the manuscript and in a point-by-point response and conclusions need to be toned down or adjusted as appropriate.

I am now writing with an 'accept in principle' decision, which means that we will proceed with publication, pending that the above concerns are addressed.

Once you have made these revisions, please use the following link to submit your corrected manuscript:

Link Not Available

If all remaining corrections have been attended to in a satisfactory manner, we can proceed with publication of your manuscript in the next available issue of EMBO reports. The following decision letter will then also include details of the further steps you need to take for the prompt inclusion of your manuscript in our next available issue.

Thank you for your contribution to EMBO reports.

Yours sincerely,

Referee #2:

The authors have responded to most of the comments/questions I raised on earlier versions of the manuscript. Although I do not fully agree with some of their responses, I believe this revised version should be accepted for publication.

Referee #3:

In the revised manuscript by Prasai et al. entitled "BBSome-deficient cells activate intraciliary CDC42 to trigger actin-dependent ciliary ectocytosis", the authors managed to address some of the concerns, although I have to say that some issues remain because the authors did not directly address my question.

I still struggle with their answer to my question why Bbs4-KO cells are Shh-responsive with respect to ciliary length? If this is downstream of Hh signaling and Bbs4-KO cilia clearly show a defect in Hh signaling, shouldn't this impair Hh signaling and its downstream effects, i.e., cilia shortening? If cilia shortening is SAG/Hh signaling dependent, why should this exert a similar effect in Bbs4-KO cells?

The authors stated that "Although we observed cilia shortening in both cell lines, we believe that it results from different mechanisms, as the cilia in WT and Bbs4 KO cells respond differently to the CDC42 inhibition." But there is not data or explanation what this mechanism might be. As this is a key read-out for the whole phenotype that they analyze, I wonder whether this statement is sufficient to address this remaining issue.

- I am also a bit puzzled by their answer to the issue that Figure 4C showed no difference in WT ciliary length upon SAG treatment whereas Figure 2G shows a significant decrease in ciliary length.

The authors state that this is due to different experimental conditions, i.e., fixed cells vs. live cell imaging. If the phenotype is robust, shouldn't it be the same independent of the condition? And this relates also to my previous point regarding the potential mechanisms underlying SAG-dependent cilia shortening in WT vs. Bbs4-KO cells. I somehow find it still confusing to combine all these explanations and findings in one model.

In summary, although I am not fully convinced, I would consider the manuscript for publication.

Point-by-point response

Referee #2

The authors have responded to most of the comments/questions I raised on earlier versions of the manuscript. Although I do not fully agree with some of their responses, I believe this revised version should be accepted for publication.

We thank the reviewer for their thoughtful review and feedback. We understand that full agreement may not always be reached on every point, and we appreciate the reviewer's support in moving forward with our revised manuscript and their recommendation for acceptance.

Finally, we consider the statement in the "revise only decision communication" that "(our) results rely on using the stable GPR161-Cherry line" as an exaggeration.

Concerning Fig. 4A-B, similar experiments with similar conclusions were performed in Fig. 2C and using endogenous GPR161. These experiments show SAG-dependent foci formation in BBS4 KO cells (Fig. 2C) and the dependency of these foci on CDC42 (Fig. 2I). The only major difference between Fig. 2I and Fig. 4A-B is the use of the genetic CDC42 inhibitor in Fig. 2I and the chemical CDC42 inhibitor in Fig. 4A-B. As the results from both endogenous and overexpressing GPR161 BBS-deficient cells are in a good agreement, we do not think that it could be objectively seen as an issue.

Comparable results in Figs. 2 and 4 in terms of the role of SAG and CDC42 in the foci formation in the BBSome-deficient cells justify the use of GPR161-overexpressing cells for the following experiments. The IF analysis in fixed cells overexpressing GPR161 in Fig. 4A was also important for showing that SMO is imported into cilia in these cells, which is another layer of validation.

Obviously, tagged GPR161 is required for live cell imaging, as explained previously. The use of the tagged line was required to increase the sensitivity of the GPR161 for the detection of ectosomes in the short pulse experiments. The ectosome quantification experiments were proposed by the referees in the first round of revision. As was revealed afterwards, the Referees suggested this experiment apparently without realizing how extremely demanding these experiments would be. We had to go for a short pulse because of the toxicity of the CDC42 inhibitor, which substantially limits the amount of shed ectosomes and decreases the detection limit of the readout. On the other hand, this assay seems to be cleaner than protocols which collect extracellular vesicles for a prolonged period of time, which might lead to the accumulation of cellular debris and death cell fragments. We still performed these challenging experiments and obtained valuable results confirming our previous conclusions. Overall, only a minority of our experiments in our manuscript rely on the GPR161 overexpressing lines whereas the majority experiments are done exclusively or additionally with cells not overexpressing GPR161. Altogether, all these results are consistent and support our conclusions.

Referee #3

In the revised manuscript by Prasai et al. entitled "BBSome-deficient cells activate intraciliary CDC42 to trigger actin-dependent ciliary ectocytosis", the authors managed to address some of the concerns, although I have to say that some issues remain because the authors did not directly address my question.

We thank to the reviewer for considering our revised manuscript for publication.

I still struggle with their answer to my question why Bbs4-KO cells are Shh-responsive with respect to ciliary length? If this is downstream of Hh signaling and Bbs4-KO cilia clearly show a defect in Hh signaling, shouldn't this impair Hh signaling and its downstream effects, i.e., cilia shortening? If cilia shortening is SAG/Hh signaling dependent, why should this exert a similar effect in Bbs4-KO cells? The authors stated that "Although we observed cilia shortening in both cell lines, we believe that it results from different mechanisms, as the cilia in WT and Bbs4 KO cells respond differently to the CDC42 inhibition." But there is not data or explanation what this mechanism might be. As this is a key read-out for the whole phenotype that they analyze, I wonder whether this statement is sufficient to address this remaining issue.

It is difficult for us to understand the critical concerns of the Reviewers if they are too brief and not providing a clear explanation and reasoning. We are not sure, if the Ref. 3 refers to the early signaling events in the cilia (such as translocation of SMO inside the cilium), intermediate steps connected to the export of GPR161 out of cilia and GLI activation, or very down-stream signaling including transcriptional changes.

The early events such as SMO translocation to the cilium are preserved in the BBSome-deficient cells (Fig. 4A). The downstream effects of the BBSome-deficiency on the transcriptional response to SAG are also relatively minor. A study by Zhang et al. (PMID: 22228099, Fig. 1D) showed only 20-30% decrease in the SAG-induced GLI1 and PATCH transcription in BBS7 KO cells in comparison to WT cells. We have observed similar results using WT and Bbs4 KO MEFs (not shown, but could be included if requested by the editor). Overall, the down-stream SHH signaling is not substantially impaired in the BBSome-deficient cells.

We concluded that the cilia shortening in the BBSome-deficient cells occurs via ectocytosis, i.e., via a mechanism, which was previously proposed for multiple conditions when the signaling receptor could not be removed from the cilia by the retrograde transport (Nager et al., PMID: 28017328). There is no indication that this actually depends on the down-stream SHH signaling/transcriptional response.

It is true that it is unclear what induces the removal of GPR161 by ectocytosis, i.e., how the cell recognizes that GPR161 could not exit the cilium and subsequently triggers the ectocytosis. When SMO enters the cilium, it enhances β -arrestin recruitment to GPR161 for export from cilium (Pal et al. PMID: 27002170). However, the β -arrestin itself does not trigger ectocytosis (Nager et al., PMID: 28017328). While the exact mechanism of SMOdependent removal of GPR161 is unclear, it has been suggested that SMO and GPR161 may exit the cilium together as a bipartite receptor complex (Pal et al. PMID: 27002170). It is plausible that in the absence of the BBSome, SMO is imported into cilia (Fig. 4A), but cannot be exported (similarly to GPR161). As Nager et al. showed that ectocytosis is triggered by activated receptors unable to exit the cilium, the activity of entrapped SMO might trigger the ectocytosis. GPR161 is then removed in a bystander manner in the bipartite complex with SMO. Another possibility is that the ubiquitination of GPR161, which serves as an export signal (Shinde et al. PMID: 33185668), might trigger an unknown mechanism leading to GPR161 ectocytosis, when its retrograde export is blocked in the absence of the BBSome. We believe that a follow-up project should look into these two scenarios.

We revised the relevant parts of the manuscript and included these potential mechanistic explanations for GPR161 ectocytosis in the context of BBS, which should be addressed in follow-up studies.

- I am also a bit puzzled by their answer to the issue that Figure 4C showed no difference in WT ciliary length upon SAG treatment whereas Figure 2G shows a significant decrease in ciliary length. The authors state that this is due to different experimental conditions, i.e., fixed cells vs. live cell imaging. If the phenotype is robust, shouldn't it be the same independent of the condition? And this relates also to my previous point regarding the potential mechanisms underlying SAG-dependent cilia shortening in WT vs. Bbs4-KO cells. I somehow find it still confusing to combine all these explanations and findings in one model.

In summary, although I am not fully convinced, I would consider the manuscript for publication.

In this study, we focus on the mechanism driving the cilia shortening and ectocytosis in BBS condition. We observe consistently that in the BBSome-deficient cells, SHH activation induces ectocytosis and cilia shortening, which can be rescued by CDC42 inhibition. We further demonstrated that the increased activity of intraciliary CDC42 induces excess actin polymerization and ectocytosis in BBS condition.

The eventual cilia shortening induced by SAG in WT cells was not in the focus of our paper. We observed this in some assays (Fig. 2E, 2G), not in other assays (Fig. 4E). Interestingly, the study by Ansari et al. (PMID: 38856684), recommended to us by the Ref. 2, also observed SAG-induced cilia shortening in some (their Fig. 2D, 2E), but not in all assays (their Fig. 1C, 1D, 3D, 4D). Another study also shows SAG-induced cilia shortening in some, but not all, cell lines (Gomez et al. PMID: 36580465). These examples suggest that these observations are highly context dependent. This controversial question would need to be addressed in a separate thorough study. However, this was not the main point of our project.

It is important that even in the assays where we observed SAG-induced cilia shortening in WT cells, this was independent of CDC42. Based on these data, we concluded that SAG-induced cilia shortening via CDC42-dependent ectocytosis is specific to the BBS conditions in our experimental system. We revised the manuscript to clarify the major focus and findings of our study and highlighted the inconsistent results of us and others concerning SAG-induced cilia shortening in WT cells.

Dr. Martina Huranova
Institute of Molecular Genetics of the Czech Academy of Sciences
Laboratory of Adaptive Immunity
Videnska 1083
Prague 14220
Czech Republic

Dear Dr. Huranova,

I am very pleased to accept your manuscript for publication in the next available issue of EMBO reports. Thank you for your contribution to our journal.

Yours sincerely,
